

# Influence of Organic Compound Functionality on Aerosol Hygroscopicity: Dicarboxylic Acids, Alkyl-Substituents, Sugars and Amino Acids

Aleksandra Marsh[1], Rachael E. H. Miles[1], Grazia Rovelli[1], Alexander G. Cowling[1], Lucy Nandy[2],
Cari S. Dutcher[2] and Jonathan. P Reid[1]

[1] School of Chemistry, University of Bristol, Bristol, BS8 1TS, UK
[2] Department of Mechanical Engineering, University of Minnesota, 111 Church Street SE, Minneapolis, MN 55455, USA

*Correspondence to:* Jonathan. P. Reid j.p.reid@bristol.ac.uk

**Abstract.** Hygroscopic data for 36 organic compounds including amino acids, organic acids, alcohols and sugars is determined using a Comparative Kinetics Electrodynamic Balance (CK-EDB). The CK-EDB employs an electric field to trap charged aqueous droplets in a temperature and relative humidity (RH) controlled chamber. The dual micro dispenser set up allows for sequential trapping of probe and sample droplets for accurate determination of droplet water activities from 0.45 to > 0.99.
Here, we validate and benchmark the CK-EDB for the homologous series of straight chain dicarboxylic acids (oxalic – pimelic) with measurements in better agreement with UNIversal quasichemical Functional group Activity Coefficients (UNIFAC) predictions than the original data used to parametrise UNIFAC. Further, a series of increasingly complex organic compounds, with subtle changes to molecular structure and branching, are used to rigorously assess the accuracy of predictions by UNIFAC, which does not explicitly account for molecular structure. We show that the changes in hygroscopicity that result from
increased branching and chain length are poorly represented by UNIFAC, with UNIFAC under-predicting hygroscopicity. Similarly, amino acid hygroscopicity is under-predicted by UNIFAC predictions, a consequence of the original data used in the parametrisation of the molecular subgroups. New hygroscopicity data are also reported for a selection of alcohols and sugars.

## 1. Introduction

The hygroscopicity of an aerosol can be defined as the capacity of an aerosol particle to absorb water and quantifies the equilibrium partitioning of water between the gas and condensed phases (Krieger et al., 2012). Aerosol hygroscopic growth impacts directly on the radiative balance of the atmosphere, with the size and refractive index of aerosol particles influencing their light scattering and absorption cross-sections (Ravishankara et al., 2015;Moise et al., 2015). Similarly, the hygroscopic response of aerosol impacts on the transport of chemical components in the environment and on atmospheric chemical
composition through heterogeneous chemistry and cloud chemistry with implications for air quality (Akimoto, 2003;Farmer et al., 2015;Hallquist et al., 2009). The activation of cloud condensation nuclei (CCN) to form cloud droplets is governed by





hygroscopic response as well as aerosol size distribution, leading to the indirect effect of aerosols on climate (Farmer et al., 2015;Lohmann and Feichter, 2005). Furthermore, hygroscopic growth on inhalation can influence the depth of penetration of aerosol into the respiratory system, with consequences for the impact of ambient aerosol and particulate matter on rates of morbidity and mortality (Haddrell et al., 2015;Pöschl and Shiraiwa, 2015). Thus, an improved characterisation and

quantification of the hygroscopic response of ambient aerosol is important for more accurate predictions of the radiative forcing of aerosol, their impact on air quality and their consequences for human health.

Atmospheric aerosol are composed of a plethora of inorganic and organic species from a diverse range of biogenic and anthropogenic sources including inorganic salts, sulphates, nitrates, metals and organic compounds, such as acids, alcohols,

amino acids and sugars (Baltensperger, 2016;Seinfeld and Pankow, 2003;Zhang et al., 2015). Organic species can dominate ambient fine aerosol mass (particles < 1µm in diameter) and have varying degrees of oxidation, molecular mass, hygroscopicity and volatility (Jimenez et al., 2009;McNeill, 2015). Further, the composition of ambient aerosol is constantly changing due to heterogeneous reaction chemistry (Hallquist et al., 2009;Jimenez et al., 2009;Pandis et al., 1995;Ziemann and Atkinson, 2012), varying relative humidity (RH) and temperature (Farmer et al., 2015), and photochemistry (Jacob, 2000;George et al., 2015).

The equilibrium response of an aerosol to changes in ambient RH is described by the Köhler equation, which is a product of the solution water activity (the solute term) and a correction for surface curvature (the Kelvin term) (Wex et al., 2008). The solute term, representing the dependence of the equilibrium water activity on the composition of the solution of inorganic and organic compounds, can be determined from thermodynamic models that represent in detail the non-ideal interactions between

the ionic and neutral species within the solution. Based on parameterisations from experimental data, these models include the Aerosol Inorganic/Organic Mixtures Functional groups Activity Coefficients (AIOMFAC) (Zuend et al., 2008;Zuend et al., 2011), the Extended Aerosol Inorganic Model (Wexler and Clegg, 2002;Clegg et al., 1998), and UManSysProp (Topping et al., 2016) which allow calculation of the activity coefficients that characterise the non-ideality of the aqueous solutions. The key challenges in generating accurate predictions include capturing accurately the non-ideality of solutions, particularly under

very dry conditions/high solute concentrations (Dutcher et al., 2013;Nandy et al., 2016;Ohm et al., 2015), ion-neutral interactions in mixed inorganics/organics (Zuend et al., 2008;Zuend et al., 2011;Losey et al., 2016), the acidity and basicity of solutes (Rindelaub et al., 2016), liquid-liquid phase separation (Zuend and Seinfeld, 2012), solubility (Pajunoja et al., 2015) and the co-condensation of semi-volatile organic compounds (SVOC) with increase in water fraction (Topping et al., 2013). To treat the organic component, AIOMFAC, E-AIM and UManSysProp use the UNIversal quasichemical Functional group

Activity Coefficients (UNIFAC) method developed by Fredenslund et al. (Fredenslund et al., 1975). In this approach, molecules are divided into characteristic molecular subgroups and the activity coefficients derived from group contributions with no consideration for molecular structure. In UManSysProp, compounds are specified using the Simplified Molecular Input Line Entry System (SMILES) which are then converted to UNIFAC groups within the programme. Using these





approaches, Petters et al. have shown that the CCN activity of organic compounds can be modelled using group contribution methods (Petters et al., 2016).

Despite their accuracy, the use of group contribution methods to predict the water uptake for a larger number of organic
components in ambient aerosol is too computationally expensive for inclusion in regional chemical transport and climate models. Reduced parameter models are instead required to represent the thermodynamic properties of ambient organic aerosols. $\kappa$-Khöler theory characterises the solute component of hygroscopic growth by a single parameter $\kappa$ applicable in the limit of dilute aqueous solution aerosol (Kreidenweis et al., 2008;Petters and Kreidenweis, 2007). It must be assumed that the compounds are fully soluble and the aerosol does not undergo phase separation. The value of $\kappa$ spans from a value close to 0
for un-hygroscopic/hydrophobic components to a value around 1 for the most hygroscopic inorganic salts (Kreidenweis et al., 2008;Petters and Kreidenweis, 2007). Values are typically determined from sub-saturated hygroscopic growth measurements and reported at the highest accessible RH (Pajunoja et al., 2015). The value of $\kappa$ can also be inferred from measurements of the critical supersaturation required for CCN activation, a measurement in a super-saturated regime (Carrico et al., 2008). Despite the inherent approximations, reported values of $\kappa$ provide a way of linking the hygroscopicity of complex ambient
aerosol with empirical measurements of chemical functionality such as the level of oxidation, often reported as the ratio O:C from aerosol mass spectrometry measurements (Jimenez et al., 2009;R. Y.-W. Chang et al., 2010). Possible correlations of $\kappa$ with chemical composition (particularly O:C) have been extensively explored and reviewed (Rickards et al., 2013;Suda et al., 2014).

Although many ambient measurements of $\kappa$ have been made, there remains a necessity to rigorously address some of the challenges in quantifying aerosol hygroscopicity through controlled laboratory measurements on well-characterised aerosol of known composition. Dicarboxylic acids from $C_1 - C_7$ have been studied extensively in the literature and have been used as the basis for providing revisions of UNIFAC for typical organic components found in the atmosphere (Peng et al., 2001). Further, previous laboratory studies have examined correlations of $\kappa$ with composition, and have focussed on identifying the influence
of certain key functional groups on $\kappa$. For example, Suda *et al.* (2014) have studied the systematic impact on $\kappa$ of hydroxyl, carboxyl, peroxy, nitro and alkene groups of varying carbon chains lengths ($C_1 - C_{25}$) (Suda et al., 2014). However, there remain many gaps in hygroscopicity data for a number of compound classes, including; highly branched dicarboxylic acids, multifunctional compounds (including ring containing species), amino acids, organo-sulphates and nitro compounds.

We report here a systematic study of the hygroscopicity of a large number of organic compounds (listed in Table 1.) of varying functionality, solubility and molecular weight. This work benefits from the application of a novel electrodynamic balance (EDB) method that offers significant advantages over both alternative single particle techniques and ensemble experimental setups (Rovelli et al., 2016;Davies et al., 2013). Measurements can be made at high water activities (approaching very close to 1) with a very accurate comparative kinetics method for determining the gas phase RH. The timescale for the measurement



to record the whole growth curve is <10 s, sufficiently fast that the growth curves of organic species with vapour pressures of >1 Pa can be measured without significant volatilisation of the organic species. A temperature regulated chamber allows for stable and prolonged temperature control of the trapping region in the range -25°C – +50°C. The use of piezo-electric droplet-on-demand dispensers allows for the use of small sample volumes, allowing measurements on expensive (small amounts) of test compounds or the use of even filter collected samples. Measurements are made on droplets spanning the radius range from 4 – 30 μm, avoiding the additional complexity of correcting the hygroscopic growth measurement for the surface curvature term and providing an unambiguous measurement of the solute term (Rovelli et al., 2016).

More specifically, we will present hygroscopic data for 36 organic compounds from 4 distinct compound classes. A series of 17 dicarboxylic acids, with subtle differences to molecular branching and chain length, are used to examine the impact of structural isomerisation on water uptake. Measurements are also presented for a series of amino acids; despite their extensive release from biogenic sources, their hygroscopic properties have yet to be fully characterised (Chan et al., 2005). Although the UNIFAC model predicts water uptake of simple structures reasonably well, we will show that increasing molecular complexity and inclusion of nitrogen containing groups leads to considerably poorer prediction of hygroscopicity. Following an introduction to the methods and materials in Sect. 2, we will present the results for these different compound classes in Sect. 3.

## 2. Methods and Materials

Hygroscopicity studies are presented with measurements from a comparative kinetics technique applied in an EDB instrument, (referred to as the comparative kinetics EDB, CK-EDB, below) with electrodes in a concentric cylindrical arrangement. The full experimental details for the CK-EDB have been discussed extensively in previous publications and will only be briefly reviewed here (Rovelli et al., 2016;Davies et al., 2013), along with a discussion of the treatments used for parameterising solution density and refractive index (Cai et al., 2016).

### 2.1 The Comparative Kinetics Electrodynamic Balance

The CK-EDB can be used to probe the hygroscopic growth of aerosol particles from low to high water activities (<0.45 to >0.99) with a greater accuracy (<±0.2 % at water activities > 0.8 and ±1 % at water activities < 0.8) than can be achieved in conventional approaches (Rovelli et al., 2016). The CK-EDB employs an electric field to trap a charged dilute aqueous droplet starting at a water activity > 0.99. The droplet evaporates towards an equilibrium composition set by the RH of the surrounding gas flow; the RH is determined accurately from an independent measurement of the evaporation profile of a probe droplet of known hygroscopic response (either a pure water droplet or an aqueous sodium chloride solution droplet). The time-dependence in size and composition of the sample droplet (typically over a period of ~10 s) is then used to infer the hygroscopic





equilibrium growth curve over the full range in water activities experienced by the droplet during evaporation. The reader is referred to Rovelli et al. (2016) and Davies et al. (2013) for a full description of the method and the analysis.

A pulse voltage is consecutively applied to two droplet dispensers (MicroFab MJ-ABP-01, orifice size 30 μm) to sequentially
generate probe and sample droplets of known starting solute concentration. The droplets are charged by an induction electrode and are trapped within the electric field of the cylindrical electrodes within 100 ms of generation. Thermally regulated water channels through the electrodes and chamber body allow the temperature to be carefully controlled (-25°C − +50°C) by a refrigerated circulator (F32-ME, Julabo) using a mixture of polyethylene glycol and water. Humidified nitrogen gas flows vertically through the cylindrical electrodes and allows control of the gas phase RH of the chamber (total flow 200 mL min$^{-1}$
equivalent to a gas velocity of 3 cm s$^{-1}$). Evaporating droplets are illuminated with a 532 nm laser (Laser Quantum Ventus CW laser) and the elastic scattered light is collected using a CCD camera over a range of angles near a scattering angle of 45°. The droplet radius ($r$) is first estimated using the geometric optics approximation (error <±1 % for droplets > 10 μm) and the angular separation of fringes in the phase function ($\Delta\varphi$, radians) (Glantschnig and Chen, 1981),

$$r = \frac{\lambda}{\Delta\varphi}\left(\cos\frac{\varphi}{2} + \frac{n\sin\frac{\varphi}{2}}{\sqrt{1 + n^2 - 2n\cos\frac{\varphi}{2}}}\right)^{-1} \tag{1}$$

where $n$ is the refractive index of the droplet, $\lambda$ is the incident laser wavelength and $\varphi$ is the median observation angle. Initially, during data collection $n$ is assumed to be that of water. However, this assumption is corrected in subsequent data processing for the change in $n$ during evaporation; the compositional dependencies of density and $n$ are described below.

## 2.2. Molar Refraction: Refractive Index and Density Treatments

Solutes in aerosol droplets can reach supersaturated concentrations as water evaporates. Thus, to represent the solution density and refractive index, bulk measurements are insufficient and must be extrapolated to account for the full compositional range, i.e. the entire range in mass fraction of solute, MFS or $\phi_s$, from 0 to 1. We have recently provided a comprehensive assessment of the parameterisations that can be used to predict the density and refractive index of supersaturated organic solutions, and we summarise below the recommendations of this study relevant to their application in this work (Cai et al., 2016).

Aqueous solutions of an organic solute are prepared up to the solubility limit of the compound, and the density and refractive index are measured using a vibrating capillary density meter (Mettler Toledo Densito, accuracy ± 0.001 g.cm$^{-3}$) and a refractometer (Misco Palm Abbe, accuracy ±0.0001 at 589 nm), respectively. If the solubility of the organic solute allows measurements above an MFS of ~0.4, a third order polynomial in $\phi_s^{0.5}$ is fit to the bulk solution density values (where $\phi_s$ is
the MFS). If bulk measurements are limited by solubility to an upper limit in MFS <0.4, an ideal mixing treatment is applied to the bulk density values to allow the estimation of the density of the solute, $\rho_s$, constraining the bulk data to the equation:





$$\frac{1}{\rho_{em}(1-\phi_s)} = \frac{\phi_s}{(1-\phi_s)\rho_s} + \frac{1}{\rho_w} \tag{2}$$

where $\rho_{em}$ is the mass density of the mixture and $\rho_w$ the density of water. These two approaches assume that the density of the pure organic solute ($\rho_s$) is not known; under the conditions of aqueous solution aerosol measurements, the density of the solute

corresponds to that of the pure sub-cooled melt with most solutes instead existing in a crystalline form at room temperature. Further details of the density measurements and parameterisations for all systems studied are provided in the Supplementary Information Section.

Once the dependence of solution density on MFS is established, a fit of the bulk solution refractive indices is constrained to

follow the molar refraction mixing rule (Liu and Daum, 2008):

$$R_e = \left(\frac{n^2-1}{n^2+2}\right)\left(\frac{M_e}{\rho}\right) \tag{3}$$

where $n$ is the refractive index of the mixture, $M_e$ the effective molecular weight and $R_e$ is the molar refraction of the mixture. This allows the estimation of the molar refraction of the pure organic solute, again as a sub-cooled melt. In subsequent use, the molar refraction can be calculated for solutions of any composition,

$R_e = (1 - x_s)R_w + x_s R_s \tag{4}$

and the value of $n$ for the solution determined by solving for $n$ from equation (3). Pure component refractive indices, determined using the molar refraction mixing rule are presented in the Supplementary Information Sections alongside parametrisations of aqueous solution densities and sub-cooled pure component melt densities. Values of aqueous density and refractive index as a function of compound mass fraction are available in the supplementary information in Cai et al. (2016). Further in SI Fig S37.1

we consider the impact of uncertainties in density and refractive index treatments to the measured hygroscopicity, for all compounds shown, the error envelope on hygroscopicity is smaller than the size of the points.

**2.3. Extraction of hygroscopic properties**

During the evaporation of an aqueous droplet, the mass flux ($I$) of a component $i$ (in this, case water) can be estimated at each

recorded time step from the change in size and the associated density for the known composition of the droplet at that time, starting with a generated droplet of known solution composition. At each time step, the loss in mass is associated solely with loss of water, allowing a calculation of the new MFS and, thus, new values of $n$ and density. The new value of $n$ allows a refinement of the estimated radius, with full details (Davies et al., 2012). The mass flux can then be used to determine the gradient in water partial pressure in the gas phase using an analytical treatment (Kulmala et al., 1993), with

$I = -4\,Sh\,\pi\,r(RH - a_w)\left(\frac{RT_\infty}{M_i\beta_M D_i \rho_i^0(T_\infty)A} + \frac{a_w L_i^2 M_i}{R\beta_T K T_\infty^2}\right)^{-1} \tag{5}$





which accounts for the limiting influence of heat transport, due to latent heat lost, on the mass flux. In this equation, the gradient in water partial pressure is the difference between the RH and $a_w$, the instantaneous water activity at the droplet surface. The RH is determined from the probe droplet measurements, as described previously (Rovelli et al., 2016). In this study, the probe droplets are either pure water (for the RH range 80 – 99 %) or aqueous NaCl (for the RH range 50 – 80 %). All quantities in

this equation are known apart from $a_w$ and this can be estimated for every time-step by rearranging the equation to solve for $a_w$. $Sh$ is the Sherwood number, accounting for the enhancement in evaporation rate due to the moving gas flow over the droplet, and $r$ is droplet radius, measured experimentally. $R$ is the ideal gas constant, $T_\infty$ is the ambient temperature, $M$ is the molecular mass of water, $D_i$ is the binary diffusion coefficient of water in nitrogen and $\rho_i^0$ is the saturation vapour pressure. $A$ is a correction factor for Stefan flow, $K$ is thermal conductivity and $L_i$ is the latent heat of vaporization. Finally $\beta_M$ and $\beta_T$

represent the Fuchs-Suttugin correction factors for mass and heat flux, respectively.

The time-dependent data can also be used to estimate sub-saturated values of κ from (Petters and Kreidenweis, 2007)

$$GF = \left(1 + \kappa \frac{a_w}{1 - a_w}\right)^{\frac{1}{3}} \qquad\qquad (6)$$

where $GF$ represents the radius growth factor which is a ratio between the wet droplet radius and dry particle radius. The dry

size is estimated from the known starting size of the solution droplet and the starting concentration (MFS) of solutes. Time-dependencies in radius for a number of compounds with different κ values are shown in Fig. 1(a), illustrating how the CK-EDB experiment can discriminate between compounds of different κ during evaporation. For increasingly hygroscopic aerosol, there is a trend to a final equilibrated size that is larger and the temporal dependence of radius shows a shape that is characterised by less rapid loss of water. A caveat must be noted, however: the profiles do also depend on starting size, solute

concentration and the exact RH of the chamber, factors which are all explicitly accounted for in the full quantitative analysis. Values of κ for all compounds studied are reported at $a_w = 0.95$ in Table 1. It should be recognised that the apparent value of κ varies with the RH at which it is reported (Rickards et al., 2013).

During a typical experiment, measurements of sample and probe droplets are taken sequentially at several steady RHs, typically

50, 60, 70 and 80 % using an aqueous NaCl probe droplet and 80 and 90 % with a water probe droplet. Furthermore, at each measured RH, 10 sample and probe droplets are taken to ensure measurement reproducibility. Final hygroscopicity data is averaged (binned in small steps in RH) and presented as a function of MFS against water activity; full hygroscopicity curves are typically the result of measurements from between 30 – 80 droplets. It must be noted that kappa, κ, values are calculated using all data points before the binning process. In Fig. 1(b) we show typical time-dependencies in radius for a series of

aqueous-glycine droplets evaporating into four different RHs. The final hygroscopicity curve for glycine is shown in Fig. 1(c): the large orange points represent data which have been averaged (binned in $a_w$ steps) from 100's of data points measured from ~50 droplets.



## 3. Results and Discussion

Graphical and tabulated hygroscopicity curves for all 36 compounds studied, UNIFAC predictions, density parametrisations, refractive index values and compound purities are available as Supporting Information. Here, we summarise and compare the behaviour observed for the different classes of chemical compounds studied and consider the trends observed in the value of the parameter κ.

### 3.1. Hygroscopic Response of Dicarboxylic Acids of Varying Complexity

Structurally similar organic acids were chosen to examine the relationship between the hygroscopicity of binary component aerosol and the degree of carbon-chain branching, chain length and O:C ratio; some of the compounds chosen are identified in Fig. 2(a) and (b). All experimental hygroscopicity data are compared with thermodynamic predictions from the UNIFAC model using Peng corrections (referred to below as simply UNIFAC) to assess whether compound hygroscopicity is accurately represented. All calculations for dicarboxylic acids were performed using the AIOMFAC model.

As a benchmark test, we consider the homologous series of dicarboxylic acids, $HOOC(CH_2)_nCOOH$, from oxalic to pimelic acid (i.e. with n=0 to 5) in Fig. 2(a). The UNIFAC model predictions mostly agree closely with experimental observations at moderate to high water activity with some deviation at lower water activity although pimelic acid is an exception with experimental data deviating significantly from the model prediction. In Fig. 2(b) we compare data from a series of compounds with a malonic acid backbone, but with varying alkyl substituents (methyl, dimethyl and diethyl). The trend towards decreasing hygroscopicity with increasing hydrophobicity (increasing number and length of alkyl substituents) on a mass basis is clear, recognised from observing that there is less water associated with the solution at constant water activity as the molecular weight increases. In addition, the UNIFAC predictions become less accurate as the added substituent becomes larger, a consequence of representing all CH, $CH_2$, and $CH_3$ substituents by $CH_n$ (Zuend et al., 2008). The approach used here is particularly valuable for low solubility organic compounds as dilute solutions at high water activity provide the starting point for the measurement. For example, the dry particle size must be measured using a Hygroscopic Tandem Differential Mobility Analyser (HTDMA), necessarily setting a lower limit on the concentration of solutes use when atomising solutions to form aerosol. In addition, the short timescale of the measurement ensures that evaporation of the semi-volatile components, such as these dicarboxylic acids, is avoided.

We compare the measurements reported here with previous data (Peng et al., 2001) in Fig. 3(a)-(d) and Fig S38.1(a)-(d) for the straight chain dicarboxylic acids for which comparison can be made, oxalic, malonic, succinic and glutaric acid. The comparisons made in Fig. 3(a)-(d) act as a form of method validation, extending our previous work; bulk and EDB measurements (Peng et al., 2001) are presented alongside our CK-EDB data and UNIFAC predictions, with good agreement for all systems. Further to this, Fig S38.1(a)-(d) show the dependence on water activity of the difference in MFS (ΔMFS)





between the current experimental data or the previously published data (Peng et al., 2001) and UNIFAC predictions, allowing a quantitative comparison of the different experimental techniques. For these four straight chain dicarboxylic acids, the average deviations, ΔMFS, between UNIFAC predictions and our CK-EDB data ($a_w$ range 0.5 − 1) and the data of Peng et al. (2001) (up to ~0.9) are 0.017 ± 0.017 and -0.0037 ± 0.065. Note that although our data corresponds to a small systematic shift from

the UNIFAC model predictions, the spread of data about this mean offset is considerably less than in the previous study and extends to much higher water activity. The differences are summarised in Fig. 4, where the grey shaded area represents the standard deviation from UNIFAC for our measurements of the straight chain carboxylic acids (Fig 3.) and these data are plotted alongside all ΔMFS values for all 13 branched dicarboxylic acids studied. Nearly all branched acids deviate more from the UNIFAC predictions than is observed for the straight chain dicarboxylic acids. This confirms our previous observation that

the thermodynamic model predictions become increasingly unreliable as branching increases.

To further illustrate this trend in failure to capture the hygroscopicity reliably, we compare a sequence of dicarboxylic acids with carbon backbone length from 3-6 and with a methyl substituent attached in Fig. 5(a). All systems are poorly reproduced by the UNIFAC predictions with a value of ΔMFS that is larger than the limit set by straight chain dicarboxylic acids,

highlighting the lack of availability in branched chain experimental data to constrain the model. Interestingly, we compare the equilibrium hygroscopic response of a sequence of branched chain dicarboxylic acids in Fig. 5(b) with compounds selected to have the same O:C ratio of 0.57. It is striking that the equilibrium response curves are so similar for these compounds; this is captured by the similarity in their κ values of 0.065, 0.054, 0.066, 0.064 and 0.060 for diethylmalonic acid, 2,2-dimethyl glutaric acid, 3,3-dimethyl glutaric acid, 3-methyl adipic acid and pimelic acid, respectively. UNIFAC predictions are only

possible for two distinct formulaic units with the measurements indicating that these compounds have a higher degree of hygroscopicity than is captured by the model.

Hygroscopicity can also be represented as a function of the number of moles of water per mole of solute, shown in Fig. 6(a) for straight chain dicarboxylic acids. This is particularly informative for compounds with similar κ values (similar

hygroscopicity), shown in Fig. 6(b)-(c), because the differences in moles of solvating water molecules per mole of solute molecule, should be indicative of molecular structure. For the homologous series of straight chain dicarboxylic acids, a higher water activity is required to achieve the same molar balance of water and solute, Fig. 6(a). For the more hydrophobic branched dicarboxylic acids, an even larger water activity is required, although the curves are notably similar for these compounds which all have the same O:C ratio, Fig. 6(b). Figure 7 compares the experimental number of moles of water per number moles of

solute compared with that predicted by UNIFAC for the four compounds with the largest deviation in ΔMFS presented in Fig. 4. Rovelli et al. (2016) presented a similar comparison of experimental data and model predictions for inorganic salts, showing remarkable agreement between experimental values and model predictions with all points for all inorganic compounds falling within the ±0.002 uncertainty envelope in $a_w$, with this uncertainty envelope shown. However, there is a significant deviation from model predictions for the case of the branched dicarboxylic acids presented.



In summary, UNIFAC predictions agree well with measurements for simple unbranched dicarboxylic acids with the exception of pimelic acid, although there is an increasing degree of deviation with decreasing water activity. However, as the level of molecular complexity increases through the addition of single or multiple alkyl branches, there is increasing disparity between

UNIFAC predictions and measurements.

### 3.2. Hygroscopic Response of Amino Acids

A selection of amino acids were chosen for their biological relevance and to represent a wide range of structures and O:C ratios. Nitrogen containing compounds are prevalent in the atmosphere; amino acids contribute to this class of compounds due to their biological origin (Matsumoto and Uematsu, 2005;Barbaro et al., 2015). Recent studies have shown that nitrogen

containing compounds react to form brown carbon species, which absorb solar radiation in the UV and visible region. Absorption by brown carbon in cloud droplets leads to water evaporation and cloud dispersion counteracting the aerosol indirect effect (Laskin et al., 2015). Despite their importance as nitrogen containing compounds in the atmosphere, the hygroscopic properties of amino acids are yet to be fully characterised (Chan et al., 2005). Amino acids form zwitterions in solution, which supresses their vapour pressure and presents challenges in representing them with current thermodynamic

models with most models not allowing the inclusion of nitrogen containing groups (e.g. AIOMFAC). UNIFAC predictions cannot be performed for all amino acids examined here; in particular, the ring structures found in proline and histidine, cannot be represented as subgroups in the current version of UNIFAC. Thermodynamic model predictions for amino acids were generated using E-AIM.

The equilibrium hygroscopic responses for glycine, DL-alanine, L-valine and L-threonine are shown in Fig. 8(a). These four compounds all contain a similar glycine subunit, but include additional methyl, ethyl and hydroxyl groups. On a MFS scale, the hygroscopic response of these compounds is similar except for L-valine which is less hygroscopic, an observation that is expected given the additional hydrophobicity of the ethyl substituent. In a similar comparison to that considered in Fig. 5(b), compounds of the same O:C are compared in Fig. 8(b) with equilibrium relationships shown for L-lysine, L-histidine and L-

arginine. Lysine (κ, 0.219) is more hygroscopic than histidine (κ, 0.188) and arginine (κ, 0.147), illustrating that compounds with the same O:C can have very different hygroscopic responses, contrary to the observations for dicarboxylic acids. For improved predictions of the amino acids measured, the multilayer adsorption isotherm based model from Dutcher et al. (2013) that includes arbitrary number of adsorbed monolayers is used  in Fig. 8c) and d) to fit to the CK-EDB data. The model uses a power law relationship for aqueous solutions to determine adsorption energy parameter, C of water molecules with a solute by

adjusting a single parameter shown in Table S0.2. The model (equation 27 in Dutcher et al. 2013) is fitted to molality experimental data with respect to water activity for finding the parameter value, which results in a significant improvement in the MFS than UNIFAC. However, the notable difference in accuracy between the two models is not surprising: the isotherm based model of Dutcher et al. 2013 has an adjustable parameter (Table S0.2), while UNIFAC is a fully predictive model.





Figure 9(a) and (b) show comparisons between CK-EDB with available literature data for the hygroscopicity of both glycine and alanine. For glycine (Fig. 9 (a)) at high water activity there is good agreement between our CK-EDB data and literature data (Ninni and Meirelles, 2001;Kuramochi et al., 1997). Further, in Fig. 9 (b) CK-EDB data for alanine agrees with Kuramochi

*et al.* (1997). However, there is relatively poor agreement across the entire water activity range between CK-EDB data points from this study for both glycine and alanine with literature data (Chan et al., 2005). The discrepancy arises from the method used to identify the *'reference state'* to which all growth measurements are compared. For example, for certain systems Chan et al. (2005) have been required to extrapolate from bulk measurements to the highest RH of the droplet measurements. A similar approach is used by Peng et al. (2001) with dicarboxylic acid measurements presented in Fig. 3. However, the bulk

data points in this case have sufficient overlap between bulk and aerosol phase measurements to require very little or no extrapolation.

Furthermore, the general trends show that the amino acids are much more hygroscopic than is currently predicted using UNIFAC; indeed, when considering all 10 amino acids included in the SI, all are more hygroscopic than their model predictions

suggest (except asparagine). This is a consequence of the current reliance of the UNIFAC parameterisation on the data of Chan et al. (2005). Increased hygroscopicity compared with dicarboxylic acids with similar O:C ratios could be due to the zwitterionic nature of amino acids with their behaviour more similar to that of a salt than an organic species.

### 3.3. Sugars and Alcohols

When retrieving hygroscopic growth curves from the comparative kinetic measurements presented here, it is of critical

importance that there is no kinetic impairment to the evaporation of water. For many of the sugars we now consider, it is well established that the diffusion constant of water is strongly dependent on water activity, diminishing by many orders of magnitude and leading to slow diffusion limited release of water under dry conditions (Rickards et al., 2015). Thus, we present data that have been carefully assessed as independent of drying rate, as established by the RH of the gas phase the droplet is drying in. For many compounds, measurements unimpeded by kinetic limitations have not been possible below 80 % RH, and

consequently data presented below 80 % do not average to a consistent series of points.

Equilibrium hygroscopicity curves for three sugars galactose, sorbitol and xylose, and the long chain alcohol erythritol, are shown in Fig. 10. Molecular structures presented in Fig. 10 are the open chain form, which must be used during modelling using UNIFAC. These have been selected to illustrate the comparable degree of hygroscopic growth for these compounds, all

of which have the same O:C ratio of 1, even though they are subtly different in molecular structure and weight. Indeed, their experimental κ values are similar (galactose, 0.134; sorbitol 0.165; erythritol 0.255) and their hygroscopic properties are reasonably well represented by AIOMFAC.





### 3.4. Trends in κ with O:C Ratio and Molecular Structure

In order to efficiently represent the hygroscopic growth of aerosols in large scale models, it is crucially important that models of low complexity are used to represent aerosol of broad ranging source and chemical complexity. Correlations of the value of the parameter κ with surrogate measures of ambient aerosol composition such as O:C have been considered (Duplissy et al., 2011;Massoli et al., 2010). We consider the trends arising from the results presented here in the variation in κ with degree of substitution and functional group identity. In Fig. 11(a), we compare the values of κ for the homologous series of dicarboxylic acids and their branched derivatives. Clearly, both increased chain length and increased branching lead to greater hydrophobicity and lower hygroscopicity. Overall trends in hygroscopicity, as represented by the dependence of MFS on water activity, can be fit to the power law model from Dutcher et al. 2013 (Table S0.1) and we show the upper and lower bounds for compounds from each class (amino acids, organic acids, sugars and alcohols) in Fig. 11(b). This clearly illustrates that the amino acids are more hygroscopic than the majority of the other compounds studied.

Further, we consider in Fig. 12(a) the variation in κ with O:C ratio for all of the compounds examined here. The variation in κ with O:C ratio for the organic acids, sugars and alcohols is well-described (within the uncertainties) by the parametrisation provided by Rickards *et al.* (2013). However, the trend for the sequence of amino acid compounds shows that they are considerably more hygroscopic than comparable dicarboxylic acids with the same or similar O:C ratios. For example succinic acid and glycine have the same O:C ratio of 1 but with experimental κ values of 0.198, and 0.671 respectively. This illustrates the additional complexity in representing hygroscopicities with a simple single parameter model when multi-functional compounds are present, likely to be typical of the composition of atmospheric aerosol. Compounds with the same O:C ratio can have κ values that span from very low hygroscopicity (less than 0.05) to very hygroscopicity (approaching 0.4), as is seen for compounds with an O:C around 0.6. Fig. 12(b) shows the correlation between κ values determined in this study and calculated κ values from UManSysProp (Topping et al., 2016) using the hygroscopic growth factors [organic systems] model with density calculated using (Girolami, 1994). During the calculations the particle was assumed to have a dry diameter of 1000 nm and surface tension of 72 mNm$^{-1}$. Although there is a reasonably clear correlation between experimentally determined κ values and calculated κ, it is also clear that the value can be over-estimated by as much as a factor of 2.

### 4. Conclusions

In conclusion we have presented equilibrium hygroscopicity data and density and refractive index parametrisations for 36 organic compounds of varying functionality, molecular weight and O:C ratio. Of these compounds straight chain dicarboxylic acids ($C_2 - C_5$) were found to be in better agreement with UNIFAC than the initial data used to parametrise UNIFAC (Peng et al., 2001). Equilibrium hygroscopicity curves of increasingly branched dicarboxylic acids are not well predicted by UNIFAC. Additionally amino acid thermodynamic model predictions are not in agreement with experimental observations, this is due to earlier experimental measurements by Chan *et al.* (2005) which have been used to parametrise UNIFAC. The discernible





differences in hygroscopicity for different compound classes shown in both hygroscopicity curves in Fig. 13(b) and κ values in Fig. 12(a) offers the potential for future modelling methods to be built on relationships between compound classes and O:C and N:C ratios. Predictive tools considering these very general and smooth relationships would be much less computationally expensive than current group contribution methods and thus could be incorporated into climate models.

**Acknowledgements**

REHM and JPR acknowledge support from the Natural Environment Research Council through grant NE/N006801/1. AM acknowledges the EPSRC for support through DTA funding. GR acknowledges the Italian Ministry of Education for the award of a PhD studentship. CSD and LN acknowledge support from the National Science Foundation through Grant Number
1554936. Young Chul Song from the University of Bristol is acknowledged for CK-EDB measurement for the compound Erythritol, data used in this work.

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



**Tables.**

**Table 1. Experimentally determined κ values at $a_w$ = 0.95 for all compounds studied, presented alongside κ values calculated using UManSysProp and the smile string used for this calculation.**

| Compound | Average Experimental κ Value ($a_w$= 0.95) | Standard Deviation in κ (±) | UManSysProp Calculated κ ($a_w$= 0.95) | Smile String |
|---|---|---|---|---|
| *Amino Acids* | | | | |
| DL-Alanine | 0.357 | 0.010 | 0.402 | O=C(O)C(N)C |
| L-Asparagine | 0.187 | 0.017 | 0.337 | O=C(N)C[C@H](N)C(=O)O |
| L-Aspartic Acid | - | - | 0.332 | O=C(O)CC(N)C(=O)O |
| L-Arginine | 0.147 | 0.005 | 0.267 | NC(CCCNC(N)=N)C(O)=O |
| Glycine | 0.671 | 0.013 | 0.621 | C(C(=O)O)N |
| L-Histidine | 0.188 | 0.003 | 0.052 | O=C([C@H](CC1=CNC=N1)N)O |
| L-Lysine | 0.219 | 0.007 | 0.250 | C(CCN)CC(C(=O)O)N |
| L-Proline | 0.272 | 0.005 | 0.273 | OC(=O)C1CCCN1 |
| L-Threonine | 0.235 | 0.001 | 0.307 | C[C@H]([C@@H](C(=O)O)N)O |
| L-Valine | 0.253 | 0.003 | 0.136 | CC(C)[C@@H](C(=O)O)N |
| *Carboxylic Acids* | | | | |
| Oxalic Acid | 0.409 | 0.005 | 0.488 | C(=O)(C(=O)O)O |
| Malonic Acid | 0.281 | 0.003 | 0.362 | O=C(O)CC(=O)O |
| Succinic Acid | 0.198 | 0.011 | 0.252 | C(CC(=O)O)C(=O)O |
| Methyl Malonic acid | 0.234 | 0.006 | 0.252 | CC(C(=O)O)C(=O)O |
| Glutaric Acid | 0.144 | 0.005 | 0.139 | C(CC(=O)O)CC(=O)O |
| Methyl Succinic Acid | 0.160 | 0.003 | 0.138 | CC(CC(=O)O)C(=O)O |
| Dimethyl Malonic Acid | 0.149 | 0.002 | 0.150 | CC(C)(C(=O)O)C(=O)O |
| Adipic Acid | 0.101 | 0.004 | 0.055 | C(CCC(=O)O)CC(=O)O |
| 2, Methyl Glutaric Acid | 0.102 | 0.005 | 0.055 | CC(CCC(=O)O)C(=O)O |
| 3, Methyl Glutaric Acid | 0.103 | 0.006 | 0.055 | CC(CC(=O)O)CC(=O)O |
| 2,2 – Dimethyl Succinic Acid | 0.116 | 0.009 | 0.061 | CC(C)(CC(=O)O)C(=O)O |
| 2,3 – Dimethyl Succinic acid | 0.130 | 0.002 | 0.054 | CC(C(C)C(=O)O)C(=O)O |
| Pimelic Acid | 0.060 | 0.003 | 0.030 | OC(=O)CCCCC(=O)O |
| 2,2-Dimethyl Glutaric Acid | 0.054 | 0.002 | 0.032 | CC(C)(CCC(=O)O)C(=O)O |
| 3, Methyl Adipic Acid | 0.064 | 0.002 | 0.030 | CC(CCC(=O)O)CC(=O)O |





| | | | | |
|---|---|---|---|---|
| 3,3-Dimethyl Glutaric Acid | 0.066 | 0.003 | 0.032 | CC(C)(CC(=O)O)CC(=O)O |
| Diethyl Malonic Acid | 0.065 | 0.001 | 0.032 | CCC(CC)(C(=O)O)C(=O)O |
| Citric Acid | 0.189 | 0.002 | 0.192 | OC(=O)CC(O)(C(=O)O)CC(=O)O |
| Tartaric Acid | 0.27 | 0.006 | 0.308 | O=C(O)C(O)C(O)C(=O)O |
| Sorbitol | 0.165 | 0.003 | 0.303 | OC([C@H](O)[C@@H](O)[C@H](O)CO)CO |
| D-(+)-Trehalose Dihydrate | 0.088 | 0.001 | 0.151 | C([C@@H]1[C@H]([C@@H]([C@H]([C@H](O1)O[C@@H]2[C@@H]([C@H]([C@@H]([C@H](O2)CO))O)O)O)O)O)O |
| Galactose | 0.134 | 0.004 | 0.246 | O[C@H]1[C@@H](O)[CH](O[C@H](O)[C@@H]1O)O |
| Xylose | - | - | | |
| PEG4 | 0.154 | 0.004 | | |
| PEG3 | 0.151 | 0.003 | | |
| Erythritol | 0.255 | 0.006 | 0.380 | OC[C@@H](O)[C@@H](O)CO |

**Table 2. κ values available in the literature for dicarboxylic acids.**

| Compound | Literature κ |
|---|---|
| Oxalic acid | 0.504 ± 0.044(Rickards et al., 2013) |
| Malonic acid | 0.44 ± 0.16 (Koehler et al., 2006) 0.227 ± 0.028 (Kumar et al., 2003) 0.292 ± 0.011(Rickards et al., 2013) |
| Succinic acid | 0.231 ± 0.065 (Hori et al., 2003) 0.216 ± 0.20(Rickards et al., 2013) |
| Glutaric acid | 0.20 ± 0.08 (Koehler et al., 2006) 0.088 (Huff-Hartz et al., 2006) 0.168 ± 0.30 (Rickards et al., 2013) |
| Adipic acid | 0.096 0.102 ± 0.009 (Kumar et al., 2003) (Rickards et al., 2013) |





**Figures.**

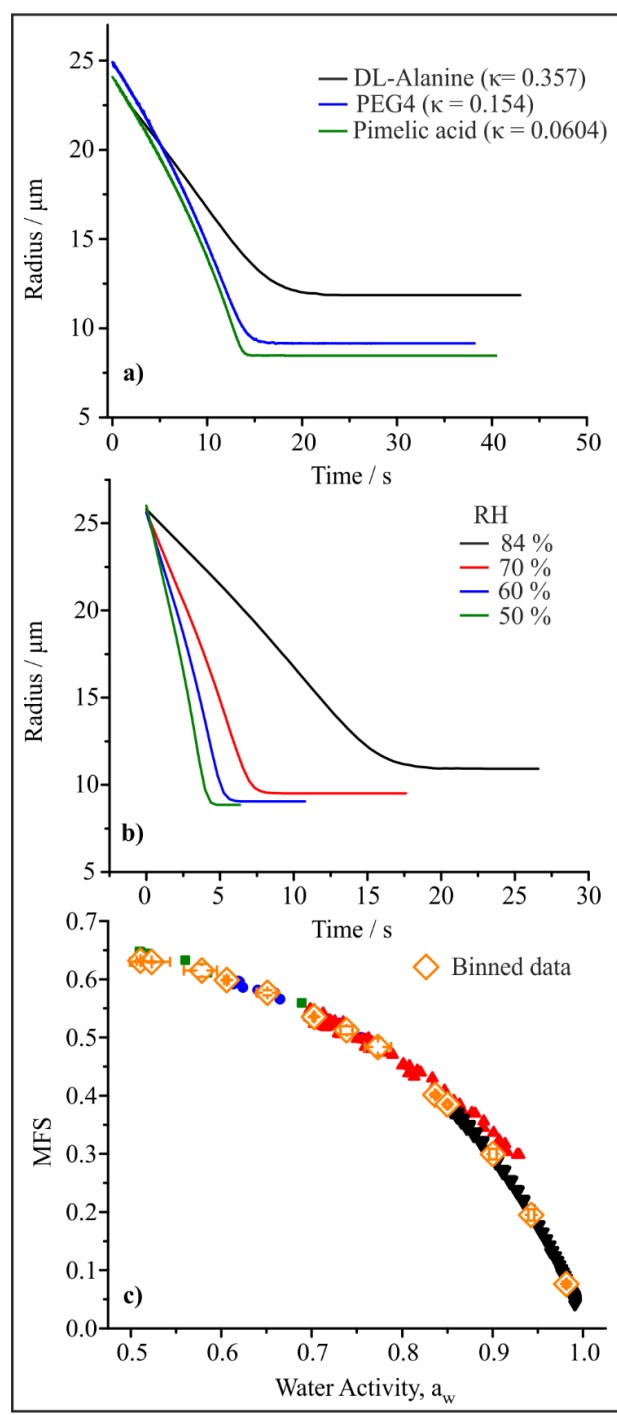

Fig. 1. **(a) Examples of the time-dependent evaporation of aqueous droplets containing compounds with varying κ value evaporating into similar RHs (~82 %). (b) The time-dependence of the radii of droplets of aqueous glycine evaporating into different RHs. (c) Equilibrium hygroscopicity curve for glycine, with binned data points (large open orange diamonds) estimate across the four different experiments at four RHs shown in (b).**





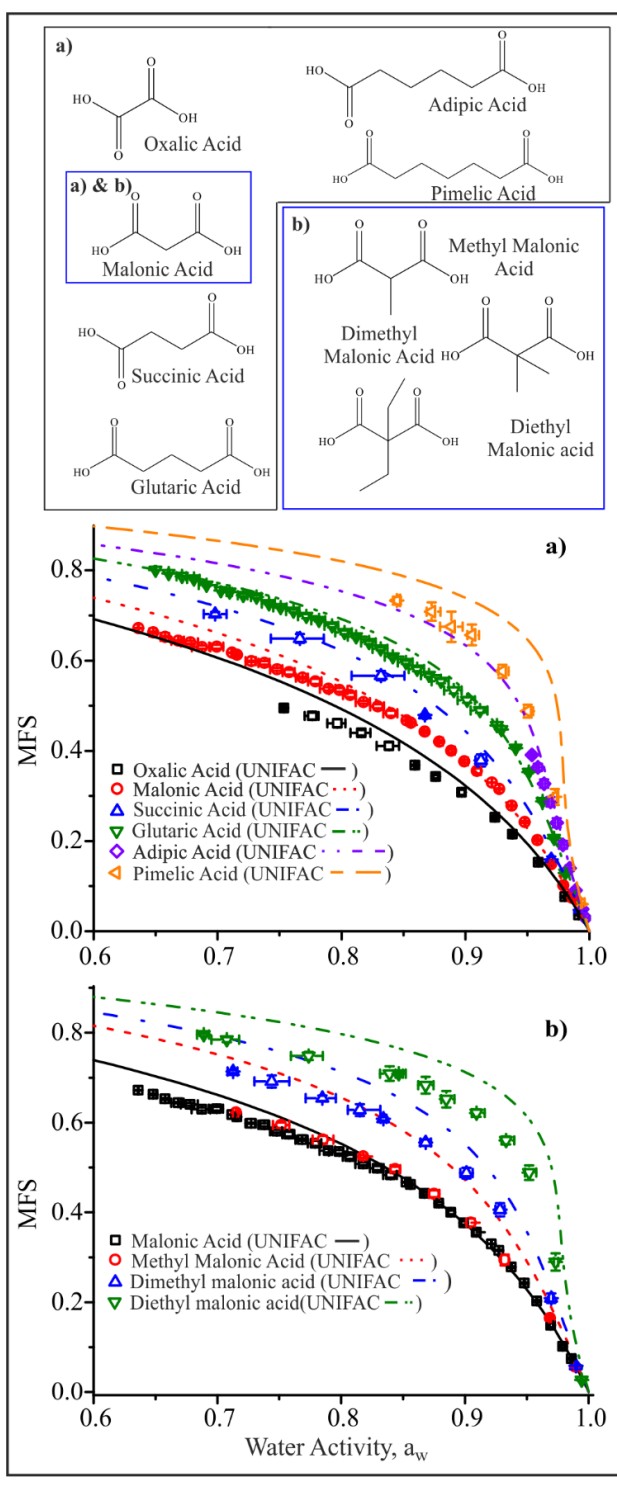

**Fig. 2. Equilibrium hygroscopic growth curves are shown in (a) for the homologous series of straight chain dicarboxylic acids and in (b) for dicarboxylic acids with a malonic acid backbone and increasing methyl substitution.**





**Fig. 3.** Hygroscopicity of dicarboxylic acid droplets measured with the CK-EDB (black open squares) compared with the EDB data of Peng *et al.* (2001) (blue up triangles) and bulk measurements (green down triangles) for (a) oxalic acid, (b) malonic acid, (c) succinic acid and (d) glutaric acid. UNIFAC predictions are shown for all compounds (solid black line).




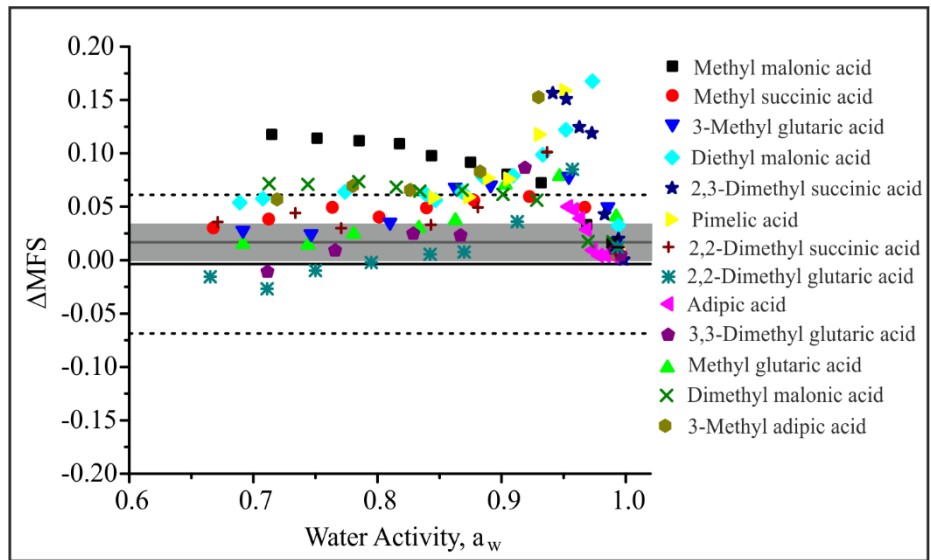

**Fig. 4. The difference between the mass fraction of solute from UNIFAC predictions and the CK-EDB data from this study (ΔMFS) for all 13 branched dicarboxylic acids studied. The average in ΔMFS for the 4 dicarboxylic acids in Fig. 3 (CK-EDB data, this study) across the whole water activity range is represented with a grey shaded area, the average represented by the dark grey line.**
5     **Additionally, the average ΔMFS (black solid line) and standard deviation (black dashed lines) derived from the Peng et al. (2001) data also shown in Fig. 3.**





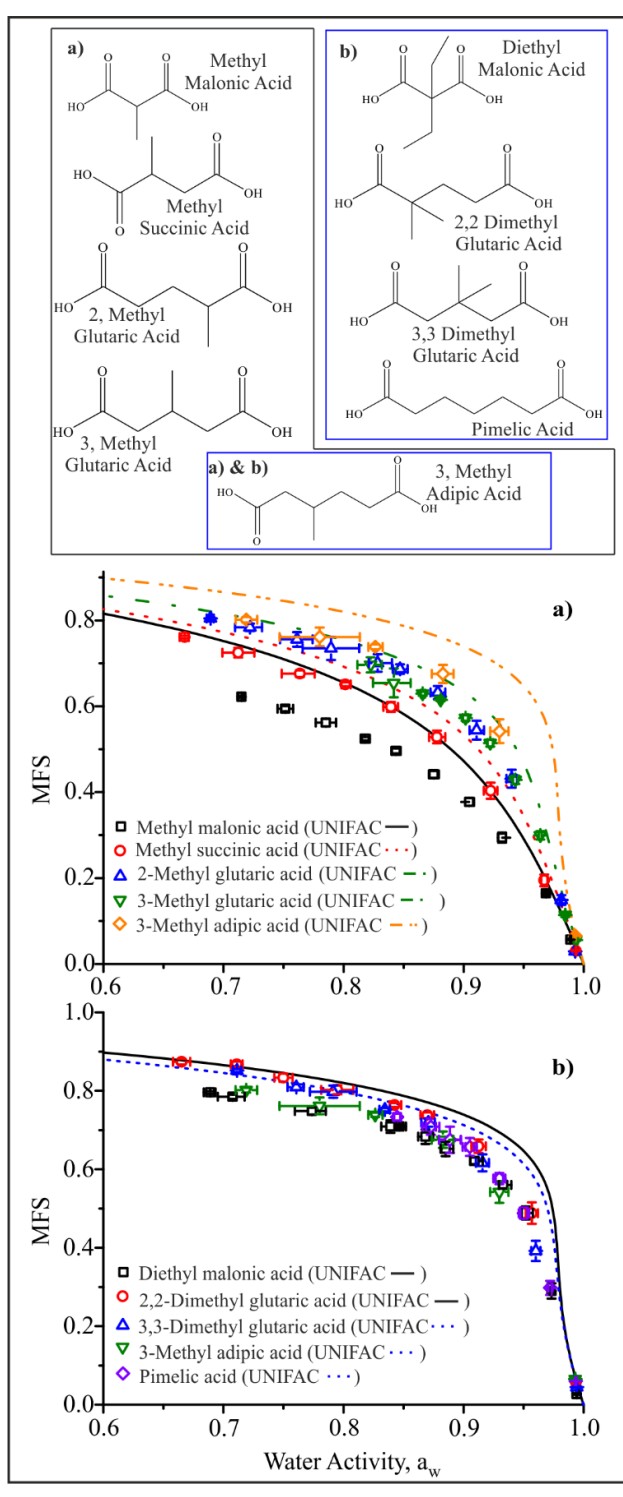

**Figure 5. Equilibrium hygroscopicity curves for a series of branched dicarboxylic acids are shown in a). In (b) CK-EDB hygroscopicity curves for a series of dicarboxylic acids with the same O:C ratio of 0.57.**




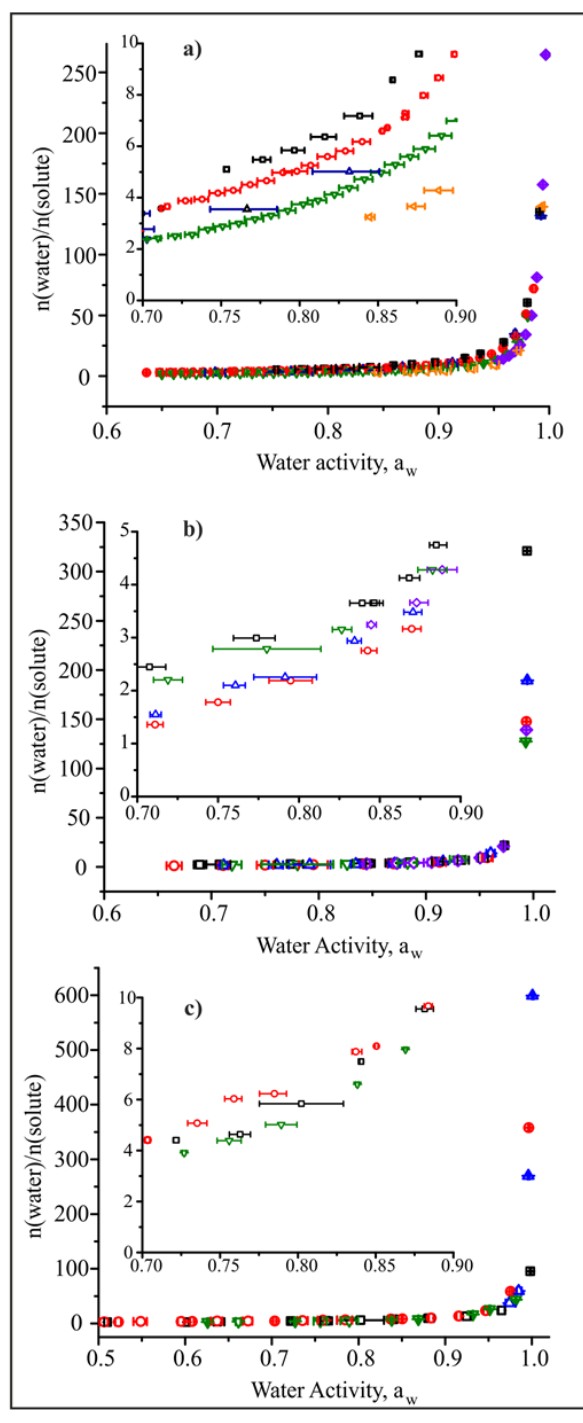

**Figure 6.** Moles of water, per mole of solute for (a) for straight chain dicarboxylic acids for oxalic (black squares), malonic acid (red circles), succinic acid (blue up triangles), glutaric acid (green down triangles), adipic acid (violet diamonds) and pimelic acid (orange left triangles). In (b) for diethylmalonic acid (black squares), 2,2-dimethyl glutaric acid (red circles), 3,3-dimethyl glutaric acid (blue triangles), 3-methyl adipic acid (pink down triangles) and pimelic acid (green diamonds). And in (c) for galactose (black squares), sorbitol (red circles), xylose (blue down triangles) and erythritol (green down triangles).





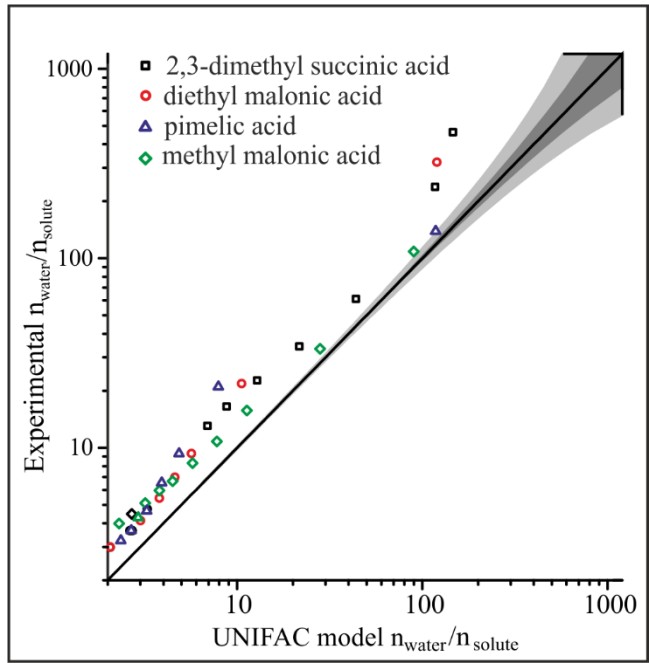

**Figure 7. Comparison of the experimentally determined number of moles of water per mole of solute and the value predicted from UNIFAC for the four dicarboxylic acids with the largest deviation from UNIFAC. Shaded regions correspond to error in aw of ±0.001 (dark shaded grey regions) and ±0.002 (light shaded grey regions).**







**Figure 8. Equilibrium hygroscopicity curves in (a) for structurally similar amino acids with different substituents alongside UNIFAC predictions. In (b) equilibrium hygroscopicity curves of amino acids with the same O:C ratio (0.33) with UNIFAC predictions. In c) and d) the same amino acids as a) and b) respectively and are presented alongside thermodynamic predictions using the isotherm model discussed in Dutcher et al. (2013) with coefficients available in Table S0.2.**





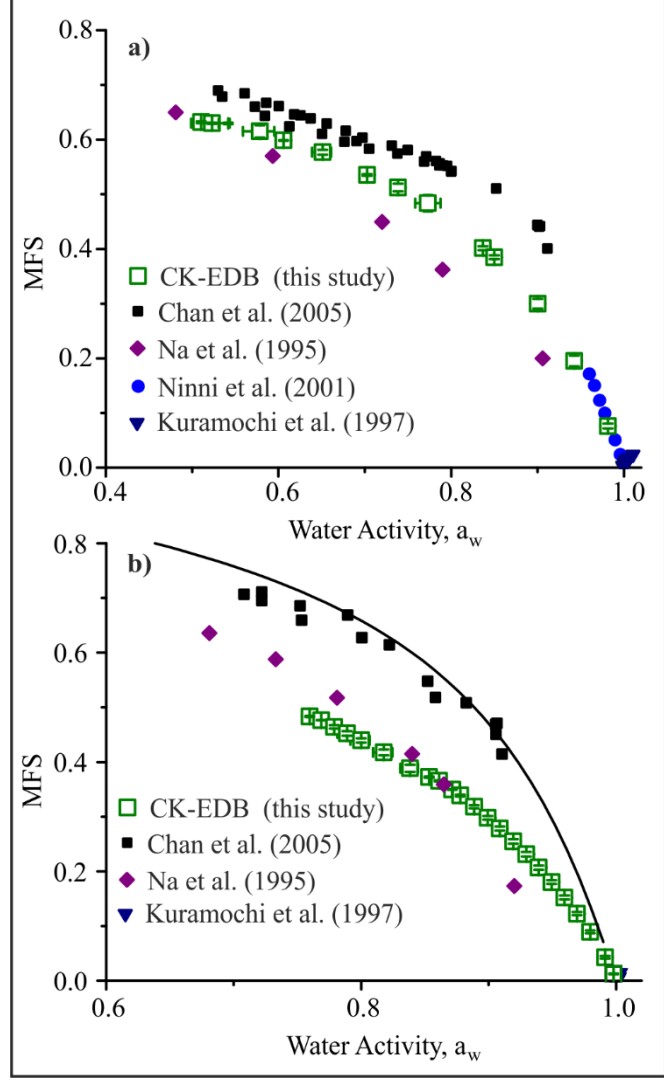

**Figure 9. Equilibrium hygroscopicity data for (a) glycine and (b) alanine. The UNIFAC model prediction for alanine is also shown (solid black line).**



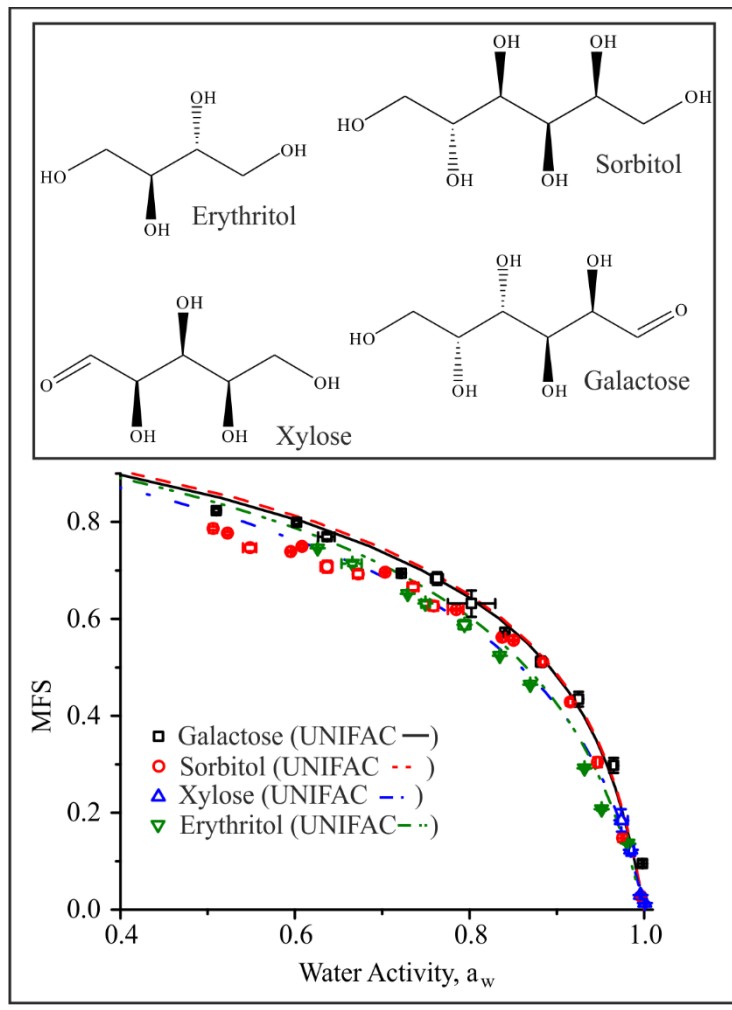

**Figure 10. Equilibrium hygroscopicity curves for sugars and alcohols with the same O:C ratio of 1.**



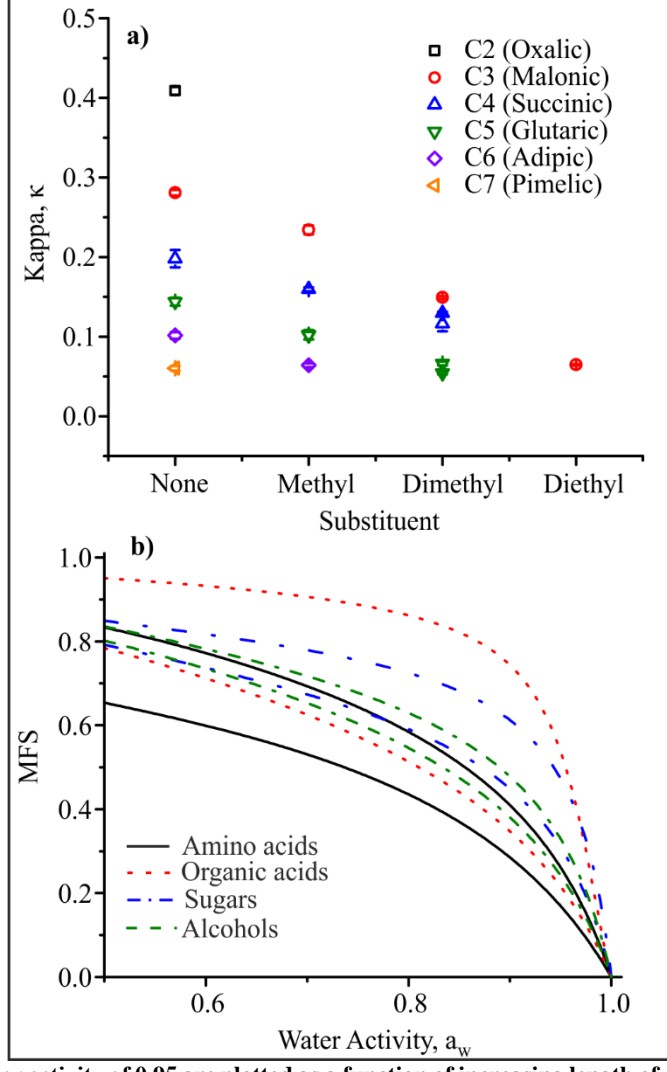

**Figure 11. In a) κ values at a water activity of 0.95 are plotted as a function of increasing length of substituent and carbon backbone. In (b) generalised equilibrium hygroscopicity curves are presented as a function of compound class. Upper and lower hygroscopicity limits for each compound class have been fitted using the isotherm model discussed in Dutcher et al. (2013) (coefficients available in Table S0.1).**





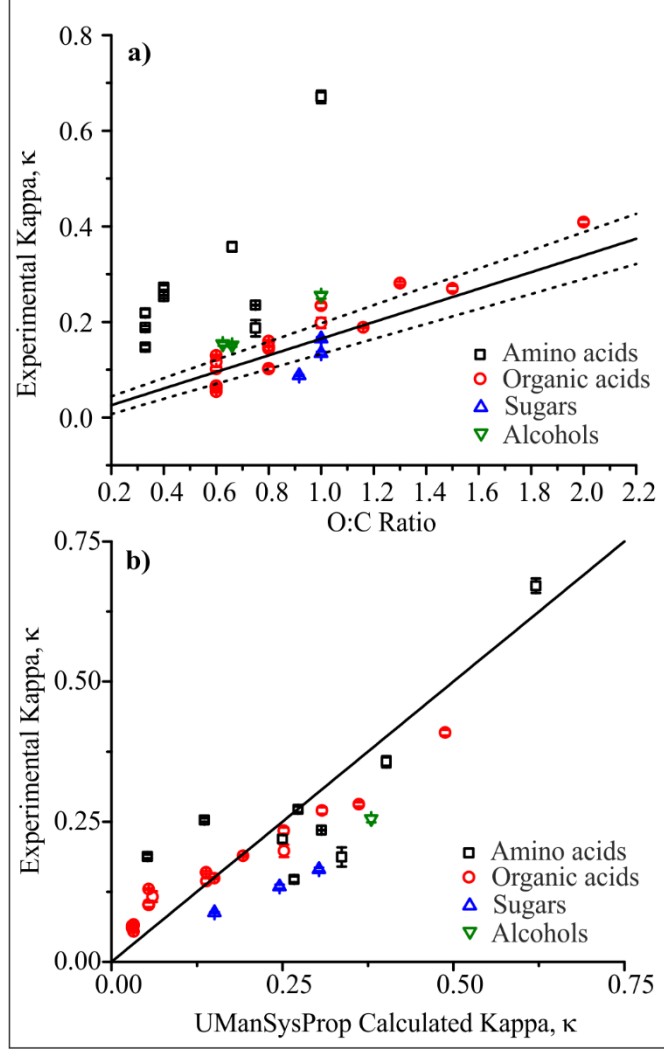

**Figure 12. All values of κ for all compound classes presented as a plot of (a) κ vs O:C ratio and (b) as a correlation plot between calculated κ and experimental κ. Errors are indicated but are smaller than some points. In (a) the black solid line overlaid of the form κ = (0.174 ± 0.017)×O:C – (0.009 ± 0.015), the parametrisation of Rickards *et al.* (2013),with the black dashed lines showing the upper and lower limits of this parametrisation. In (b) the line represents y=x.**

