# Peer review of "Influence of Organic Compound Functionality on Aerosol Hygroscopicity: Dicarboxylic Acids, Alkyl-Substituents, Sugars and Amino Acids"

_Atmospheric Chemistry and Physics, 2016_

## Referee Comment (RC1) · A. Zuend (Referee) · 18 Jan 2017

ACPD article:

**Influence of Organic Compound Functionality on Aerosol Hygroscopicity: Dicarboxylic Acids, Alkyl-Substituents, Sugars and Amino Acids**

*Article by Aleksandra Marsh et al.*

Review by Andreas Zuend

Marsh et al. present new measurements and discussion of the hygroscopicity of 36 different organic compounds using a custom-built "comparative kinetics" electrodynamic balance (CK-EDB). This instrument was designed to allow for hygroscopicity measurements of super-micron aqueous solution droplets over a wide range in relative humidity (RH), including the range above 90 % RH, which was inaccessible with reasonable precision by past EDB designs. These new data sets are certainly of value as the CK-EDB enabled measurements into the supersaturated RH (or water activity) range inaccessible by most bulk techniques, yet important for the measurement of the liquid state hygroscopicity of organic compounds with low solubility in water. Furthermore, the authors point out that a relatively short time for a set of measurements on a sample droplet enables measurements involving semi-volatile organic compounds without significant organic evaporation effects affecting the particle composition.

Processed experimental data, including error estimates, are tabulated in a practical format in the Supplementary Information (SI) to the article, which is commendable. This promises to be useful for independent comparison with other measurements and for future improvements of thermodynamic model parameterisations, such as the UNIFAC. The authors discuss the agreement/disagreement of UNIFAC-predicted water activity vs. mass fraction of organic in comparison to the new data sets and available experimental data for a subset of the systems. In addition, the authors parameterised an isotherm-based model by Dutcher et al. with a subset of the new data, providing inter- and extrapolation of the binary systems to concentrations not considered experimentally.

I regard this work as a valuable research contribution well within the scope of the ACP journal. Overall, the article is concise and well structured. I am generally supportive of this article and the wealth of new experimental data reported. However, there are several points that should be clarified in a revision of the manuscript before publication is recommended. In particular, the comparison of the measurements to the group-contribution model UNIFAC requires further clarifications about model versions/parametrisations and the description of UNIFAC's limited ability of accounting for slight differences in molecular structure, e.g. branching at constant molar mass and/or O:C ratio, need to be improved to avoid confusion and incorrect explanations. Addressing the issues raised should be relatively straightforward, see the general and specific comments below.

**General comments**

- Section 2. Methods and Materials. While the reader is referred to Rovelli et al. (2016) and Davis et al. (2013) for a detailed description of the CK-EDB method, a general description of the chemicals used, their purities and solution preparation is missing. Some of that information is provided in the SI

only. I suggest that a brief description is also given in the main text and that the reader should be informed about additional information on this in the SI.

- Temperature range and droplet temperatures. In the first paragraph of page 4 it is highlighted that the temperature in the EDB trapping region can be controlled well over a ~75 K range, however, throughout the main text information about the actual temperature used is missing (including tables and figures). As far as I can tell from the temperature information given in the SI, all experiments and model calculations were carried out at 293.15 K. Were hygroscopicity measurements at other temperatures considered (which would be useful, e.g. for improved, temperature-dependent thermodynamic model parameterisations given the temperatures were sufficiently different)? Hence, a discussion on the EDB temperature range used and the actual droplet surface temperature during the evaporation experiments will require some discussion. From Rovelli et al. (2016) it seems clear that the time scale of the evaporation will lead to deviations between droplet and surrounding gas phase temperature. Moreover, given that the evaporation rate from a relatively concentrated solution droplet is different from the evaporation rate of the probe droplet, a discussion of such temperature related issues with respect to the retrieval of the sample droplet's water activity at a particular temperature seems appropriate.

- In contrast to the inorganic solutes used in Rovelli et al. (2016), the present study involves organic solutes, some of which may cause a significant increase in mixture viscosity with decreasing droplet water content during evaporation in the EDB. In this context, the time scale of 10 s for the evaporation from the droplets may become an issue for droplets > 10 μm radius, potentially impeding the droplet-gas mass transfer (e.g. Koop et al., 2011) and potentially violating assumptions about a homogeneous, concentration-gradient-free mixing of water and organic compound within sample droplets aside from a developing temperature gradient within a rapidly evaporating droplet. The authors discuss the viscosity concern in Section 3.3, where it is mentioned that for many compounds measurements unimpeded by kinetic limitations were not possible below 80 % RH. Because this consideration may not only apply to sugars and alcohols, but to many of the multifunctional organics of higher molar mass, a more general discussion of kinetic limitations and consequences for the CK-EDB data processing should be provided in Section 2 where the method is described. If a relatively viscous binary aqueous droplet is exposed to low RH and evaporates water quickly, there may be insufficient time for homogeneous mixing in the droplet bulk compared to the near-surface volume of the droplet, which could lead to a concentration gradient and a higher solute concentration in the surface region of the droplet, affecting the local water activity there. Under such conditions, an organic solute may appear as more hygroscopic than it would be under actual gas-particle equilibrium conditions. Did the authors consider such effects in their method and the data processing? It is also not clear whether the authors considered a longer measurement time scale with slower evaporation settings for systems where substantial kinetic limitations may occur (and for which organic evaporation may not be a concern). Please discuss.

- UNIFAC models – three general comments and clarifications:

   1. The authors compare many of the measurements to predictions by "the" UNIFAC model, however, the information about the specific model version used and its parameterisation for some of the compounds is incomplete in the manuscript. While the original UNIFAC model theory by Fredenslund et al. (1975) is mentioned on page 2, several UNIFAC modifications

(changes to model equations, e.g. UNIFAC-Dortmund, UNIFAC-Lyngby, etc.) and several revisions of UNIFAC parameter tables applicable to certain UNIFAC versions have been published in the past 40 years. For example, the AIOMFAC model (Zuend et al., 2008; 2011), which includes a UNIFAC model based on the original theory of Fredenslund et al. (1975), relies mostly on the revised parameter set by Hansen et al. (1991). However, Zuend et al. (2011) discuss several modifications of the parameter database, including the use of improved interaction parameters determined by Marcolli and Peter (2005) for alcohols and multifunctional compounds containing hydroxyl groups, as well as modified interaction parameters by Peng et al. (2001) for a subset of interactions involving carboxylic acid groups. These modifications are detailed in Zuend et al. (2011) and are used in the online version of the AIOMFAC model (which was used for several comparisons with measurements in the present paper; see also [www.aiomfac.caltech.edu/about.html](www.aiomfac.caltech.edu/about.html)). Similarly, the online UNIFAC versions in UManSysProp ([http://umansysprop.seaes.manchester.ac.uk](http://umansysprop.seaes.manchester.ac.uk); Topping et al., 2016), which includes AIOMFAC and a UNIFAC version, and the E-AIM website's UNIFAC ([www.aim.env.uea.ac.uk/aim/aim.php](www.aim.env.uea.ac.uk/aim/aim.php)) contain modified parameter sets from Peng et al. (2001) and from other sources of UNIFAC parameter revisions (see also [http://www.aim.env.uea.ac.uk/aim/phpmain/edit_help.php#section100](http://www.aim.env.uea.ac.uk/aim/phpmain/edit_help.php#section100) for details on UNIFAC in E-AIM). While some of these newer parameterisations lead to only slight changes to predicted water activities compared to the original UNIFAC by Fredenslund et al. (1975) with the Hansen et al. (1991) parameters, others are significant – and e.g. in the case of AIOMFAC, the description of alcohols and sugars is substantially modified by the introduction of specific subgroups and main groups in the model for these compounds (of relevance for the comparisons with CK-EDB data made in this study). Therefore, to provide sufficient detail for clarity and reproducibility, it is necessary to specify which models and parameterisations were actually applied (e.g. in the Methods section).

2. Contrary to the statements on page 2, lines 30 – 32 and on page 8, lines 20-21, UNIFAC (and AIOMFAC) actually account for the molecular structure and for certain differences between branched and straight-chain dicarboxylic molecules of the same molar mass – albeit in a limited way. For example, via the differing number in hydrogen atoms on $CH_2$, CH, and C subgroups, which leads to different values of the relative Van der Waals volume and surface area terms in the combinatorial part of the UNIFAC model for these alkyl subgroups (affecting predicted activity coefficients). For this reason, the UNIFAC subgroup assignments, as listed in Table S0 of the SI, are incomplete/incorrect in the case of the dicarboxylic acids. For example, $CH_n$ is not a UNIFAC/AIOMFAC *sub*group and as such does not sufficiently characterise the compound; instead the appropriate subgroups need to be stated. For example, correct subgroup assignments show that the three distinct $C_7$-dicarboxylic acids (see also Table 3 of Zuend et al. (2011): 3-methyl adipic acid, $(CH_3)(CH)(CH_2)_3(COOH)_2$, 3,3-dimethylglutaric acid, $(CH_3)_2(C)(CH_2)_2(COOH)_2$, and pimelic acid, $(CH_2)_5(COOH)_2$, have slightly different subgroup formulas in UNIFAC/AIOMFAC and consequently there should be distinct model curves in Fig. 5b and UNIFAC structure formulas in Table S0 of the SI. Although, this reviewer agrees that the differences between UNIFAC predictions for such similar dicarboxylic acids are likely small.

3. The UNIFAC group-contribution method also offers another way to account for proximity effects by neighboring subgroups in organic molecules: specific subgroups can be assigned to larger sections of a molecular structure and that has been proposed for modified UNIFAC parameterisations in the case of amino acids. For example, Gupta and Heidemann (1990) introduced a specific "proline" UNIFAC subgroup (including a subset of determined interaction parameters for aqueous solutions of amino acids). Kuramochi et al. (1997) introduced a series of new functional groups and determined UNIFAC parameters for the description of most amino acids, including histidine, for a modified UNIFAC version based on "Larsen's UNIFAC". Thus, statements like (page 10, line 15): "*UNIFAC predictions cannot be performed for all amino acids examined here; in particular, the ring structures found in proline and histidine, cannot be represented as subgroups in the current version of UNIFAC.*" are not generally correct – the UNIFAC parameterisation by Kuramochi et al. covers most amino acids studied experimentally in this work. However, it is correct that those "specialized" UNIFAC modifications are not implemented in the online versions of AIOMFAC and UNIFAC in E-AIM (see point (1) above), so they are not conveniently available for calculations, which is likely what is meant by the authors' statement. Such general statements should therefore be revised accordingly and the work by Gupta and Heidemann (1990), Kuramochi et al. (1997) and others mentioned. Consider also that parameter sets that were determined for different UNIFAC model versions are typically not compatible and the use of specific subgroups with only a limited set of interaction parameters determined, e.g. for aqueous mixtures of amino acid solutions only, disqualifies the applicability of such models for predictions of complex, multi-component and multi-functional mixtures of interest in atmospheric aerosol chemistry (as discussed in Section 5.4 of Zuend et al., 2011).

**Specific comments and technical corrections**

- Abstract, first sentence and page 4, line 9: "Hygroscopic data" should be "Hygroscopicity data" (the data itself is likely not hygroscopic).

- P3, line 7: correct spelling of "Köhler"

- P3, l. 11 – 13: "Values are typically determined from sub-saturated hygroscopic growth measurements and reported at the highest accessible RH (Pajunoja et al., 2015). The value of $\kappa$ can also be inferred from measurements of the critical supersaturation required for CCN activation, a measurement in a super-saturated regime (Carrico et al., 2008)." It would be appropriate to state that $\kappa$ values determined at different RH and, to a lesser extent temperature, can vary substantially, especially when comparing $\kappa$ determined from CCN activation data at water super-saturation compared to sub-saturation conditions, as, e.g., discussed by Hodas et al. (2016) and references mentioned therein.

- P3, l. 33: correct "(approaching [values] very close to 1)"

- P4, l. 25: clarify the accuracy statement: "with a greater accuracy ($< \pm 0.2$ % at water activities $> 0.8\ldots$" do you mean $< \pm 0.2$ % error in water activity or in hygroscopic growth factor or MFS?

- P6, l. 5: "with most solutes instead", better: "with most pure organic compounds instead" since this is not about a solution but about the pure components.

- P6, title 2.3: Replace "hygroscopic" by hygroscopicity

- P7, l. 2: "In this equation, the gradient in water partial pressure is the difference between the RH and $a_w$, the instantaneous water activity at the droplet surface." First, given the evaporation setup with an RH profile dependent on the distance from the droplet, it needs to be stated which RH (and measured where) is meant, i.e. is it the RH at the droplet surface or the RH (sufficiently) far away from the droplet. Second, the difference (RH - $a_w$) or rather saturation ratio $S - a_w$ (as in Rovelli et al., 2016) alone does not constitute a "gradient". Also, since the component subscript "i" in Eq. (5) denotes water (i.e. subscript "w" as in $a_w$), it would seem better to use "w" instead of "i".

- P7, l. 9: "is the latent heat of vaporization"; add "of water" at temperature $T_\infty$(?).

- P8, l. 10: "using Peng corrections" the meaning of this is unclear. Also, as detailed above, the UNIFAC models likely used by the authors actually include further modifications in terms of the used parameter sets and/or subgroup assignments.

- P8, l. 20: "In addition, the UNIFAC predictions become less accurate as the added substituent becomes larger, a consequence of representing all CH, $CH_2$, and $CH_3$ substituents by $CH_n$ (Zuend et al., 2008)." There seems to be a misunderstanding about the UNIFAC (AIOMFAC) way of group-contribution calculations, see the general comment above. Only group-group interactions in the residual UNIFAC expressions are common for all $CH_n$ subgroups (with n = 0,1,2,3), but the volume and surface area terms (combinatorial part) are not. This is the case in all variants of UNIFAC.

- P9, l. 20 and l. 17: There are actually more than two distinct UNIFAC group formulas for the different $C_7$-dicarboxylic acids, see the general comment above. Also, given that the UNIFAC (AIOMFAC) model predictions of water activity show a deviation from the CK-EDB data for the straight-chain pimelic acid, the model-measurement deviations shown in Fig. 5b are expected and at least consistent in that sense. Related to the statement on line 17, the observed similarity in hygroscopicity of the different $C_7$-dicarboxylic acids suggests that the degree of branching and/or lengths of alkyl substituents may not always play a substantial role, in particular above a water activity of 0.8. This seems to be a counter-example to the trends observed for the smaller dicarboxylic acids with alkyl substitutions (and a hint for a general underestimation of the hygroscopicity-contribution by the $CH_n$ groups as represented in UNIFAC/AIOMFAC).

- P10, l. 15: The sentence should be revised as certain nitrogen containing compounds are available in most UNIFAC models (including in AIOMFAC for organics + water systems) since the parameter set by Hansen et al. (1991) includes amine, amid, nitro, nitrile and pyridine groups and some version include organonitrate groups (Compernolle et al., 2009; Zuend and Seinfeld, 2012) and proline and histidine groups Kuramochi et al. (1997).

- P10, l. 18: clarify which model for activity coefficients was used in E-AIM for the amino acids.

- P10, l. 22: "except for L-valine"; According to Fig. 8, L-Threonine behaves similar to L-valine even though it contains a hydroxyl group instead of a methyl group. So it seams that L-valine is not an exception or not the only one. Also, the UNIFAC prediction for glycine is missing in Fig. 8a.

- P10, l. 30: "is fitted to molality experimental data"; molality of what? The last part of that sentence needs to be rephrased as well.

- P11, l. 13: Statement needs to be revised given the above clarification about specific UNIFAC parameterisations for aqueous solutions of amino acids.

- P11, l. 15: "This is a consequence of the current reliance of the UNIFAC parameterisation on the data of Chan et al. (2005)." This statement is incorrect, because the UNIFAC models used by the authors do in fact not contain the modified parameters by Chan et al. (2005); rather, they are based on Hansen et al. (1991) and Peng et al. (2001) parameters for the amino acids. Also, as is clearly shown in Chan et al. (2005), their modified UNIFAC parameterisation yields similar results to the Peng et al. version in many cases and the Peng et al. parameterisation is in reasonable agreement with their own experimental data (e.g. for threonine). Therefore, the discrepancies between the new CK-EDB data and the UNIFAC model curves shown indicate clear discrepancies among different experimental data sets, as is discussed by the authors in the first paragraph of page 11.

- P11, l. 27: "Molecular structures presented in Fig. 10 are the open chain form, which must be used during modelling using UNIFAC."; Why "must"? AIOMFAC also allows you to use the cyclic structure of sugars in aqueous solution, e.g. glucopyranose instead of glucose, if desired.

- P12, l. 10: and Fig. 11 & 12: replace the compound class labelled "organic acids" by a more appropriate label, e.g. "dicarboxylic acids", since amino acids are also organic acids but not part of that class.

- P12, l. 32: Statement is incorrect, see comment to P11, l. 15.

- P13, first paragraph. With respect to the applicability of the determined component-kappa values from binary data with a simple mixing rule for a complex mixture's total hygroscopicity parameter kappa, I suggest the authors consider in this section that it remains rather uncertain whether the kappa values determined based on binary water + amino acid data apply in multicomponent mixtures of relevance for atmospheric aerosol. This is because the substantial hygroscopicity exhibited by many of the amino acids, due to their zwitterionic nature in aqueous solution, may be affected substantially by the presence of inorganic acids and dissolved salts in aerosol mixtures, altering the partial water uptake contribution by the amino acids in a non-linear manner. This may motivate further experimental investigations for organic-inorganic mixtures with the CK-EDB and other setups.

- Table 1: State the temperature (range) for the measurements. Also the caption text and table header concerning SMILES needs revision.

- Fig. 1: Lower panel, at around 0.9 water activity, the red triangles-up and black triangles-down symbols suggest a larger scatter in experimental data than the binned data and error bars account for. It is unclear why if it is assumed that the different drying rates have similar measurement uncertainty? A brief discussion may be useful.

- Fig. 2: state the UNIFAC parameterisation used, if AIOMFAC-web was used, then stating that would be sufficiently specific.

- Fig. 6: the y-axis label "n(water)/n(solute)" would be better written as in Fig. 7 or perhaps in abbreviated form, such as $n_w/n_s$.

- Figs. 8 and 9: The UNIFAC (Peng et al. parameterisation) model curve for Glycine is missing.

- Fig. 9: Comparing this figure to Fig. 1 of Chan et al. (2005), it is clear that many experimental data points from Na et al. are missing, as well as bulk data by Kuramochi et al. (1997) to higher MFS/lower $a_w$ than shown and data by Ninni and Mereilles (2001) in Fig. 9b. Including all these measurements in Fig. 9 will provide a better comparison for the discussion concerning the substantial discrepancies found among the experimental data sets and in comparison to model predictions.

- Supplementary Information: It would be useful to briefly state at the end of the main text what information is provided in the SI.

- SI, Table S0: the page numbers for different systems are listed, but the pages in the SI were not numbered. Also, the AIOMFAC subgroups stated for the dicarboxylic acids with $CH_n$ groups should be revised, see general comment. The "$CH_n(OH)$" groups stated for citric acid, tartaric acid and other compounds should be stated with OH preferentially in superscript (e.g. $CH_2^{[OH]}$ for a $CH_2$ subgroup bonded to an OH group, which is specified separately) to avoid confusion about the number of OH groups present in the molecular structure (see also Table 3 of Zuend et al., 2011).

- SI, Fig. S8.1: Check the caption text and symbols in the figure. I do not see any coloured curves for data at different temperatures stated in the caption.

- SI, S26 and S27: For aqueous PEG mixtures, much improved interaction parameters have been determined for a PEG-specific version of AIOMFAC, but these are not yet included in AIOMFAC-web (see also Hodas et al., 2016).

**References**

Compernolle, S., Ceulemans, K., and Müller, J.-F.: Influence of non-ideality on condensation to aerosol, Atmos. Chem. Phys., 9, 1325-1337, doi:10.5194/acp-9-1325-2009, 2009.

Davies, J. F., Haddrell, A. E., Rickards, A. M. J., and Reid, J. P.: Simultaneous Analysis of the Equilibrium Hygroscopicity and Water Transport Kinetics of Liquid Aerosol, Anal. Chem., 85, 5819-5826, doi:10.1012/ac4005502, 2013.

Fredenslund, A., Jones, R. L., and Prausnitz, J. M.: Group-contribution estimation of activity coefficients in nonideal liquid mixtures, AIChE Journal, 21, 1086-1099, doi:10.1002/aic.690210607, 1975.

Gupta, R. B. and Heidemann, R. A.: Solubility models for amino acids and antibiotics, AIChE Journal, 36, 333-341, 1990.

Hansen, H. K., Rasmussen, P., Fredenslund, A., Schiller, M., and Gmehling, J.: Vapor-liquid-equilibria by UNIFAC group contribution. 5. Revision and extension, Ind. Eng. Chem. Res., 30, 2352–2355, 1991.

Hodas, N., Zuend, A., Schilling, K., Berkemeier, T., Shiraiwa, M., Flagan, R. C., and Seinfeld, J. H.: Discontinuities in hygroscopic growth below and above water saturation for laboratory surrogates of oligomers in organic atmospheric aerosols, Atmos. Chem. Phys., 16, 12767-12792, doi:10.5194/acp-16-12767-2016, 2016.

Koop, T., Bookhold, J., Shiraiwa, M., and Poschl, U.: Glass transition and phase state of organic compounds: dependency on molecular properties and implications for secondary organic aerosols in the atmosphere, Phys. Chem. Chem. Phys., 13, 19238–19255, doi:10.1039/c1cp22617g, 2011.

Kuramochi, H., Noritomi, H., Hoshino, D., and Nagahama, K.: Representation of activity coefficients of fundamental biochemicals in water by the UNIFAC model, Fluid phase equilibria, 130, 117-132, 1997.

Marcolli, C. and Peter, T.: Water activity in polyol/water systems: new UNIFAC parameterization, Atmos. Chem. Phys., 5, 1545–1555, doi:10.5194/acp-5-1545-2005, 2005.

Peng, C., Chan, M. N., and Chan, C. K.: The hygroscopic properties of dicarboxylic and multifunctional acids: Measurements and UNIFAC predictions, Environ. Sci. Technol., 35, 4495–4501, doi:10.1021/es0107531, 2001.

Rovelli, G., Miles, R. E. H., Reid, J. P., and Clegg, S. L.: Accurate Measurements of Aerosol Hygroscopic Growth over a Wide Range in Relative Humidity, J. Phys. Chem. A, 120, 4376−4388, doi:10.1021/acs.jpca.6b04194, 2016.

Zuend, A. and Seinfeld, J. H.: Modeling the gas-particle partitioning of secondary organic aerosol: the importance of liquid-liquid phase separation, Atmos. Chem. Phys., 12, 3857-3882, doi:10.5194/acp-12-3857-2012, 2012.

Zuend, A., Marcolli, C., Booth, A. M., Lienhard, D. M., Soonsin, V., Krieger, U. K., Topping, D. O., McFiggans, G., Peter, T., and Seinfeld, J. H.: New and extended parameterization of the thermodynamic model AIOMFAC: calculation of activity coefficients for organic-inorganic mixtures containing carboxyl, hydroxyl, carbonyl, ether, ester, alkenyl, alkyl, and aromatic functional groups, Atmos. Chem. Phys., 11, 9155–9206, doi:10.5194/acp-11-9155-2011, 2011.

---

## Referee Comment (RC2) · Anonymous Referee #2 · 13 Feb 2017

Review on manuscript "Influence of Organic Compound Functionality on Aerosol Hygroscopicity: Dicarboxylic Acids, Alkyl-Substituents, Sugars and Amino Acids" by Marsh et al., submitted to ACP.

The authors reported the hygroscopic data of a series of dicarboxylic acids (DCAs) with subtle molecular structure changes, amino acids, and some sugars and alcohols using the Comparative Kinetics Electrodynamic Balance (CK-EDB). It allows the measurements of a mfs/RH curve within 10 seconds, with the advantage of reducing the loss of volatile organics from the particles. Experimental results have indicated potential improvements for the UNIversal quasichemical Functional group Activity Coefficients (UNIFAC) model and are compared with isotherm models in terms of kappa values. The manuscript was in general well written but some of the wordings are unnecessarily strong that make discussions somewhat confusing or even misleading.

Page 1 line 12: "The dual micro dispenser set up allows for sequential trapping of probe and sample droplets for accurate determination of droplet water activities from 0.45 to > 0.99." This sentence is not entirely correct.  The CK-EDB is based on kinetic measurements and it does have the advantage of fast measurements that reduce evaporation of volatile materials. However, the fast measurements would also potentially lead to non-equilibrium measurements, especially for some organics at low RH. The authors seem to admit a potential shortcoming of the technique on page 11, line 23, "For many compounds, measurements unimpeded by kinetic limitations have not been possible below 80 % RH, and consequently data presented below 80 % do not average to a consistent series of points".   More evidence to demonstrate that the measurements presented in this paper are equilibrium measurements would be needed.

Page 1 line 15: This significance of this sentence is not clear. New data agree better with the UNIFAC predictions than the old data, from which UNIFAC parameters were derived, do.   Are the UNIFAC parameters/predictions useful or not?

Page 1 line 22: The authors should discuss the agreement between the measured hygroscopicities and UNIFAC predictions on sugars/alcohols.

Page 8, line 25: "In addition, the short timescale of the measurement ensures that evaporation of the semi-volatile components, such as these dicarboxylic acids, is avoided". Has this been verified or is this merely an assumption? Did they experimentally verify this with some other semi-volatile solutes? Can they really say that evaporation is avoided?   In reality, there must be a range of vapor pressure that evaporation is "negligible" but appreciable at larger values.

Page 10, line 1: "In summary, UNIFAC predictions agree well with measurements for simple unbranched dicarboxylic acids with the exception of pimelic acid," Is there any explanation why the UNIFAC cannot predict the hygroscopicity of pimelic acid?   Can the authors provide suggestions to make improve the predictions of UNIFAC?

Page 10, line 30: "The model (equation 27 in Dutcher et al. 2013) is fitted to molality experimental

data with respect to water activity for finding the parameter value, which results in a significant improvement in the MFS than UNIFAC." Why does the multilayer adsorption isotherm based model from Dutcher et al. (2013) give a better prediction beyond the use of an adjustable parameter? Page 10 line 20. From Figure 8a, it seems L-threonine, rather than L-valine, deviates from the other three compounds most.

Page 11 line 6: Is there any other possible explanation for the discrepancy between measured and literature (Chan et al., 2005) data on those amino acids? Can they rule out the possibility of mass transfer effects or evaporation of solute? They show that their data are consistent with Na et al. (1995), which are compromised by evaporation of solute since they made EDB measurements in vacuum. Furthermore, Chan et al. also made measurements of these amino acids to lower RH to determine the mfs of solid of unity. The assertion that Chan et al. were wrong, which is possible, need to be accompanied by the discussions on how that would affect the mfs of the dried particles. Would the new data provided here yield unreasonable mfs of the dried particles?

Page 11 line 23: It is unclear why kinetic limitations will not affect the hygroscopicity measurements of sugars and alcohols in this study by saying "as established by the RH of the gas phase the droplet is drying in". Elaborate please.

Page 11 line 26: Would the C4-polyol be described as "long chain"? Would sorbitol be classified as sugar?

Page 12 line 25: The authors discussed on the over-estimation of kappa parameter. However, it seems the UManSysProp model can over-estimate as well as under-estimate the kappa (Figure 12b).

The authors seem to be sending very mixed messages on the reliability of simple parameterized models/equations for predicting hygroscopicity. On one hand, they criticized the limitation of UNIFAC in predicting hygroscopicity of branched acids. On the other hand, they promoted the use of $\kappa$ values and O:C and N:C ratios based on Figure 11(b) and 12 (a), which do show discrepancies between model predictions and experimental results in $\kappa$. When plotted in the form of mfs hygroscopic data, some of these differences are not much smaller than those between the measurements of the branched DCAs and the UNIFAC predictions based on parameterization of simpler acids. Furthermore, the comparison of data and model Kappa parameters are evaluated at 95% but the comparison of UNIFAC related results are in mfs as a function of RH. How would the results look like if Kappa values are evaluated at lower RH? Finally, are these predictive tools considering these very general and smooth relationships really much less computationally expensive than current group contribution methods? $\kappa$ was calculated by isotherms that require an adjustable parameters. Overall, the comments made by the authors on the use of UNIFAC, Kappa/isotherm models, and more elaborated models such as AIOMFAC and UManSysProp appear not to be unbiased. Elaboration is needed.

---

## Author Response (AR1)

**Response to Referee #1 (Andreas Zuend) on "Influence of Organic Compound Functionality on Aerosol Hygroscopicity: Dicarboxylic Acids, Alkyl-Substituents, Sugars and Amino Acids"**

Aleksandra Marsh[1], Rachael E. H. Miles[1], Grazia Rovelli[1], Alexander G. Cowling[1], Lucy Nandy[2], Cari S. Dutcher[2] and Jonathan. P Reid[1]
[1] School of Chemistry, University of Bristol, Bristol, BS8 1TS, UK
[2] Department of Mechanical Engineering, University of Minnesota, 111 Church Street SE, Minneapolis, MN 55455, USA

*Correspondence to:* Jonathan. P. Reid j.p.reid@bristol.ac.uk

The authors would like to thank Andreas Zuend (Referee #1) for his generally positive comments on the manuscript. We respond to the specific comments made by the referee below and identify the changes we have made to the manuscript.

On consideration of the comments, all predictions generated by AIOMFAC-web have been repeated with careful consideration of the functional groups used. Methyl malonic acid, 3-methyl adipic acid, dimethyl malonic acid, 2,3-dimethyl succinic acid and pimelic acid were found to differ marginally from the original predictions. All figures (Figure 2, 4, 5 and 7), supporting information and tabulated data have been updated to reflect these changes. Although the corrections lead to slight numerical changes, they do not alter the overall conclusions of the manuscript.

**Response to general comments**
*Referee Comment: Section 2. Methods and Materials. While the reader is referred to Rovelli et al. (2016) and Davis et al. (2013) for a detailed description of the CK-EDB method, a general description of the chemicals used, their purities and solution preparation is missing. Some of that information is provided in the SI 2 only. I suggest that a brief description is also given in the main text and that the reader should be informed about additional information on this in the SI.*

Response: As recommended by the referee, we have added a brief comment on P4 L21-23 to refer the reader to the detailed information in the SI: 'Purity and supplier for all compounds is presented in the supplementary information. Further, all measurements presented in this work are taken at 293.15 K. All solutions are prepared using HPLC grade water (VWR Chemicals).'

*Referee Comment: Temperature range and droplet temperatures. In the first paragraph of page 4 it is highlighted that the temperature in the EDB trapping region can be controlled well over a ~75 K range, however, throughout the main text information about the actual temperature used is missing (including tables and figures). As far as I can tell from the temperature information given in the SI, all experiments and model calculations were carried out at 293.15 K. Were hygroscopicity measurements at other temperatures considered (which would be useful, e.g. for improved, temperature-dependent thermodynamic model parameterisations given the temperatures were sufficiently different)?*

Response: We have now noted the temperature of all measurements (see previous response). We have conducted temperature dependent hygroscopicity experiments for a number of organic systems presented in this work and for a number of additional inorganic systems. However, these will be detailed in a subsequent publication due to the length of this manuscript and the length of discussion involved on the effect of temperature on hygroscopicity.

*Referee Comment: ...a discussion on the EDB temperature range used and the actual droplet surface temperature during the evaporation experiments will require some discussion. From Rovelli et al. (2016) it seems clear that the time scale of the evaporation will lead to deviations between droplet and surrounding gas phase temperature. Moreover, given that the evaporation rate from a relatively concentrated solution droplet*

*is different from the evaporation rate of the probe droplet, a discussion of such temperature related issues with respect to the retrieval of the sample droplet's water activity at a particular temperature seems appropriate.*

Response: We agree with the referee that accounting for the temperature suppression is important when retrieving the hygroscopic growth curve. We have provided an extensive discussion of this in our recent paper (Rovelli et al. 2016, as identified by the referee) where we provided considerable evidence to validate and benchmark the technique. Having previously provided this information in great detail, we feel that any further discussion provided here would be rather insubstantial and inferior to our previous discussion. Instead, we feel it is important that the reader be referred to the comprehensive account in our earlier report, with all procedures used in this manuscript carefully following our earlier recommendation. On P7 L11 we add the following comment: "It is imperative that the evaporative cooling be accounted for as this suppresses the apparent vapour pressure at any instant, particularly at early time when the mass flux is larger. Indeed, equation (5) explicitly accounts for the latent heat lost from the droplet. At very early times and when evaporating into low RH, the temperature suppression can be sufficient (>3 K) so as to reduce the accuracy of approximations made when deriving equation 5. Under these circumstances, when the temperature suppression is larger than this limit, we do not infer equilibrium water activities, but instead only retrieve the equilibrium hygroscopic growth when the temperature suppression is smaller than 3 K. This procedure has been discussed and verified in detail in our earlier work, and the reader is referred to Rovelli et al. (2016) for further details."

*Referee Comment: In contrast to the inorganic solutes used in Rovelli et al. (2016), the present study involves organic solutes, some of which may cause a significant increase in mixture viscosity with decreasing droplet water content during evaporation in the EDB. In this context, the time scale of 10 s for the evaporation from the droplets may become an issue for droplets > 10 μm radius, potentially impeding the droplet-gas mass transfer (e.g. Koop et al., 2011) and potentially violating assumptions about a homogeneous, concentration-gradient-free mixing of water and organic compound within sample droplets aside from a developing temperature gradient within a rapidly evaporating droplet. The authors discuss the viscosity concern in Section 3.3, where it is mentioned that for many compounds measurements unimpeded by kinetic limitations were not possible below 80 % RH. Because this consideration may not only apply to sugars and alcohols, but to many of the multifunctional organics of higher molar mass, a more general discussion of kinetic limitations and consequences for the CKEDB data processing should be provided in Section 2 where the method is described. If a relatively viscous binary aqueous droplet is exposed to low RH and evaporates water quickly, there may be insufficient time for homogeneous mixing in the droplet bulk compared to the near-surface volume of the droplet, which could lead to a concentration gradient and a higher solute concentration in the surface region of the droplet, affecting the local water activity there. Under such conditions, an organic solute may appear as more hygroscopic than it would be under actual gas-particle equilibrium conditions. Did the authors consider such effects in their method and the data processing? It is also not clear whether the authors considered a longer measurement time scale with slower evaporation settings for systems where substantial kinetic limitations may occur (and for which organic evaporation may not be a concern). Please discuss.*

Response: We have added the following discussion and Figures to the SI, "The kinetic modelling framework used in the analysis of the droplet evaporation events is valid only in the absence of a bulk-kinetic limitation on near surface composition, i.e. the particle must be assumed to be homogeneous in composition. Such a limitation was obvious for hygroscopicity measurements of trehalose, galactose and sorbitol at RH's lower than 80 %. To ensure the measurements are not compromised by bulk diffusion, we consider two important factors.

Firstly, the impact of viscosity on the hygroscopicity retrievals becomes very obvious when we consider the consistency and uncertainty in the raw hygroscopic growth curves determined from different droplets evaporating into differing RHs. Droplets drying into different RHs reach different compositions at different times, and will retain different amounts of water because of different drying rates. This leads to an artificially low MFS at a particular RH which then slowly returns to the equilibrium curve overtime. Thus, an inconsistency is apparent between retrieved hygroscopic growth curves (or MFS vs $a_w$) when drying into different RHs. An example of this is shown in Figure S39.1, where we report unbinned hygroscopicity data for alanine (a non-viscous amino acid) and trehalose (viscous at RHs lower than 80%). It is clear here that the different portions of the hygroscopic curves retrieved from measurements at different RHs are consistent for alanine but not for trehalose. A further easy way to identify this retention of water in a particle that is not fully

equilibrated is simply to measure the much longer time-dependence in size once the initial evaporation of water has stopped. In droplets that have reached a bulk diffusion limitation, the existence of a kinetic limitation is apparent in a steadily decreasing size as water continues to leave over a timescale longer than 10 s.

**Fig S39.1 a) Unbinned hygroscopicity data for the compound alanine.  b) Unbinned hygroscopicity data for the compound trehalose. At 50 % RH trehalose has a viscosity of 3.8 x 10$^5$ Pa.s (Song et al. 2016).**

[Figure]

Secondly, we can determine the expected conditions under which we might expect problems to arise in retrieving hygroscopic growth curves from an evaporation measurement. Considering again trehalose at 80 % RH, an aqueous-trehalose droplet has a viscosity of 0.5 Pa.s, increasing to $3.8 \times 10^5$ Pa.s at 50 % RH (Song et al. 2016). Therefore, as the RH of the gas phase for the evaporation measurement is lowered, we can expect the increasing viscosity/decreasing diffusivity to become increasingly important. By contrast, for aqueous-carboxylic acid droplets, the viscosity never gets above 1 Pa s even at the driest RHs considered here (Song et al. 2016).

With these known dependencies of viscosity on water activity, we can estimate the timescale for diffusional mixing within a droplet, assuming that this provides an estimate of the timescale for an evaporating droplet to form a homogeneous mixture. This timescale must be considerably shorter than the evaporation timescale for our hygroscopicity estimations to be valid. First, the Stokes-Einstein equation is used to estimate the diffusion constant of water at varying viscosity (varying RH).

$$D = \frac{k_B T}{6 \pi r_{mol} \eta} \qquad (1.1)$$

$D$ is the diffusion constant, $k_B$ is the Boltzmann constant, $T$ is temperature, $r_{mol}$ is the molecular radius of water (taken as 1.375 Å) and $\eta$ is the viscosity. It should be noted that equation (1.1) is likely to provide a significant underestimate of the diffusion constant due to the failure of the Stokes-Einstein equation. At a viscosity of 100 Pa s, the diffusion constant for water in sucrose is already more than one order of magnitude larger than estimated from the viscosity (Power et al. 2013). However, using diffusion constants estimated from (1.1) will provide an upper limit on the diffusional mixing timescale. The timescale for diffusional mixing, τ, is then estimated using the expression

$$\tau = \frac{a^2}{\pi^2 D} \qquad (1.2)$$

where $a$ is the droplet radius (set as 10 microns in this calculation).

We compare the diffusional mixing timescales for aqueous droplets of trehalose, NaCl, NaNO$_3$ and glutaric acid in the newly added supplemental Figure S39.2 (and repeated below). Given that we have been able to report accurate hygroscopic growth curves for NaNO$_3$ down to 50 % RH (see Rovelli et al. 2016 and the

response to referee 2), it is clear that a final viscosity at 50 % of ~ 0.1 Pa.s (Baldelli et al.) is insufficient to impede accurate measurement of the hygroscopicity. Indeed, this suggests that water transport in any aerosol droplet that maintains a viscosity lower than 0.1 Pa.s during drying should remain sufficiently fast to avoid a bulk diffusion limitation, permitting accurate hygrosocpicity measurements. As an example of the diacarboyxlic acids considered in this study, glutaric acid has a considerably lower viscosity at 50 % RH of ~ 0.01 Pa.s (Song et al. 2016), indicative of what we might expect for all such similar systems. By contrast, aqueous-trehalose droplets cross the 0.1 Pa.s viscosity threshold at a water activity of ~0.85 (Song et al. 2016), commensurate with the deviation and increased scatter in the hygroscopicity measurements reported above for this compound.

Again, we must reiterate that the true diffusion constants are generally found to be much larger than values estimated from the Stokes-Einstein equation. A droplet with a viscosity of 0.1 Pa s takes ~0.3 s to mix by diffusion based on our analysis here, but this is an upper limit on the timescale.

Based on the two considerations above and to indicate clearly the water activity ranges over which we consider the hygroscopicity measurements to be valid for trehalose (S30), galactose (S31) and sorbitol (S29), we have added a dashed line to indicate where the data appear to become kinetically limited. We have added the following words to the captions of these Figures: "Data taken at RHs lower than indicated by the dashed black line show increased error in hygroscopicity retrieval due to the imposition of a kinetic limitation on water transport."

**Fig S39.2 a) Viscosity of Trehalose, NaCl, NaNO3 and Glutaric Acid as a function of RH. b) Estimated diffusion constant as a function of RH. c) Timescale for diffusional mixing at the RH shown on x-axis. Dashed green line represents 1 second timescale for diffusional mixing.**

[Figure]

A. Baldelli, R. M. Power, R. E. H. Miles, J. P. Reid and R. Vehring *Effect of crystallization kinetics on the properties of spray dried microparticles,* Aerosol Science and Technology, 2016, 50:7, 693-704, DOI:10.1080/02786826.2016.1177163

R. M. Power, S. H. Simpson, J. P. Reid and A. J. Hudson, *The transition from liquid to solid-like behaviour in ultrahigh viscosity aerosol particles,* Chemical Science, 2013, 4 , 2597, DOI: 10.1039/c3sc50682g

Y. Chul Song, A. E. Haddrell, B. R. Bzdek, J. P. Reid, T. Bannan, D. O. Topping,, C. Percival, and C. Cai *Measurements and Predictions of Binary Component Aerosol Particle Viscosity* J. Phys. Chem. A 2016, 120, 8123−8137, DOI: 10.1021/acs.jpca.6b07835"

*Referee Comment: UNIFAC models – three general comments and clarifications:*
*1. The authors compare many of the measurements to predictions by "the" UNIFAC model, however, the information about the specific model version used and its parameterisation for some of the compounds is incomplete in the manuscript. While the original UNIFAC model theory by Fredenslund et al. (1975) is mentioned on page 2, several UNIFAC modifications 3 (changes to model equations, e.g. UNIFAC-Dortmund, UNIFAC-Lyngby, etc.) and several revisions of UNIFAC parameter tables applicable to certain UNIFAC versions have been published in the past 40 years. For example, the AIOMFAC model (Zuend et al., 2008; 2011), which includes a UNIFAC model based on the original theory of Fredenslund et al. (1975), relies mostly on the revised parameter set by Hansen et al. (1991). However, Zuend et al. (2011) discuss several modifications of the parameter database, including the use of improved interaction parameters determined by Marcolli and Peter (2005) for alcohols and multifunctional compounds containing hydroxyl groups, as well as modified interaction parameters by Peng et al. (2001) for a subset of interactions involving carboxylic acid groups. These modifications are detailed in Zuend et al. (2011) and are used in the online version of the AIOMFAC model (which was used for several comparisons with measurements in the present paper; see also www.aiomfac.caltech.edu/about.html). Similarly, the online UNIFAC versions in UManSysProp (http://umansysprop.seaes.manchester.ac.uk; Topping et al., 2016), which includes AIOMFAC and a UNIFAC version, and the E-AIM website's UNIFAC (www.aim.env.uea.ac.uk/aim/aim.php) contain modified parameter sets from Peng et al. (2001) and from other sources of UNIFAC parameter revisions (see also http://www.aim.env.uea.ac.uk/aim/phpmain/edit_help.php#section100 for details on UNIFAC in E-AIM). While some of these newer parameterisations lead to only slight changes to predicted water activities compared to the original UNIFAC by Fredenslund et al. (1975) with the Hansen et al. (1991) parameters, others are significant – and e.g. in the case of AIOMFAC, the description of alcohols and sugars is substantially modified by the introduction of specific subgroups and main groups in the model for these compounds (of relevance for the comparisons with CK-EDB data made in this study). Therefore, to provide sufficient detail for clarity and reproducibility, it is necessary to specify which models and parameterisations were actually applied (e.g. in the Methods section).*

Response: All UNIFAC model predictions presented in this paper for dicarboxylic acids, sugars and alcohols were performed using AIOMFAC-web. This has been specified both in section 3.1. (Hygroscopic Response of Dicarboxylic Acids of Varying Complexity) and section 3.3. (Sugars and Alcohols) and in all relevant captions (Figure 2, 3, 5 and 10). We have added the following to ensure clarity:

P8 L11: 'All calculations for dicarboxylic acids were performed using the AIOMFAC-web model.'
P2 L33: 'AIOMFAC-web implements several improved parameters which are detailed by Zuend et al. (2011).'
P10 L13: With regard to amino acid modelling we have added the following: 'Hence UNIFAC (AIOMFAC-web) thermodynamic model predictions for amino acids were generated using E-AIM, using the UNIFAC model with Peng et al. parameterization (Peng et al., 2001) and Model III (Clegg et al., 1998).'

*Referee Comment: 2. Contrary to the statements on page 2, lines 30 – 32 and on page 8, lines 20-21, UNIFAC (and AIOMFAC) actually account for the molecular structure and for certain differences between branched and straight-chain dicarboxylic molecules of the same molar mass – albeit in a limited way. For example, via the differing number in hydrogen atoms on $CH_2$, CH, and C subgroups, which leads to different values of the relative Van der Waals volume and surface area terms in the combinatorial part of the UNIFAC model for these alkyl subgroups (affecting predicted activity coefficients). For this reason, the UNIFAC subgroup assignments, as listed in Table S0 of the SI, are incomplete/incorrect in the case of the dicarboxylic acids. For example, $CH_n$ is not a UNIFAC/AIOMFAC subgroup and as such does not sufficiently characterise the compound; instead the appropriate subgroups need to be stated. For example, correct subgroup assignments show that the three distinct C7- dicarboxylic acids (see also Table 3 of Zuend et al. (2011): 3-methyl adipic acid, (CH3)(CH)(CH2)3(COOH)2, 3,3-dimethylglutaric acid, (CH3)2(C)(CH2)2(COOH)2, and pimelic acid, (CH2)5(COOH)2, have slightly different subgroup formulas in UNIFAC/AIOMFAC and consequently there should be distinct model curves in Fig. 5b and UNIFAC structure formulas in Table S0 of the SI. Although,*

*this reviewer agrees that the differences between UNIFAC predictions for such similar dicarboxylic acids are likely small.*

Response: We have amended P2 L 31 to read: 'In this approach, molecules are divided into characteristic molecular subgroups and the activity coefficients derived from group contributions with limited consideration for molecular structure.' Further, we have removed the text on P8 L20-21 that reads: '…a consequence of representing all CH, CH2, and CH3 substituents by CHn (Zuend et al., 2008).'

We would like to apologise because these AIOMFAC-web predictions were labelled incorrectly in the previous version of Figure 5(b) and have been corrected in the new Figure 5(b). The caption now reads: "….where the AIOMFAC-web prediction for 3-methyl adipic acid, $[(CH_3)(CH)(CH_2)_3(COOH)_2]$, 3,3-dimethylglutaric acid, $[(CH_3)_2(C)(CH_2)_2(COOH)_2]$, 2,2-dimethylglutaric acid, $[(CH_3)_2(C)(CH_2)_2(COOH)_2]$ is represented by the blue dashed line. Note that the equilibrium curves for the first 4 compounds are in such close agreement and indistinguishable on this scale that only one curve is shown for clarity. The prediction for pimelic acid $[(CH_2)_5(COOH)_2]$ is shown as a black solid line."

With respect to the AIOMFAC web prediction for 3-methyl adipic acid $[(CH_3)(CH)(CH_2)_3(COOH)_2]$, 3,3-dimethylglutaric acid $[(CH_3)_2(C)(CH_2)_2(COOH)_2]$, 2,2-dimethylglutaric acid $[(CH_3)_2(C)(CH_2)_2 (COOH)_2]$ and diethyl malonic acid $[(CH3)_2(CH_2)_2(C)(COOH)_2]$, the predicted equilibrium activity curves (mfs vs water activity) are so similar that they are indistinguishable, as shown in the figure below. As a consequence, only one curve is used to represent all four compounds in Fig. 5.

We have now explicitly included all functional groups used in the prediction of the AIOMFAC–web curves in Table S0 of the SI as suggested by the referee.

[Figure]

*Referee Comment: 3. The UNIFAC group-contribution method also offers another way to account for proximity effects by neighboring subgroups in organic molecules: specific subgroups can be assigned to larger sections of a molecular structure and that has been proposed for modified UNIFAC parameterisations in the case of amino acids. For example, Gupta and Heidemann (1990) introduced a specific "proline" UNIFAC subgroup (including a subset of determined interaction parameters for aqueous solutions of amino acids). Kuramochi et al. (1997) introduced a series of new functional groups and determined UNIFAC parameters for the description of most amino acids, including histidine, for a modified UNIFAC version based on "Larsen's UNIFAC". Thus, statements like (page 10, line 15): "UNIFAC predictions cannot be performed for all amino acids examined here; in particular, the ring structures found in proline and histidine, cannot be represented as subgroups in the current version of UNIFAC." are not generally correct – the UNIFAC parameterisation by Kuramochi et al. covers most amino acids studied experimentally in this work. However, it is correct that those "specialized" UNIFAC modifications are not implemented in the online versions of AIOMFAC and UNIFAC in E-AIM (see point (1) above), so they are not conveniently available for*

*calculations, which is likely what is meant by the authors' statement. Such general statements should therefore be revised accordingly and the work by Gupta and Heidemann (1990), Kuramochi et al. (1997) and others mentioned. Consider also that parameter sets that were determined for different UNIFAC model versions are typically not compatible and the use of specific subgroups with only a limited set of interaction parameters determined, e.g. for aqueous mixtures of amino acid solutions only, disqualifies the applicability of such models for predictions of complex, multi-component and multifunctional mixtures of interest in atmospheric aerosol chemistry (as discussed in Section 5.4 of Zuend et al., 2011).*

Response: We have amended section 3.2. P10 L14 now reads: "Amino acids form zwitterions in solution, supressing the vapour pressure of the acid, and this presents a challenge to current thermodynamic models with most not allowing the inclusion of nitrogen amine containing groups (e.g. AIOMFAC-web). AIOMFAC-web only allows for the inclusion of organonitrate and peroxy acyl nitrate sub groups. Hence, model predictions for amino acids were generated using E-AIM, using the UNIFAC model with Peng et al. parameterization (Peng et al., 2001) and Model III (Clegg et al., 1998). Even then UNIFAC predictions cannot be performed for all the amino acids examined here. In particular, the ring structures found in proline and histidine cannot be represented as subgroups in the current version of E-AIM, although these could be represented with the further parametrisations reported by Kuramochi et al. (1997b) or Gupta and Heidemann (1990)."

**Specific comments and technical corrections**
*Referee Comment: Abstract, first sentence and page 4, line 9: "Hygroscopic data" should be "Hygroscopicity data" (the data itself is likely not hygroscopic).*
Response: Corrected to: 'Hygroscopicity data for 36 organic compounds'

*Referee Comment: P3, line 7: correct spelling of "Köhler"*
Response: Corrected to: 'Köhler'

*Referee Comment: P3, l. 11 – 13: "Values are typically determined from sub-saturated hygroscopic growth measurements and reported at the highest accessible RH (Pajunoja et al., 2015). The value of κ can also be inferred from measurements of the critical supersaturation required for CCN activation, a measurement in a super-saturated regime (Carrico et al., 2008)." It would be appropriate to state that κ values determined at different RH and, to a lesser extent temperature, can vary substantially, especially when comparing κ determined from CCN activation data at water super-saturation compared to sub-saturation conditions, as, e.g., discussed by Hodas et al. (2016) and references mentioned therein.*
Response: P3 L15 We have added 'Further, κ values reported at different RHs can vary significantly and can also differ substantially from measurements in the supersaturated regime (Hodas et al., 2016).'

*Referee Comment: P3, l. 33: correct "(approaching [values] very close to 1)"*
Response: We have amended to read "(approaching values very close to 1)"

*Referee Comment: P4, l. 25: clarify the accuracy statement: "with a greater accuracy (< ± 0.2 % at water activities > 0.8…" do you mean < ± 0.2 % error in water activity or in hygroscopic growth factor or MFS?*
Response: We have clarified the statement to read: "accuracy (<±0.2 % error in water activity at water activities > 0.8 and ±1 % error in water activity at water activities < 0.8) than can be achieved in conventional approaches"

*Referee Comment: P6, l. 5: "with most solutes instead", better: "with most pure organic compounds instead" since this is not about a solution but about the pure components.*
Response: We have amended to read: 'corresponds to that of the pure sub-cooled melt with most pure organic compounds'

*Referee Comment:* *P6, title 2.3: Replace "hygroscopic" by hygroscopicity*
Response: We have amended to read '2.3. Extraction of Hygroscopicity properties'

*Referee Comment:* *P7, l. 2: "In this equation, the gradient in water partial pressure is the difference between the RH and aw, the instantaneous water activity at the droplet surface." First, given the evaporation setup with an RH profile dependent on the distance from the droplet, it needs to be stated which RH (and measured where) is meant, i.e. is it the RH at the droplet surface or the RH (sufficiently) far away from the droplet. Second, the difference (RH - aw) or rather saturation ratio S – aw) (as in Rovelli et al., 2016) alone does not constitute a "gradient". Also, since the component subscript "i" in Eq. (5) denotes water (i.e. subscript "w" as in aw), it would seem better to use "w" instead of "i".*

Response: We have added the following sentence to clarify that the probe droplet is trapped in exactly the same position as the sample droplet: "In this study, the probe droplets are trapped in exactly the same position within the gas flow as the sample droplets which allows the measurement of the RH in situ. The probe droplets are either pure water (for the RH range 80 – 99 %) or aqueous NaCl (for the RH range 50 – 80 %)."

When referring to gradient in the text, we are referring to the gradient in water partial pressure and we believe this is correct. We do not refer to a gradient formed from ($RH-a_w$). To be consistent with our previous publications, we have removed the subscript *i* entirely from the equation but not replaced it with *w*.

*Referee Comment:* *P7, l. 9: "is the latent heat of vaporization"; add "of water" at temperature $T_\infty$(?).*
Response: Added 'L is the latent heat of vaporization of water at $T_\infty$.'

*Referee Comment:* *P8, l. 10: "using Peng corrections" the meaning of this is unclear. Also, as detailed above, the UNIFAC models likely used by the authors actually include further modifications in terms of the used parameter sets and/or subgroup assignments.*
Response: All UNIFAC predictions (with the exception of amino acids) were performed using AIOMFAC-web and we hope we have now made this clear in the manuscript.

*Referee Comment:* *P8, l. 20: "In addition, the UNIFAC predictions become less accurate as the added substituent becomes larger, a consequence of representing all CH, CH2, and CH3 substituents by CHn (Zuend et al., 2008)." There seems to be a misunderstanding about the UNIFAC (AIOMFAC) way of group contribution calculations, see the general comment above. Only group-group interactions in the residual UNIFAC expressions are common for all CHn subgroups (with n = 0,1,2,3), but the volume and surface area terms (combinatorial part) are not. This is the case in all variants of UNIFAC.*
Response: We have removed 'a consequence of representing all CH, CH2, and CH3 substituents by CHn (Zuend et al., 2008).'

*Referee Comment:* *P9, l. 20 and l. 17: There are actually more than two distinct UNIFAC group formulas for the different C7-dicarboxylic acids, see the general comment above. Also, given that the UNIFAC (AIOMFAC) model predictions of water activity show a deviation from the CK-EDB data for the straight-chain pimelic acid, the model-measurement deviations shown in Fig. 5b are expected and at least consistent in that sense. Related to the statement on line 17, the observed similarity in hygroscopicity of the different C7-dicarboxylic acids suggests that the degree of branching and/or lengths of alkyl substituents may not always play a substantial role, in particular above a water activity of 0.8. This seems to be a counter-example to the trends observed for the smaller dicarboxylic acids with alkyl substitutions (and a hint for a general underestimation of the hygroscopicity contribution by the CHn groups as represented in UNIFAC/AIOMFAC).*

Response: We have now included all functional groups used for AIOMFAC-web predictions in the table in the SI for each compound. The referee makes an interesting observation about the general consequences of the observation of the underestimation of the contribution from CHn groups.

*Referee Comment: P10, l. 15: The sentence should be revised as certain nitrogen containing compounds are available in most UNIFAC models (including in AIOMFAC for organics + water systems) since the parameter set by Hansen et al. (1991) includes amine, amid, nitro, nitrile and pyridine groups and some version include organonitrate groups (Compernolle et al., 2009; Zuend and Seinfeld, 2012) and proline and histidine groups Kuramochi et al. (1997).*

Response: We have amended the section on P10, 3.2. Hygroscopic Response of Amino Acids, to read: "Amino acids form zwitterions in solution, supressing the vapour pressure of the acid, and this presents a challenge to current thermodynamic models with most not allowing the inclusion of nitrogen amine containing groups (e.g. AIOMFAC-web). AIOMFAC-web only allows for the inclusion of organonitrate and peroxy acyl nitrate sub groups. Hence, model predictions for amino acids were generated using E-AIM, using the UNIFAC model with Peng et al. parameterization (Peng et al., 2001) and Model III (Clegg et al., 1998). Even then UNIFAC predictions cannot be performed for all the amino acids examined here. In particular, the ring structures found in proline and histidine cannot be represented as subgroups in the current version of E-AIM, although these could be represented with the further parametrisations reported by Kuramochi et al. (1997b) or Gupta and Heidemann (1990)."

In addition, on P10 L28-29 we have stated: "We used the UNIFAC model with Peng et al. parameterization (Peng et al. 2001), typically run in E-AIM in the Model III mode (Clegg et al. 1998)."

*Referee Comment: P10, l. 22: "except for L-valine"; According to Fig. 8, L-Threonine behaves similar to L-valine even though it contains a hydroxyl group instead of a methyl group. So it seems that L-valine is not an exception or not the only one. Also, the UNIFAC prediction for glycine is missing in Fig. 8a.*
Response: We have added the UNIFAC prediction for glycine to Fig. 8a. We have also amended Page 10 line 20 "On a MFS scale, the hygroscopic response of these compounds is similar except for L-threonine which is less hygroscopic, an observation that is not expected given the additional hydrophilicity of the hydroxyl substituent."

*Referee Comment: P10, l. 30: "is fitted to molality experimental data"; molality of what? The last part of that sentence needs to be rephrased as well.*
Response: We have amended P10 L30 to read: "The model (equation 27 in Dutcher et al. 2013) is fitted to experimental data for solute molality as a function of water activity, in order to determine the adjustable model parameter. The model predicts solute activities and concentrations across all water activities, by combining short-range adsorption isotherm and long-range Debye-Huckel expressions. The isotherm model results in improvement in MFS predictions when compared to UNIFAC. However, the notable difference in accuracy between the two models is not overly surprising: the isotherm based model of Dutcher et al. 2013 has an adjustable parameter (Table S0.2), while UNIFAC is a fully predictive model."

*Referee Comment: P11, l. 13: Statement needs to be revised given the above clarification about specific UNIFAC parameterisations for aqueous solutions of amino acids.*
Response: On P11 L15, we have removed: "This is a consequence of the current reliance of the UNIFAC parameterisation on the data of Chan et al. (2005)."

*Referee Comment: P11, l. 15: "This is a consequence of the current reliance of the UNIFAC parameterisation on the data of Chan et al. (2005)." This statement is incorrect, because the UNIFAC models used by the authors do in fact not contain the modified parameters by Chan et al. (2005); rather, they are based on Hansen et al. (1991) and Peng et al. (2001) parameters for the amino acids. Also, as is clearly shown in Chan et al. (2005), their modified UNIFAC parameterisation yields similar results to the Peng et al. version in many cases and the Peng et al. parameterisation is in reasonable agreement with their own experimental data (e.g. for threonine). Therefore, the discrepancies between the new CK-EDB data and the UNIFAC model curves shown*

*indicate clear discrepancies among different experimental data sets, as is discussed by the authors in the first paragraph of page 11.*

Response: On P11 L15, we have removed "This is a consequence of the current reliance of the UNIFAC parameterisation on the data of Chan et al. (2005)."

*Referee Comment: P12, l. 27: "Molecular structures presented in Fig. 10 are the open chain form, which must be used during modelling using UNIFAC."; Why "must"? AIOMFAC also allows you to use the cyclic structure of sugars in aqueous solution, e.g. glucopyranose instead of glucose, if desired.*

Response: Cyclic sugar structures do not appear to be available on AIOMFAC-web. Amended P11 L27 to read 'Molecular structures presented in Fig. 10 are the open chain form, which must be used during modelling using AIOMFAC-web.'

*Referee Comment: P12, l. 10: and Fig. 11 & 12: replace the compound class labelled "organic acids" by a more appropriate label, e.g. "dicarboxylic acids", since amino acids are also organic acids but not part of that class.*

Response: Labels in Figures 11 and 12 have been amended from organic acids to dicarboxylic acids as suggested.

*Referee Comment: P12, l. 32: Statement is incorrect, see comment to P11, l. 15.*

Response: We have removed the statement: 'this is due to earlier experimental measurements by Chan et al. (2005) which have been used to parametrise UNIFAC'

*Referee Comment: P13, first paragraph. With respect to the applicability of the determined component-kappa values from binary data with a simple mixing rule for a complex mixture's total hygroscopicity parameter kappa, I suggest the authors consider in this section that it remains rather uncertain whether the kappa values determined based on binary water + amino acid data apply in multicomponent mixtures of relevance for atmospheric aerosol. This is because the substantial hygroscopicity exhibited by many of the amino acids, due to their zwitterionic nature in aqueous solution, may be affected substantially by the presence of inorganic acids and dissolved salts in aerosol mixtures, altering the partial water uptake contribution by the amino acids in a non-linear manner. This may motivate further experimental investigations for organic-inorganic mixtures with the CK-EDB and other setups.*

Response: We agree with the referee's comment and will indeed soon progress to measurements of the hygroscopic response of mixtures.

*Referee Comment: Table 1: State the temperature (range) for the measurements. Also the caption text and table header concerning SMILES needs revision.*

Response: We have added the temperature of the measurements to caption to read: "Table 1. Experimentally determined κ values at $a_w = 0.95$ for all compounds studied at 293.15 K, presented alongside κ values calculated using UManSysProp and the smile string used for this calculation." We have changed 'smile string' to 'SMILES String'

*Referee Comment: Fig. 1: Lower panel, at around 0.9 water activity, the red triangles-up and black triangles-down symbols suggest a larger scatter in experimental data than the binned data and error bars account for. It is unclear why if it is assumed that the different drying rates have similar measurement uncertainty? A brief discussion may be useful.*

Response: Each dataset has an associated uncertainty on water activity depending on the RH at which the evaporation occurs (Rovelli et al. 2016). The higher the RH in the gas phase, the slower the evaporation, and there is consequently a greater density of measured data points. This is why the dataset at the higher RH has a higher weighting in the overall averaged data.

*Referee Comment: Fig. 2: state the UNIFAC parameterisation used, if AIOMFAC-web was used, then stating that would be sufficiently specific.*

Response: AIOMFAC-web was used and this has been stated in the caption on appropriate figures: "UNIFAC predictions using AIOMFAC-web."

*Referee Comment: Fig. 6: the y-axis label "n(water)/n(solute)" would be better written as in Fig. 7 or perhaps in abbreviated form, such as nw/ns.*
Response: This has been amended to match that in Figure 7.

*Referee Comment: Figs. 8 and 9: The UNIFAC (Peng et al. parameterisation) model curve for Glycine is missing.*
Response: The UNIFAC prediction for Glycine has been added in Figures 8 and 9.

*Referee Comment: Fig. 9: Comparing this figure to Fig. 1 of Chan et al. (2005), it is clear that many experimental data points from Na et al. are missing, as well as bulk data by Kuramochi et al. (1997) to higher MFS/lower $a_w$ than shown and data by Ninni and Mereilles (2001) in Fig. 9b. Including all these measurements in Fig. 9 will provide a better comparison for the discussion concerning the substantial discrepancies found among the experimental data sets and in comparison to model predictions.*

Response: As suggested, more data points from Na et al. have been added in Figure 9(a) and (b). It should be noted that in Na et al. the parametrisation of the hygroscopicity data for glycine does not accurately reproduce the experimental data presented in their manuscript. Thus these points in Figure 9(a) were determined by reading a number of points from the graph and a curve with the following formula fit to them MFS = 0.65967 + 0.5305 $a_w$ -1.1458 $a_w^2$, and this equation was used to generate additional points now plotted on Figure 9a). All available data from Ninni and Mereilles (2001) and Kuramochi et al. (1997) have now been included in both Figure 9(a) and (b).

*Referee Comment: Supplementary Information: It would be useful to briefly state at the end of the main text what information is provided in the SI.*
Response: This statement has been added on P13 L4-5: 'The supplementary information provides tabulated hygroscopicity data for all compounds measured in this study, it also details compound purities, density and refractive index parametrisations for all compounds.'

*Referee Comment: SI, Table S0: the page numbers for different systems are listed, but the pages in the SI were not numbered. Also, the AIOMFAC subgroups stated for the dicarboxylic acids with CHn groups should be revised, see general comment. The "CHn(OH)" groups stated for citric acid, tartaric acid and other compounds should be stated with OH preferentially in superscript (e.g. CH2[OH] for a CH2 subgroup bonded to an OH group, which is specified separately) to avoid confusion about the number of OH groups present in the molecular structure (see also Table 3 of Zuend et al., 2011).*
Response: Supporting Information pages have now been numbered, groups in Table S0 have been labelled according to the suggestions above; an example for tartaric acid is $(COOH)_2 (OH)_2 (CH)_2^{(OH)}$.

*Referee Comment: SI, Fig. S8.1: Check the caption text and symbols in the figure. I do not see any coloured curves for data at different temperatures stated in the caption.*
Response: S8.1. This caption has been altered to remove any reference to temperature dependent data. This will be reported in a subsequent paper. 'Hygroscopicity of L-Valine, (Sigma Aldrich, Purity ≥ 98 %), at 293.15 K Open symbols, these CK-EDB experiments; black filled circles, literature data (Kuramochi et al.); solid black line, UNIFAC model (293.15 K).'

*Referee Comment: SI, S26 and S27: For aqueous PEG mixtures, much improved interaction parameters have been determined for a PEG-specific version of AIOMFAC, but these are not yet included in AIOMFAC web (see also Hodas et al., 2016).*
Response: We are grateful to the referee for highlighting this and will provide a further comparison when the refined parameters are available in AIOMFAC.

**Response to Anonymous Referee #2 on "Influence of Organic Compound Functionality on Aerosol Hygroscopicity: Dicarboxylic Acids, Alkyl-Substituents, Sugars and Amino Acids"**

Aleksandra Marsh[1], Rachael E. H. Miles[1], Grazia Rovelli[1], Alexander G. Cowling[1], Lucy Nandy[2], Cari S. Dutcher[2] and Jonathan. P Reid[1]

[1] School of Chemistry, University of Bristol, Bristol, BS8 1TS, UK

[2] Department of Mechanical Engineering, University of Minnesota, 111 Church Street SE, Minneapolis, MN 55455, USA

*Correspondence to:* Jonathan. P. Reid j.p.reid@bristol.ac.uk

The authors would like to thank the referee for their generally positive comments on the quality of the manuscript. We respond to the specific comments made by the referee below and identify the changes we have to the manuscript.

*Referee Comment: Page 1 line 12 "The dual micro dispenser set up allows for sequential trapping of probe and sample droplets for accurate determination of droplet water activities from 0.45 to > 0.99." This sentence is not entirely correct. The CK-EDB is based on kinetic measurements and it does have the advantage of fast measurements that reduce evaporation of volatile materials.*

Response: In prior publications we have provided considerable data that confirm the validity of the technique for a wide range of inorganic and organic systems. See particularly:

Rovelli, G., Miles, R. E. H., Reid, J. P., & Clegg, S. L. (2016). Hygroscopic Properties of Aminium Sulphate Aerosols. Atmospheric Chemistry and Physics Discussions. https://doi.org/10.5194/acp-2016-959

Rovelli, G., Miles, R. E. H., Reid, J. P., & Clegg, S. L. (2016). Accurate Measurements of Aerosol Hygroscopic Growth Over a Wide Range in Relative Humidity. The Journal of Physical Chemistry A, 120, 4376−4388. https://doi.org/10.1021/acs.jpca.6b04194

Davies, J. F., Haddrell, A. E., Rickards, A. M. J., & Reid, J. P. (2013). Simultaneous Analysis of the Equilibrium Hygroscopicity and Water Transport Kinetics of Liquid Aerosol. Analytical Chemistry, 85, 5819–5826. https://doi.org/dx.doi.org/10.1021/ac4005502

The referee is correct in suggesting that the measurements are in essence kinetic, however we still consider that our statement is accurate. Compared with the timescale of internal dynamics within the droplet (diffusional mixing, heat transport), the evaporative process may actually be considered to be sufficiently slow that the droplet adopts a uniform composition, a particular water activity and, consequently, the equilibrium vapour pressure for the solution composition at that moment in time. Indeed, based on the expression for the mass flux, equation (5), it is the water activity in the droplet that is the determining quantity and it is this that is measured.

*Referee Comment: However, the fast measurements would also potentially lead to non-equilibrium measurements, especially for some organics at low RH. The authors seem to admit a potential shortcoming of the technique on page 11, line 23, "For many compounds, measurements unimpeded by kinetic limitations have not been possible below 80 % RH, and consequently data presented below 80 % do not average to a consistent series of points". More evidence to demonstrate that the measurements presented in this paper are equilibrium measurements would be needed.*

Response: We have responded in detail to this comment in our response to referee #1. In brief, we have clearly shown in our earlier publications that equilibrium hygroscopicity curves can be retrieved with high accuracy. We reproduce below Figure 5 from Rovelli et al. (2016) as an example, comparing hygroscopicity measurements of inorganic salts taken using the CK-EDB with equilibrium thermodynamic model predictions,

all of which are in very good agreement. For some organic components that lead to viscous droplets on drying, departure from equilibrium behaviour with varying water content is clearly apparent from the measurements. We show in our response to referee #1 (and discuss further below) that the hygroscopicity measurement remains extremely accurate unless the viscosity increases above 0.1 Pa s. We have included the data for such systems in this manuscript to be entirely transparent and to demonstrate when the approach does not work, as well as showing when it does.

[Figure]

**Measured mfs vs $a_w$ plots for (NH₄)₂SO4, NaNO₃, Na₂SO₄, and NaCl (panels A–D). Symbols: filled circles, experimental data; solid lines, calculation from the E-AIM model. Note: error bars are smaller than the data point when not shown.**

*Referee Comment: Page 1 line 15: This significance of this sentence is not clear. New data agree better with the UNIFAC predictions than the old data, from which UNIFAC parameters were derived, do. Are the UNIFAC parameters/predictions useful or not?*

Response: The reproducibility of our measurement is much better than that of Peng et al. and, thus, our method is much better equipped to help refine the UNIFAC model parameters than the work of Peng et al.

*Referee Comment: Page 1 line 22 - The authors should discuss the agreement between the measured hygroscopicities and UNIFAC predictions on sugars/alcohols.*

Response: The level of agreement between the measurements and predictions for the sugars/alcohols is highly system dependent and no general statements can be made. Instead, we have added the phrase: "and show variable levels of agreement with predictions" to the end of line 22.

*Referee Comment: Page 8, line 25: "In addition, the short timescale of the measurement ensures that evaporation of the semi-volatile components, such as these dicarboxylic acids, is avoided". Has this been verified or is this merely an assumption? Did they experimentally verify this with some other semi-volatile solutes? Can they really say that evaporation is avoided? In reality, there must be a range of vapor pressure*

*that evaporation is "negligible" but appreciable at larger values.*

Response: As a first indicator, it is clear that the evaporation of water (with a gradient in vapour pressure of >100 Pa from the droplet surface to infinite distance at ~80 % RH) occurs in a timescale of ~ 10 s. Thus, components less than 1 Pa can be expected to take >1000 s to evaporate. More precisely, we can estimate the lower limit in vapour pressure that the CK-EDB is sensitive to in a 10 s period, equivalent to the hygroscopicity measurement timescale. The uncertainty in radius in a CK-EDB measurement is ± 100 nm. Then, assuming a droplet size change from 10.1 to 10 μm in the measurement period of 10 s due to volatilisation, we can infer the vapour pressure of the volatile component, $p_i$ (Pa) would be 0.43 Pa, using the Maxwell equation:

$$\frac{dr^2}{dt} = -\frac{2M_i D_i}{RT\rho}\, p_i \qquad\qquad (1.0)$$

$r$ is radius (m), $t$ is time (s), $M_i$ is molecular weight (kg mol$^{-1}$, taken here as 100 g mol$^{-1}$), $D_i$ is the gas phase diffusion constant (~1×10$^{-5}$ m$^2$ s$^{-1}$) , $R$ is the ideal gas constant, $T$ is the temperature (K) and $\rho$ is the droplet solution density (kg mol$^{-1}$, taken here as 1.46 g cm$^{-3}$). This is considerably more volatile than any of the species that we investigate in this study; for example the vapour pressure of malonic acid (one of the most volatile species in this study) is 10$^{-4}$ Pa. Thus, during the timescale of the measurement only water is lost from the droplet.

*Referee Comment: Page 10, line 1 - "In summary, UNIFAC predictions agree well with measurements for simple unbranched dicarboxylic acids with the exception of pimelic acid," Is there any explanation why the UNIFAC cannot predict the hygroscopicity of pimelic acid? Can the authors provide suggestions to make improve the predictions of UNIFAC?*

Response: We can only comment that pimelic acid was not included in the original compounds used to parametrise the UNIFAC groups for the dicarboxylic acids.

*Referee Comment: Page 10, line 30 - "The model (equation 27 in Dutcher et al. 2013) is fitted to molality experimental data with respect to water activity for finding the parameter value, which results in a significant improvement in the MFS than UNIFAC." Why does the multilayer adsorption isotherm based model from Dutcher et al. (2013) give a better prediction beyond the use of an adjustable parameter?*

Response: We agree that this was not clear.  The sentences now read (P10 L30): "The model (equation 27 in Dutcher et al. 2013) is fitted to experimental data for solute molality as a function of water activity, in order to determine the adjustable model parameter. The model predicts solute activities and concentrations across all water activities, by combining short-range adsorption isotherm and long-range Debye-Huckel expressions. The isotherm model results in improvement in MFS predictions when compared to UNIFAC. However, the notable difference in accuracy between the two models is not overly surprising: the isotherm based model of Dutcher et al. 2013 has an adjustable parameter (Table S0.2), while UNIFAC is a fully predictive model."

*Referee Comment: Page 10 line 20. From Figure 8a, it seems L-threonine, rather than L-valine, deviates from the other three compounds most.*
Response: We agree with the reviewer and have amended the sentence  Page 10 line 20 "On a MFS scale, the hygroscopic response of these compounds is similar except for L-threonine which is less hygroscopic, an observation that is not expected given the additional hydrophilicity of the hydroxyl substituent."

*Referee Comment: Page 11 line 6: Is there any other possible explanation for the discrepancy between measured and literature (Chan et al., 2005) data on those amino acids? Can they rule out the possibility of mass transfer effects or evaporation of solute? They show that their data are consistent with Na et al. (1995), which are compromised by evaporation of solute since they made EDB measurements in vacuum. Furthermore, Chan et al. also made measurements of these amino acids to lower RH to determine the mfs of solid of unity. The assertion that Chan et al. were wrong, which is possible, need to be accompanied by the discussions on*

*how that would affect the mfs of the dried particles. Would the new data provided here yield unreasonable mfs of the dried particles?*

Response: Volatilisation is not a concern for the amino acid measurements on the timescale of our measurement: once solvated in water, amino acids become zwitterionic and behave more like inorganic salts. In addition, in our previous response we have shown that volatilisation is insufficient to give any appreciable change in composition during the measurement time frame for any component with a vapour pressure less than 1 Pa. At a MFS as low as 0.0075, the pH of an aqueous solution of glycine is high enough for nearly 100 % of the solution to contain zwitterionic amino acid species (Kuramochi et al. (1997)). To further confirm this, we show below measurements made using the aerosol optical tweezers technique for an aqueous glycine droplet, suspended at variable RH's for thousands of seconds. Even over such prolonged periods, once water was removed following a step in RH from 75 % to 55 %, there was minimal observed volatilisation. For example, over a period of 1000 s, the size decreases from 3227 to 3218 nm (between times of 5000 and 6000 s). Thus, over a 10 s measurement time in our CK-EDB measurements, volatilisation can be ignored. By contrast, the data taken by Chan et al. (2005) require much longer confinement of droplets in the EDB (10 – 20 hours) (Chan et al., 2005).

Further, our CK-EDB measurements reach a sufficiently high enough RH to *overlap* with all bulk data available from Kuramochi et al. (1997) and Ninni and Mereilles (2001), which both show very good agreement. Instead, we believe that the discrepancy between our data and that of Chan et al. arises because of the extrapolation required to reconcile their first aerosol phase measurement (at the highest water activity) with the bulk data.

Based on these two facts, our data are not compromised by volatilisation and overlap with bulk data where available. By contrast, neither of these two concerns can be ignored when considering the data of Chan et al. We must stress that there are challenges associated with these measurements, particularly in reaching a high enough water activity to overlap and compare with bulk data, a challenge which our technique is able to address.

Chan, M. N., Choi, M. Y., Ng, N. L., and Chan, C. K.: Hygroscopicity of Water-Soluble Organic Compounds in Atmospheric Aerosols, Environ. Sci. Technol., 39, 1555-1562, 2005.

[Figure]

**Glycine hygroscopicity experiment a) Radius vs time and RI vs time.**

*Referee Comment: Page 11 line 23: It is unclear why kinetic limitations will not affect the hygroscopicity measurements of sugars and alcohols in this study by saying "as established by the RH of the gas phase the droplet is drying in". Elaborate please.*

Response: We have added the following discussion and Figures to the SI, "The kinetic modelling framework used in the analysis of the droplet evaporation events is valid only in the absence of a bulk-kinetic limitation on near surface composition, i.e. the particle must be assumed to be homogeneous in composition. Such a limitation was obvious for hygroscopicity measurements of trehalose, galactose and sorbitol at RH's lower than 80 %. To ensure the measurements are not compromised by bulk diffusion, we consider two important factors.

Firstly, the impact of viscosity on the hygroscopicity retrievals becomes very obvious when we consider the consistency and uncertainty in the raw hygroscopic growth curves determined from different droplets

evaporating into differing RHs. Droplets drying into different RHs reach different compositions at different times, and will retain different amounts of water because of different drying rates. This leads to an artificially low MFS at a particular RH which then slowly returns to the equilibrium curve overtime. Thus, an inconsistency is apparent between retrieved hygroscopic growth curves (or MFS vs $a_w$) when drying into different RHs. An example of this is shown in Figure S39.1, where we report unbinned hygroscopicity data for alanine (a non-viscous amino acid) and trehalose (viscous at RHs lower than 80%). It is clear here that the different portions of the hygroscopic curves retrieved from measurements at different RHs are consistent for alanine but not for trehalose. A further easy way to identify this retention of water in a particle that is not fully equilibrated is simply to measure the much longer time-dependence in size once the initial evaporation of water has stopped. In droplets that have reached a bulk diffusion limitation, the existence of a kinetic limitation is apparent in a steadily decreasing size as water continues to leave over a timescale longer than 10 s.

**Fig S39.1 a) Unbinned hygroscopicity data for the compound alanine.  b) Unbinned hygroscopicity data for the compound trehalose. At 50 % RH trehalose has a viscosity of 3.8 x 10⁵ Pa.s (Song et al. 2016).**

[Figure]

Secondly, we can determine the expected conditions under which we might expect problems to arise in retrieving hygroscopic growth curves from an evaporation measurement. Considering again trehalose at 80 % RH, an aqueous-trehalose droplet has a viscosity of 0.5 Pa.s, increasing to $3.8 \times 10^5$ Pa.s at 50 % RH (Song et al. 2016). Therefore, as the RH of the gas phase for the evaporation measurement is lowered, we can expect the increasing viscosity/decreasing diffusivity to become increasingly important. By contrast, for aqueous-carboxylic acid droplets, the viscosity never gets above 1 Pa s even at the driest RHs considered here (Song et al. 2016).

With these known dependencies of viscosity on water activity, we can estimate the timescale for diffusional mixing within a droplet, assuming that this provides an estimate of the timescale for an evaporating droplet to form a homogeneous mixture. This timescale must be considerably shorter than the evaporation timescale for our hygroscopicity estimations to be valid. First, the Stokes-Einstein equation is used to estimate the diffusion constant of water at varying viscosity (varying RH).

$$D = \frac{k_B T}{6\pi r_{mol}\eta} \qquad (1.1)$$

$D$ is the diffusion constant, $k_B$ is the Boltzmann constant, $T$ is temperature, $r_{mol}$ is the molecular radius of water (taken as 1.375 Å) and $\eta$ is the viscosity. It should be noted that equation (1.1) is likely to provide a significant underestimate of the diffusion constant due to the failure of the Stokes-Einstein equation. At a viscosity of 100 Pa s, the diffusion constant for water in sucrose is already more than one order of magnitude larger than estimated from the viscosity (Power et al. 2013). However, using diffusion constants estimated from (1.1) will provide an upper limit on the diffusional mixing timescale. The timescale for diffusional mixing, τ, is then estimated using the expression

$$\tau = \frac{a^2}{\pi^2 D} \qquad (1.2)$$

where $a$ is the droplet radius (set as 10 microns in this calculation).

We compare the diffusional mixing timescales for aqueous droplets of trehalose, NaCl, NaNO₃ and glutaric acid in the newly added supplemental Figure S39.2 (and repeated below). Given that we have been able to report accurate hygroscopic growth curves for NaNO₃ down to 50 % RH (see Rovelli et al. 2016 and the response to referee 2), it is clear that a final viscosity at 50 % of ~ 0.1 Pa.s (Baldelli et al.) is insufficient to impede accurate measurement of the hygroscopicity. Indeed, this suggests that water transport in any aerosol droplet that maintains a viscosity lower than 0.1 Pa.s during drying should remain sufficiently fast to avoid a bulk diffusion limitation, permitting accurate hygrosocpicity measurements. As an example of the diacarboyxlic acids considered in this study, glutaric acid has a considerably lower viscosity at 50 % RH of ~ 0.01 Pa.s (Song et al. 2016), indicative of what we might expect for all such similar systems. By contrast, aqueous-trehalose droplets cross the 0.1 Pa.s viscosity threshold at a water activity of ~0.85 (Song et al. 2016), commensurate with the deviation and increased scatter in the hygroscopicity measurements reported above for this compound.

Again, we must reiterate that the true diffusion constants are generally found to be much larger than values estimated from the Stokes-Einstein equation. A droplet with a viscosity of 0.1 Pa s takes ~0.3 s to mix by diffusion based on our analysis here, but this is an upper limit on the timescale.

Based on the two considerations above and to indicate clearly the water activity ranges over which we consider the hygroscopicity measurements to be valid for trehalose (S30), galactose (S31) and sorbitol (S29), we have added a dashed line to indicate where the data appear to become kinetically limited. We have added the following words to the captions of these Figures: "Data taken at RHs lower than indicated by the dashed black line show increased error in hygroscopicity retrieval due to the imposition of a kinetic limitation on water transport."

**Fig S39.2 a) Viscosity of Trehalose, NaCl, NaNO3 and Glutaric Acid as a function of RH. b) Estimated diffusion constant as a function of RH. c) Timescale for diffusional mixing at the RH shown on x-axis. Dashed green line represents 1 second timescale for diffusional mixing.**

[Figure]

A. Baldelli, R. M. Power, R. E. H. Miles, J. P. Reid and R. Vehring *Effect of crystallization kinetics on the properties of spray dried microparticles,* Aerosol Science and Technology, 2016, 50:7, 693-704, DOI:10.1080/02786826.2016.1177163

R. M. Power, S. H. Simpson, J. P. Reid and A. J. Hudson, *The transition from liquid to solid-like behaviour in ultrahigh viscosity aerosol particles,* Chemical Science, 2013, 4 , 2597, DOI: 10.1039/c3sc50682g
Y. Chul Song, A. E. Haddrell, B. R. Bzdek, J. P. Reid, T. Bannan, D. O. Topping,, C. Percival, and C. Cai *Measurements and Predictions of Binary Component Aerosol Particle Viscosity* J. Phys. Chem. A 2016, 120, 8123−8137, DOI: 10.1021/acs.jpca.6b07835"

*Referee Comment: Page 11 line 26: Would the C4-polyol be described as "long chain"? Would sorbitol be classified as sugar?*

Response: We have removed 'long chain' in the description of erythritol. Sorbitol and erythritol are best described as polyols (sugar alcohols) and they have now been correctly labelled in the amended text Page 11 line 26: "Equilibrium hygroscopicity curves for the two sugars galactose and xylose, and two sugar alcohols (polyols) erythritol and sorbitol, are shown in Fig. 10."

*Referee Comment: Page 12 line 25: The authors discussed on the over-estimation of kappa parameter. However, it seems the UManSysProp model can over-estimate as well as under-estimate the kappa (Figure 12b).*

Response: We agree with the referees comment and have expanded this sentence to indicate that the parameter can be underestimated. We have added on P12 line 25 "Further, UManSysProp predictions can also lead to an underestimation of $\kappa$ for a limited number of compounds, including valine, histidine, and glutaric and methyl succinic acid."

*Referee Comment: The authors seem to be sending very mixed messages on the reliability of simple parameterized models/equations for predicting hygroscopicity. On one hand, they criticized the limitation of UNIFAC in predicting hygroscopicity of branched acids. On the other hand, they promoted the use of $\kappa$ values and O:C and N:C ratios based on Figure 11(b) and 12 (a), which do show discrepancies between model predictions and experimental results in $\kappa$. When plotted in the form of mfs hygroscopic data, some of these differences are not much smaller than those between the measurements of the branched DCAs and the UNIFAC predictions based on parameterization of simpler acids. Furthermore, the comparison of data and model Kappa parameters are evaluated at 95% but the comparison of UNIFAC related results are in mfs as a function of RH. How would the results look like if Kappa values are evaluated at lower RH? Finally, are these predictive tools considering these very general and smooth relationships really much less computationally expensive than current group contribution methods? $\kappa$ was calculated by isotherms that require an adjustable parameters. Overall, the comments made by the authors on the use of UNIFAC, Kappa/isotherm models, and more elaborated models such as AIOMFAC and UManSysProp appear not to be unbiased. Elaboration is needed.*

Response: The main purpose of this manuscript is to provide an evaluation of the current thermodynamic models for predicting the hygroscopic response of aerosols. Models such as AIOMFAC, UNIFAC, E-AIM and UManSysProp will always be the preferred choice when a full and accurate representation of the equilibrium response is required (e.g. over a wide range in RH as the referee suggests). However, simple parametrisations such as $\kappa$ are finding very widespread use in providing an albeit limited characterisation of the hygroscopic response – a single parameter is easier to report and measure than a full hygroscopicity curve. We are not aware of anywhere in the paper where we make a value judgement on the relative merits of these two approaches: both methods have the strengths and uses in different domains and are used to provide the appropriate level of information required in any application. Given that the $\kappa$ treatment is increasingly used, we felt it was of importance to try and connect the more detailed thermodynamic models with the much more simplistic approach. We hope we have achieved this in the revised manuscript.

[revised manuscript text omitted]

**Table S0.1** Fitted parameters for upper and lower MFS vs water activity of compounds in each class, amino and organic acids, sugars and alcohols, as shown in Figure 11b) in the manuscript. The power law coefficient $P$ is used to calculate energy parameter $C$ for the first to $(n-1)$th layers, hence $C_i = (i/n)^P$, where $i$ is the layer number and $n$ is the total number of hydration layers, here $n = 8$ for all compounds except glycine ($n = 3$) and 2,2-dimethyl glutaric acid ($n = 16$). MSE is a normalized mean-square error, equal to $\left(\frac{1}{n_p}\right)\sum_{i=1}^{n_p}((m_{model,i} - m_{data,i})/(m_{model,i}))^2$, where $n_p$ is the number of data points.

| Solute | $P$ | MSE |
|---|---|---|
| Amino acid Upper (Glycine) | -1.934 | 0.00321 |
| Amino acid Lower (Asparagine) | -0.171 | 0.04151 |
| Organic acid Upper (Malonic acid) | -0.212 | 0.00819 |
| Organic acid Lower (2,2 dimethyl glutaric acid) | 0.206 | 0.08315 |
| Sugar Upper (Sorbitol) | -0.522 | 0.01025 |
| Sugar Lower (Trehalose) | -0.870 | 0.01687 |
| Alcohol Upper (Erythritol) | -0.238 | 0.01311 |
| Alcohol Lower (PEG4) | -1.180 | 0.16205 |

**Table S0.2** Fitted parameters for nine amino acids. The power law coefficient $P$ is used to calculate energy parameter $C$ for the first to $(n-1)$th layers, hence $C_i = (i/n)^P$, where $i$ is the layer number and $n$ is the total number of hydration layers, here $n = 8$ for all compounds except glycine ($n = 3$) and threonine ($n = 5$). MSE is a normalized mean-square error, equal to $\left(\frac{1}{n_p}\right)\sum_{i=1}^{n_p}((m_{model,i} - m_{data,i})/(m_{model,i}))^2$, where $n_p$ is the number of data points. (Parameter for L-aspartic acid could not be determined due to data range available.)

| Solute | $P$ | MSE |
|---|---|---|
| Alanine | -0.356 | 0.00051 |
| Asparagine | -0.171 | 0.04151 |
| Arginine | -0.993 | 0.04039 |
| Glycine | -1.934 | 0.00321 |
| Histidine | -0.502 | 0.02211 |
| Lysine | -1.225 | 0.00667 |
| Proline | -0.619 | 0.03764 |
| Threonine | -0.960 | 0.20107 |
| Valine | -0.892 | 0.00397 |

**S1 DL-Alanine Hygroscopicity**

**Fig. S1.1**: Hygroscopicity of DL-Alanine (Sigma Aldrich, Purity 99 %) at 293.15 K.

[Figure]

[Figure]

**Table S1.1:** Pure component refractive index ($n_{melt}$) is determined using molar refraction, assuming ideal mixing for calculation of the melt density ($\rho_{melt}$), from bulk data available in Cai et al. (2016). The variation of density as a function of the root of solute mass fraction (MFS$^{1/2}$ =x) is represented by polynomial fit parameters. *Upper* and *lower* refer to 95 % confidence limits for fits to experimental data, (Section 2.2 in manuscript).

| | $n_{melt}$ | $\rho_{melt}$/ g.cm$^{-3}$ | Polynomial fit ( $\rho_{sol} = a + b_1x + b_2x^2 + b_3x^3$) | | | |
| | | | a | $b_1$ | $b_2$ | $b_3$ |
|---|---|---|---|---|---|---|
| *Best* | 1.6205 | 1.4961 | 999 | 94.14 | -66.93 | 466.48 |
| *Upper* | 1.6222 | 1.5042 | 999 | 97.38 | -76.61 | 480.88 |
| *Lower* | 1.6188 | 1.4881 | 999 | 90.98 | -57.61 | 452.44 |

**Table S1.2:** Tabulated experimental data points shown in **Fig S1.1.**

| $a_w$ | error $a_w$ (+ve) | error $a_w$ (-ve) | MFS | error MFS |
|---|---|---|---|---|
| 0.75966 | 0.00182 | 0.00228 | 0.48336 | 8.66E-04 |
| 0.76866 | 1.02E-03 | 0.00128 | 0.47642 | 3.48E-04 |
| 0.77876 | 0.00413 | 0.00519 | 0.46428 | 0.00314 |
| 0.78887 | 0.00613 | 0.00771 | 0.45228 | 0.00412 |
| 0.80001 | 0.00674 | 0.00847 | 0.43959 | 0.00395 |
| 0.81774 | 0.00674 | 8.47E-03 | 0.41748 | 0.00512 |
| 0.83836 | 6.17E-03 | 7.75E-03 | 0.38848 | 6.39E-03 |
| 0.85334 | 0.00473 | 3.00E-03 | 0.37246 | 0.00116 |
| 0.86108 | 0.00116 | 7.54E-04 | 0.3655 | 8.92E-04 |
| 0.87144 | 6.02E-04 | 4.18E-04 | 0.34973 | 5.50E-04 |
| 0.87774 | 0.00139 | 9.81E-04 | 0.3386 | 0.00165 |
| 0.88866 | 0.00217 | 0.00153 | 0.31805 | 0.00274 |
| 0.89931 | 2.66E-03 | 1.92E-03 | 0.2981 | 0.00333 |
| 0.9087 | 0.00257 | 0.00183 | 0.27841 | 0.00381 |
| 0.91923 | 0.00256 | 0.00191 | 0.25483 | 0.00426 |
| 0.92957 | 0.00248 | 0.00181 | 0.23142 | 0.00416 |
| 0.93936 | 0.00243 | 0.00179 | 0.20672 | 0.00392 |
| 0.94936 | 0.00206 | 1.54E-03 | 0.18027 | 0.00361 |
| 0.9595 | 1.95E-03 | 1.43E-03 | 0.15211 | 0.00347 |
| 0.96935 | 1.49E-03 | 1.11E-03 | 0.12252 | 0.003 |
| 0.97954 | 0.00127 | 9.28E-04 | 0.08929 | 0.00254 |
| 0.99143 | 6.34E-04 | 7.22E-04 | 0.04259 | 0.00238 |

**S2 L-Arginine Hygroscopicity**

**Fig S2.1**: Hygroscopicity of L-Arginine, (Acros Organics, Purity > 98 %), at 293.15 K. Open squares, these experiments.

[Figure]

**Table S2.1:** Pure component refractive index ($n_{melt}$) is determined using molar refraction, assuming ideal mixing for calculation of the melt density ($\rho_{melt}$), from bulk data available in Cai et al. (2016). The variation of density as a function of the root of solute mass fraction (MFS $^{1/2}$ =x) is represented by polynomial fit parameters. *Upper* and *lower* refer to 95 % confidence limits for fits to experimental data, (Section 2.2 in manuscript).

| | $n_{melt}$ | $\rho_{melt}$/ g.cm$^{-3}$ | Polynomial fit ( $\rho_{sol} = a + b_1 x + b_2 x^2 + b_3 x^3$) | | | |
| --- | --- | --- | --- | --- | --- | --- |
| | | | a | $b_1$ | $b_2$ | $b_3$ |
| *Best* | 1.637 | 1.3995 | 998.6 | 59.85 | 28.54 | 310.48 |
| *Upper* | 1.6382 | 1.4045 | 998.6 | 61.44 | 24.47 | 317.9 |
| *Lower* | 1.6358 | 1.3945 | 998.6 | 58.28 | 32.51 | 303.13 |

**Table S2.2:** Tabulated experimental data points shown in **Fig S2.1.**

| $a_w$ | error $a_w$ (+ve) | error $a_w$ (-ve) | MFS | error MFS |
| --- | --- | --- | --- | --- |
| 0.50205 | 0.00177 | 0.0021 | 0.69041 | 0.00113 |
| 0.59399 | 0.00171 | 0.00206 | 0.65822 | 0.00115 |
| 0.69132 | 0.00127 | 0.00157 | 0.62391 | 1.74E-04 |
| 0.71788 | 0.01297 | 0.01296 | 0.61768 | 0.00607 |
| 0.74796 | 0.0139 | 0.01716 | 0.6026 | 0.00755 |
| 0.80315 | 7.87E-04 | 0.001 | 0.55889 | 4.50E-04 |
| 0.84439 | 0.00739 | 0.00741 | 0.52351 | 0.014 |
| 0.89694 | 0.00128 | 0.00112 | 0.47038 | 0.00138 |
| 0.91074 | 0.00174 | 0.00175 | 0.44361 | 0.00473 |
| 0.96538 | 0.00317 | 0.00317 | 0.24814 | 0.01569 |
| 0.99761 | 5.53E-04 | 5.28E-04 | 0.03416 | 0.00266 |

**S3 Glycine Hygroscopicity**

**Fig S3.1**: Hygroscopicity of Glycine, (Santa Cruz Biotech LTD), at 293.15 K. Solid line standard UNIFAC prediction.

[Figure]

[Figure]

**Table S3.1:** Pure component refractive index ($n_{melt}$) is determined using molar refraction, assuming ideal mixing for calculation of the melt density ($\rho_{melt}$), from bulk data available in Cai et al. (2016). The variation of density as a function of the root of solute mass fraction (MFS$^{1/2}$ =x) is represented by polynomial fit parameters. *Upper* and *lower* refer to 95 % confidence limits for fits to experimental data, (Section 2.2 in manuscript).

| | $n_{melt}$ | $\rho_{melt}/ g.cm^{-3}$ | Polynomial fit ($\rho_{sol} = a + b_1x + b_2x^2 + b_3x^3$) | | | |
|---|---|---|---|---|---|---|
| | | | a | $b_1$ | $b_2$ | $b_3$ |
| *Best* | 1.6634 | 1.6905 | 999.47 | 186.75 | -363.66 | 860.4 |
| *Upper* | 1.6654 | 1.7006 | 999.47 | 192.41 | -382.69 | 883.61 |
| *Lower* | 1.6613 | 1.6805 | 999.47 | 181.22 | -345.14 | 837.67 |

**Table S3.2:** Tabulated experimental data points shown in **Fig S3.1.**

| $a_w$ | error $a_w$ (+ve) | error $a_w$ (-ve) | MFS | error MFS |
|---|---|---|---|---|
| 0.51061 | 0.00328 | 0.00389 | 0.63189 | 0.00159 |
| 0.52315 | 0.0204 | 0.02421 | 0.62993 | 0.00129 |
| 0.57855 | 0.01673 | 0.01985 | 0.61512 | 0.01113 |
| 0.60598 | 0.00228 | 0.00276 | 0.59862 | 6.15E-04 |
| 0.65105 | 0.00995 | 0.01205 | 0.57691 | 0.00441 |
| 0.70256 | 0.00157 | 0.00195 | 0.53551 | 0.00146 |
| 0.73844 | 0.0068 | 0.00678 | 0.51233 | 0.00686 |
| 0.77309 | 0.01453 | 0.01453 | 0.48382 | 0.01515 |
| 0.83663 | 0.0021 | 0.00115 | 0.4015 | 0.00347 |
| 0.84998 | 0.00206 | 0.00204 | 0.38496 | 0.00336 |
| 0.90029 | 0.00391 | 0.00391 | 0.29984 | 0.00906 |
| 0.94266 | 0.00341 | 0.00339 | 0.19519 | 0.00966 |
| 0.98152 | 0.00147 | 0.00147 | 0.07624 | 0.00455 |

¶

**S4 Histidine Hygroscopicity**

**Fig S4.1**: Hygroscopicity of L-Histidine, (VWR Chemicals), open symbols, these CC-EDB experiments.

[Figure]

**Table S4.1:** Pure component refractive index ($n_{melt}$) is determined using molar refraction, assuming ideal mixing for calculation of the melt density ($\rho_{melt}$), from bulk data available in Cai et al. (2016). The variation of density as a function of the root of solute mass fraction (MFS $^{1/2}$ =x) is represented by polynomial fit parameters. *Upper* and *lower* refer to 95 % confidence limits for fits to experimental data, (Section 2.2 in manuscript).

| | $n_{melt}$ | $\rho_{melt}$/ $g.cm^{-3}$ | Polynomial fit ( $\rho_{sol} = a + b_1x + b_2x^2 + b_3x^3$) | | | |
|---|---|---|---|---|---|---|
| | | | a | $b_1$ | $b_2$ | $b_3$ |
| *Best* | 1.6892 | 1.5378 | 998.9 | 111.5 | -119.61 | 542.86 |
| *Upper* | 1.6914 | 1.5462 | 998.9 | 115.17 | -130.97 | 558.8 |
| *Lower* | 1.6871 | 1.5296 | 998.9 | 107.98 | -108.77 | 527.49 |

**Table S4.2:** Tabulated experimental data points shown in **Fig S4.1.**

| $a_w$ | error $a_w$ (+ve) | error $a_w$ (-ve) | MFS | error MFS |
|---|---|---|---|---|
| | | 293.15 K | | |
| 0.66801 | 0.00175 | 0.00214 | 0.64888 | 5.85836E-4 |
| 0.75174 | 0.00105 | 0.00131 | 0.61182 | 3.82887E-4 |
| 0.77265 | 0.00527 | 0.00661 | 0.59614 | 0.00177 |
| 0.83375 | 0.0064 | 0.00643 | 0.51198 | 0.01825 |
| 0.87281 | 0.00111 | 0.00101 | 0.48826 | 0.0027 |
| 0.9239 | 8.9548E-4 | 9.46372E-4 | 0.38721 | 0.00439 |
| 0.99296 | 6.37951E-4 | 6.3374E-4 | 0.03829 | 0.00295 |

**S5 L-Lysine Hygroscopicity**

**Fig S5.1**: Hygroscopicity of L-Lysine, (Sigma Aldrich, Purity $\geq$ 98 %), at 293.15 K. Open squares, these experiments; solid line, UNIFAC model.

[Figure]

**Table S5.1:** Pure component refractive index ($n_{melt}$) determined using molar refraction where the melt density ($\rho_{melt}$), is determined using a polynomial fit of density to the square root of MFS ($MFS^{1/2} = x$). Bulk values used are available in Cai et al. (2016). *Upper* and *lower* refer to 95 % confidence limits for fits to experimental data.

| | $n_{melt}$ | $\rho_{melt}$/ $g.cm^{-3}$ | Polynomial fit ( $\rho_{sol} = a + b_1x + b_2x^2 + b_3x^3$) | | | |
|---|---|---|---|---|---|---|
| | | | a | $b_1$ | $b_2$ | $b_3$ |
| *Best* | 1.5586 | 1.2362 | 998.2 | -15.22 | 309.14 | -56.02 |
| *Upper* | 1.5614 | 1.2418 | 998.2 | -4.29 | 271.92 | -23.99 |
| *Lower* | 1.5558 | 1.2306 | 998.2 | -25.93 | 346.35 | -88.05 |

**Table S5.2:** Tabulated experimental data points shown in **Fig S5.1.**

| $a_w$ | error $a_w$ (+ve) | error $a_w$ (-ve) | MFS | error MFS |
|---|---|---|---|---|
| 0.50605 | 0.00267 | 0.00316 | 0.65479 | 0.00157 |
| 0.52404 | 0.01187 | 0.01406 | 0.63621 | 0.00337 |
| 0.57666 | 0.02059 | 0.02439 | 0.60815 | 0.00948 |
| 0.58372 | 0.02793 | 0.03308 | 0.60867 | 0.01931 |
| 0.64049 | 0.00405 | 0.00494 | 0.58275 | 4.04084E-4 |
| 0.64559 | 0.00205 | 0.00251 | 0.57997 | 4.926E-4 |
| 0.67292 | 0.00999 | 0.0122 | 0.56365 | 0.00675 |
| 0.68839 | 0.0225 | 0.02735 | 0.54807 | 0.02742 |
| 0.71755 | 0.00885 | 0.01092 | 0.53328 | 0.00647 |
| 0.72732 | 0.00179 | 0.00223 | 0.53359 | 0.00179 |
| 0.75098 | 0.0056 | 0.00696 | 0.51408 | 0.00626 |
| 0.77291 | 0.00939 | 0.01164 | 0.49095 | 0.01194 |
| 0.79224 | 0.00736 | 0.00909 | 0.46681 | 0.01099 |
| 0.80926 | 0.01092 | 0.01352 | 0.45505 | 0.01535 |
| 0.82751 | 0.01292 | 0.01604 | 0.43489 | 0.02026 |
| 0.85916 | 0.00152 | 0.00197 | 0.41093 | 0.00309 |
| 0.87288 | 0.00143 | 0.00143 | 0.39407 | 0.00288 |
| 0.88688 | 0.00151 | 0.00151 | 0.3739 | 0.00294 |
| 0.90999 | 0.00294 | 0.00337 | 0.33998 | 0.00983 |
| 0.93683 | 3.20551E-4 | 3.24824E-4 | 0.27222 | 0.00154 |
| 0.94931 | 0.00162 | 0.00162 | 0.23179 | 0.00544 |
| 0.97255 | 0.00147 | 0.00147 | 0.14683 | 0.00623 |
| 0.99465 | 4.49456E-4 | 4.50168E-4 | 0.03174 | 0.0021 |
| 1.00277 | 0.00113 | 0.00149 | 0.02039 | 0.00198 |

**S6 L-Proline Hygroscopicity**

**Figure S6.1**: Hygroscopicity of L-Proline, (Acros Organics, Purity + 99 %), at 293.15 K. Open squares, these experiments.

[Figure]

**Table S6.1:** Pure component refractive index ($n_{melt}$) is determined using molar refraction, assuming ideal mixing for calculation of the melt density ($\rho_{melt}$), from bulk data available in Cai et al. (2016). The variation of density as a function of the root of solute mass fraction (MFS$^{1/2}$ =x) is represented by polynomial fit parameters. *Upper* and *lower* refer to 95 % confidence limits for fits to experimental data, (Section 2.2 in manuscript).

| | $n_{melt}$ | $\rho_{melt}/g.cm^{-3}$ | Polynomial fit ( $\rho_{sol} = a + b_1x + b_2x^2 + b_3x^3$) | | | |
| --- | --- | --- | --- | --- | --- | --- |
| | | | a | $b_1$ | $b_2$ | $b_3$ |
| *Best* | 1.5948 | 1.3866 | 999 | 55.7 | 39.01 | 291 |
| *Upper* | 1.5964 | 1.3945 | 999 | 58.13 | 32.96 | 302.44 |
| *Lower* | 1.5932 | 1.3788 | 999 | 53.36 | 44.73 | 279.93 |

**Table S6.2:** Tabulated experimental data points shown in **Fig S6.1.**

| $a_w$ | error $a_w$ (+ve) | error $a_w$ (-ve) | MFS | error MFS |
| --- | --- | --- | --- | --- |
| 0.52739 | 0.00183 | 0.00218 | 0.647 | 1.53E-04 |
| 0.59111 | 0.01061 | 0.01264 | 0.62414 | 0.00218 |
| 0.61213 | 0.00154 | 0.00186 | 0.6013 | 1.44E-04 |
| 0.64995 | 0.00905 | 0.01098 | 0.58549 | 0.00122 |
| 0.70619 | 9.52E-04 | 0.00118 | 0.54716 | 3.08E-04 |
| 0.73823 | 0.0057 | 0.00705 | 0.52349 | 0.00436 |
| 0.79982 | 7.99E-04 | 0.00101 | 0.46617 | 7.45E-04 |
| 0.80883 | 0.00112 | 0.00112 | 0.45742 | 1.45E-03 |
| 0.87515 | 1.30E-03 | 0.00103 | 0.36217 | 2.56E-03 |
| 0.91551 | 0.00145 | 0.00184 | 0.30701 | 0.00352 |
| 0.93455 | 9.07E-04 | 9.29E-04 | 0.24258 | 0.00279 |
| 0.99172 | 5.01E-04 | 5.60E-04 | 0.01734 | 0.0011 |

**S7 L-Threonine Hygroscopicity**

**Fig S7.1**: Hygroscopicity of L-Threonine, (Acros Organics, Purity 98 %), at 293.15 K. Open squares, these experiments; solid line, UNIFAC model.

[Figure]

**Table S7.1:** Pure component refractive index ($n_{melt}$) is determined using molar refraction, assuming ideal mixing for calculation of the melt density ($\rho_{melt}$), from bulk data available in Cai et al. (2016). The variation of density as a function of the root of solute mass fraction ($MFS^{1/2} = x$) is represented by polynomial fit parameters. *Upper* and *lower* refer to 95 % confidence limits for fits to experimental data, (Section 2.2 in manuscript).

| | $n_{melt}$ | $\rho_{melt}/\ g.cm^{-3}$ | Polynomial fit ( $\rho_{sol} = a + b_1x + b_2x^2 + b_3x^3$) | | | |
|---|---|---|---|---|---|---|
| | | | a | $b_1$ | $b_2$ | $b_3$ |
| *Best* | 1.6185 | 1.4977 | 999.4 | 94.57 | -68.14 | 468.44 |
| *Upper* | 1.6274 | 1.5403 | 999.4 | 112.31 | -121.99 | 546.4 |
| *Lower* | 1.6102 | 1.4575 | 999.4 | 79.24 | -23.63 | 399.69 |

**Table S7.2:** Tabulated experimental data points shown in **Fig S7.1.**

| $a_w$ | error $a_w$ (+ve) | error $a_w$ (-ve) | MFS | error MFS |
|---|---|---|---|---|
| 0.47807 | 0.00212 | 0.0025 | 0.70511 | 0.00173 |
| 0.53888 | 0.0052 | 0.0062 | 0.69319 | 0.00723 |
| 0.58237 | 0.01442 | 0.01711 | 0.67043 | 0.00714 |
| 0.60978 | 0.00127 | 0.00154 | 0.65941 | 3.88E-04 |
| 0.63867 | 0.01393 | 0.01689 | 0.65875 | 0.00707 |
| 0.68779 | 0.00158 | 0.00195 | 0.61529 | 0.00161 |
| 0.73352 | 0.0081 | 0.00812 | 0.59255 | 0.0041 |
| 0.75945 | 0.02291 | 0.02781 | 0.58815 | 0.04754 |
| 0.80118 | 0.00157 | 7.31E-04 | 0.51778 | 0.00135 |
| 0.86674 | 7.26E-04 | 4.84E-04 | 0.44784 | 3.66E-04 |
| 0.89045 | 0.00426 | 0.00419 | 0.39429 | 0.0104 |
| 0.93289 | 0.00438 | 0.00418 | 0.29104 | 0.01212 |
| 0.98064 | 0.00213 | 0.00214 | 0.10966 | 0.00862 |
| 0.99865 | 7.16E-04 | 4.45E-04 | 0.02317 | 0.00125 |

**S8 L-Valine Hygroscopicity**

**Figure S8.1**: Hygroscopicity of L-Valine, (Sigma Aldrich, Purity ≥ 98 %), at 293.15 K (blue Open symbols, these CK-EDB experiments; black filled circles, literature data (Kuramochi *et al.*); solid black line, UNIFAC model (293.15 K).

[Figure]

**Table S8.1:** Pure component refractive index ($n_{melt}$) is determined using molar refraction, assuming ideal mixing for calculation of the melt density ($\rho_{melt}$), from bulk data available in Cai et al. (2016). The variation of density as a function of the root of solute mass fraction ($MFS^{1/2} = x$) is represented by polynomial fit parameters. *Upper* and *lower* refer to 95 % confidence limits for fits to experimental data, (Section 2.2 in manuscript).

| | $n_{melt}$ | $\rho_{melt}/\ g.cm^{-3}$ | Polynomial fit ( $\rho_{sol} = a + b_1x + b_2x^2 + b_3x^3$) | | | |
| --- | --- | --- | --- | --- | --- | --- |
| | | | a | $b_1$ | $b_2$ | $b_3$ |
| *Best* | 1.5791 | 1.2824 | 998.77 | 28.73 | 94.37 | 159.64 |
| *Upper* | 1.58 | 1.2872 | 998.77 | 29.8 | 92.82 | 164.91 |
| *Lower* | 1.5781 | 1.2776 | 998.77 | 27.71 | 95.81 | 154.45 |

**Table S8.2:** Tabulated experimental data points shown in **Fig S8.1.**

| $a_w$ | error $a_w$ (+ve) | error $a_w$ (-ve) | MFS | error MFS |
| --- | --- | --- | --- | --- |
| | | 293.15 K | | |
| 0.92062 | 0.0027 | 0.00232 | 0.29295 | 0.00499 |
| 0.93004 | 2.26E-03 | 0.00195 | 0.26962 | 4.18E-03 |
| 0.93941 | 0.00173 | 0.00148 | 0.24396 | 0.00404 |
| 0.94943 | 0.00169 | 0.00145 | 0.21478 | 0.00403 |
| 0.9599 | 0.00138 | 0.00118 | 0.18125 | 0.00388 |
| 0.97008 | 0.00118 | 0.00101 | 0.14482 | 0.00345 |
| 0.98014 | 8.66E-04 | 7.41E-04 | 0.10314 | 2.94E-03 |
| 0.99117 | 5.34E-04 | 5.69E-04 | 0.04451 | 0.00201 |
| 0.99669 | 4.13E-04 | 4.57E-04 | 0.02205 | 8.65E-04 |

**S9 Citric Acid Hygroscopicity**

**Figure S9.1**: Hygroscopicity of Citric Acid, (Sigma Aldrich, Purity 99 %), at 293.15 K. Open squares, these EDB experiments; solid line, UNIFAC model.

[Figure]

**Table S9.1:** Pure component refractive index ($n_{melt}$) determined using molar refraction where the melt density ($\rho_{melt}$) is determined using a polynomial fit of density to the square root of MFS ($MFS^{1/2} = x$). Bulk values used are available in Cai et al. (2016). *Upper* and *lower* refer to 95 % confidence limits for fits to experimental data.

| | $n_{melt}$ | $\rho_{melt}$/ $g.cm^{-3}$ | Polynomial fit ( $\rho_{sol} = a + b_1x + b_2x^2 + b_3x^3$) | | | |
|---|---|---|---|---|---|---|
| | | | a | $b_1$ | $b_2$ | $b_3$ |
| *Best* | 1.5054 | 1.550 | 998.0 | 25.0 | 253.84 | 273.2 |
| *Upper* | 1.5071 | 1.5565 | 998.0 | 37.88 | 211.13 | 309.49 |
| *Lower* | 1.5037 | 1.5436 | 998.0 | 12.11 | 296.56 | 236.92 |

**Table S9.2:** Tabulated experimental data points shown in **Fig S9.1.**

| $a_w$ | error $a_w$ (+ve) | error $a_w$ (-ve) | MFS | error MFS |
|---|---|---|---|---|
| 0.53894 | 0.0024 | 0.00286 | 0.76226 | 0.00233 |
| 0.59688 | 0.01043 | 0.01244 | 0.73375 | 0.00875 |
| 0.62961 | 0.00223 | 0.00271 | 0.70793 | 7.62E-04 |
| 0.6837 | 0.00876 | 0.01065 | 0.67914 | 0.00409 |
| 0.72762 | 0.00123 | 0.00153 | 0.64368 | 0.00135 |
| 0.74592 | 0.00403 | 0.00404 | 0.63069 | 0.00342 |
| 0.80229 | 0.00504 | 0.0029 | 0.58246 | 0.00401 |
| 0.82734 | 0.00237 | 0.00196 | 0.56406 | 0.00314 |
| 0.88104 | 0.00107 | 0.0012 | 0.4682 | 0.00149 |
| 0.90968 | 0.00331 | 0.00327 | 0.41761 | 0.00738 |
| 0.95487 | 0.0028 | 0.00279 | 0.26562 | 0.01165 |
| 0.99255 | 6.21E-04 | 6.70E-04 | 0.03973 | 0.00355 |

**S10 L-Tartaric Acid Hygroscopicity**

**Figure S10.1**: Hygroscopicity of Tartaric Acid, (Sigma Aldrich, Purity $\geq$ 99.5 %), at 293.15 K. Open squares, these EDB experiments; solid line, UNIFAC model.

[Figure]

**Table S10.1:** Pure component refractive index ($n_{melt}$) determined using molar refraction where the melt density ($\rho_{melt}$) is determined using a polynomial fit of density to the square root of MFS (MFS$^{\frac{1}{2}}$ = x). Bulk values used are available in Cai et al. (2016). *Upper* and *lower* refer to 95 % confidence limits for fits to experimental data.

| | $n_{melt}$ | $\rho_{melt}$/ g.cm$^{-3}$ | Polynomial fit ( $\rho_{sol} = a + b_1x + b_2x^2 + b_3x^3$) | | | |
| --- | --- | --- | --- | --- | --- | --- |
| | | | a | $b_1$ | $b_2$ | $b_3$ |
| *Best* | 1.4992 | 1.6007 | 999 | 15.08 | 325.84 | 260.78 |
| *Upper* | 1.4996 | 1.6128 | 999 | 29.23 | 273.11 | 311.49 |
| *Lower* | 1.4936 | 1.5886 | 999 | 93.2 | 378.58 | 210.06 |

**Table S10.2:** Tabulated experimental data points shown in **Fig S10.1.**

| $a_w$ | error $a_w$ (+ve) | error $a_w$ (-ve) | MFS | error MFS |
| --- | --- | --- | --- | --- |
| 0.49764 | 0.00285 | 0.00337 | 0.74075 | 0.00145 |
| 0.59229 | 0.00275 | 0.00332 | 0.70005 | 0.00184 |
| 0.62457 | 0.00897 | 0.01082 | 0.68273 | 0.00361 |
| 0.67107 | 0.00165 | 0.00203 | 0.64893 | 7.95E-04 |
| 0.70799 | 0.00711 | 0.00826 | 0.6255 | 0.0076 |
| 0.75229 | 0.00853 | 0.01048 | 0.59046 | 0.00337 |
| 0.79666 | 0.00778 | 0.00946 | 0.56049 | 0.00992 |
| 0.84739 | 9.37E-04 | 5.51E-04 | 0.45906 | 0.00122 |
| 0.86463 | 0.00206 | 0.00206 | 0.44068 | 0.00269 |
| 0.91248 | 0.00302 | 0.00302 | 0.34362 | 0.00774 |
| 0.95599 | 0.00217 | 0.00216 | 0.20415 | 0.00789 |
| 0.98847 | 0.00104 | 0.00105 | 0.03363 | 0.00337 |

**S11 Oxalic Acid Hygroscopicity**

**Fig S11.1**: Hygroscopicity of Oxalic Acid, (Sigma Aldrich, Purity 98 %), at 293.15 K. Open squares, these experiments; solid line, UNIFAC model.

[Figure]

**Table S11.1:** Pure component refractive index ($n_{melt}$) is determined using molar refraction, assuming ideal mixing for calculation of the melt density ($\rho_{melt}$), from bulk data available in Cai et al. (2016). The variation of density as a function of the root of solute mass fraction ($MFS^{1/2} = x$) is represented by polynomial fit parameters. *Upper* and *lower* refer to 95 % confidence limits for fits to experimental data, (Section 2.2 in manuscript).

| | $n_{melt}$ | $\rho_{melt}/$ g.cm$^{-3}$ | Polynomial fit ( $\rho_{sol} = a + b_1x + b_2x^2 + b_3x^3 + b_4x^4 + b_5x^5 + b_6x^6$) | | | | | | |
|---|---|---|---|---|---|---|---|---|---|
| | | | a | $b_1$ | $b_2$ | $b_3$ | $b_4$ | $b_5$ | $b_6$ |
| *Best* | 1.5167 | 1.7237 | 998.4 | -14.98 | 636.47 | -1074.2 | 2603.92 | -2596.5 | 1170.54 |
| *Upper* | 1.5185 | 1.7403 | 998.4 | -16.27 | 660.48 | -1165.1 | 2811.06 | -2809 | 1260.66 |
| *Lower* | 1.5149 | 1.7073 | 998.4 | -13.78 | 613.65 | -989.39 | 2409.96 | -2397.5 | 1085.84 |

**Table S11.2:** Tabulated experimental data points shown in **Fig S11.1.**

| $a_w$ | error $a_w$ (+ve) | error $a_w$ (-ve) | MFS | error MFS |
|---|---|---|---|---|
| 0.75352 | 0.00146 | 0.00183 | 0.49497 | 0.00148 |
| 0.77652 | 0.00502 | 0.00629 | 0.47731 | 0.0077 |
| 0.79664 | 0.00652 | 0.00817 | 0.46116 | 0.00912 |
| 0.81614 | 0.00716 | 0.00896 | 0.44009 | 0.00613 |
| 0.83841 | 0.00803 | 0.01005 | 0.41068 | 0.00808 |
| 0.85938 | 0.0012 | 7.60E-04 | 0.36829 | 0.00113 |
| 0.87602 | 0.00199 | 0.00188 | 0.34275 | 0.00355 |
| 0.89702 | 0.00235 | 0.00215 | 0.30804 | 0.00596 |
| 0.92388 | 8.50E-04 | 0.00115 | 0.25275 | 0.00331 |
| 0.93784 | 0.00106 | 0.00106 | 0.21515 | 0.00299 |
| 0.9589 | 8.73E-04 | 8.75E-04 | 0.15314 | 0.00313 |
| 0.98012 | 5.72E-04 | 5.68E-04 | 0.07645 | 0.00227 |
| 0.99129 | 2.93E-04 | 3.54E-04 | 0.03567 | 8.98E-04 |

**S12 Malonic Acid Hygroscopicity**

**Figure S12.1**: Hygroscopicity of Malonic Acid, (Sigma Aldrich, Purity 98 %), at 293.15 K. Open squares, these experiments; solid line, UNIFAC model.

[Figure]

**Table S12.1:** Pure component refractive index determined using molar refraction where the melt density is determined using a polynomial fit of density to the square root of MFS ($MFS^{1/2} = x$). Bulk values used are available in Cai et al. (2016). *Upper* and *lower* refer to 95 % confidence limits for fits to experimental data.

| | $n_{melt}$ | $\rho_{melt}/g.cm^{-3}$ | Polynomial fit ( $\rho_{sol} = a + b_1x + b_2x^2 + b_3x^3$ ) | | | |
| | | | a | $b_1$ | $b_2$ | $b_3$ |
|---|---|---|---|---|---|---|
| *Best* | 1.4611 | 1.4558 | 997.2 | 13.47 | 262.36 | 182.76 |
| *Upper* | 1.4627 | 1.4612 | 997.2 | 20.7 | 235.91 | 207.37 |
| *Lower* | 1.4594 | 1.4504 | 997.2 | 6.24 | 288.82 | 158.15 |

**Table S12.2:** Tabulated experimental data points shown in **Fig S12.1.**

| $a_w$ | error $a_w$ (+ve) | error $a_w$ (-ve) | MFS | error MFS |
|---|---|---|---|---|
| 0.63613 | 4.14E-04 | 5.04E-04 | 0.6718 | 1.62E-04 |
| 0.64803 | 0.00253 | 0.00308 | 0.66275 | 0.00161 |
| 0.65776 | 0.00301 | 0.00367 | 0.65273 | 0.00197 |
| 0.66822 | 0.00457 | 0.00558 | 0.64421 | 0.00162 |
| 0.67795 | 0.00624 | 0.00761 | 0.64043 | 0.00324 |
| 0.68747 | 0.00679 | 0.00828 | 0.62992 | 0.00337 |
| 0.6994 | 0.0051 | 0.00625 | 0.63085 | 0.00413 |
| 0.7117 | 4.61E-04 | 5.72E-04 | 0.6176 | 2.85E-04 |
| 0.71572 | 0.00132 | 0.00164 | 0.61275 | 0.00135 |
| 0.72728 | 0.00371 | 0.00458 | 0.59849 | 0.00119 |
| 0.73786 | 0.00398 | 0.0049 | 0.59485 | 0.00362 |
| 0.74792 | 0.00441 | 0.00544 | 0.58075 | 0.00309 |
| 0.75777 | 0.00438 | 0.00539 | 0.57442 | 0.00518 |
| 0.76852 | 0.00437 | 0.00539 | 0.56217 | 0.00227 |
| 0.77901 | 0.00483 | 0.00595 | 0.55399 | 0.00568 |
| 0.78948 | 0.00564 | 0.00694 | 0.53743 | 0.0039 |
| 0.79831 | 0.00665 | 0.00816 | 0.53501 | 0.00406 |
| 0.80703 | 0.00469 | 0.00576 | 0.52388 | 0.00468 |
| 0.81779 | 0.00517 | 0.00637 | 0.50809 | 0.00307 |
| 0.82931 | 0.00495 | 0.0061 | 0.49855 | 0.00449 |
| 0.83997 | 0.00501 | 0.00616 | 0.48341 | 0.0058 |
| 0.85259 | 0.0013 | 8.20E-04 | 0.46721 | 9.32E-04 |
| 0.85596 | 0.00112 | 6.98E-04 | 0.46233 | 0.00114 |
| 0.86726 | 0.00223 | 0.00146 | 0.44257 | 0.00286 |
| 0.87898 | 0.00278 | 0.00183 | 0.4203 | 0.0033 |
| 0.8885 | 0.00291 | 0.00204 | 0.4002 | 0.0044 |
| 0.89906 | 0.00294 | 0.00205 | 0.37643 | 0.00371 |

| | | | | |
|---|---|---|---|---|
| 0.90919 | 0.00337 | 0.00233 | 0.35563 | 0.00559 |
| 0.9213 | 1.65E-04 | 2.03E-04 | 0.32987 | 3.33E-04 |
| 0.92743 | 2.61E-04 | 2.85E-04 | 0.31545 | 0.00122 |
| 0.93737 | 3.90E-04 | 4.01E-04 | 0.27835 | 0.00185 |
| 0.94793 | 4.69E-04 | 4.71E-04 | 0.24233 | 0.00205 |
| 0.95802 | 4.67E-04 | 4.69E-04 | 0.20227 | 0.00223 |
| 0.96932 | 0.0018 | 0.0011 | 0.14857 | 0.00427 |
| 0.97897 | 0.00144 | 8.79E-04 | 0.10171 | 0.00358 |
| 0.98599 | 0.00158 | 9.62E-04 | 0.07431 | 0.00205 |

**S13 Succinic Acid Hygroscopicity**

**Fig S13.1**: Hygroscopicity of Succinic Acid, (Sigma Aldrich, Purity 99 %), at 293.15 K. Open squares, these experiments; solid line, UNIFAC model.

[Figure]

**Table S13.1:** Pure component refractive index ($n_{melt}$) is determined using molar refraction, assuming ideal mixing for calculation of the melt density ($\rho_{melt}$), from bulk data available in Cai et al. (2016). The variation of density as a function of the root of solute mass fraction ($MFS^{1/2} = x$) is represented by polynomial fit parameters. *Upper* and *lower* refer to 95 % confidence limits for fits to experimental data, (Section 2.2 in manuscript).

| | $n_{melt}$ | $\rho_{melt}/$ g.cm$^{-3}$ | Polynomial fit ( $\rho_{sol} = a + b_1x + b_2x^2 + b_3x^3 + b_4x^4 + b_5x^5 + b_6x^6$) | | | | | | |
|---|---|---|---|---|---|---|---|---|---|
| | | | a | $b_1$ | $b_2$ | $b_3$ | $b_4$ | $b_5$ | $b_6$ |
| *Best* | 1.4928 | 1.4185 | 998.2 | -1.96 | 324.69 | -146.48 | 426.3 | -373.62 | 191.37 |
| *Upper* | 1.4935 | 1.4249 | 998.2 | -2.08 | 329.57 | -155.12 | 447.91 | -395.04 | 201.45 |
| *Lower* | 1.4920 | 1.4122 | 998.2 | -1.85 | 319.93 | -138.32 | 405.79 | -353.36 | 181.79 |

**Table S13.2:** Tabulated experimental data points shown in **Fig S13.1.**

| $a_w$ | error $a_w$ (+ve) | error $a_w$ (-ve) | MFS | error MFS |
|---|---|---|---|---|
| 0.69803 | 0.00919 | 0.00918 | 0.70299 | 0.00502 |
| 0.76653 | 0.0191 | 0.02355 | 0.64896 | 0.00963 |
| 0.83176 | 0.0191 | 0.02355 | 0.56672 | 0.01018 |
| 0.86728 | 0.00142 | 0.00142 | 0.47926 | 0.0025 |
| 0.91247 | 0.00444 | 0.0044 | 0.37868 | 0.01328 |
| 0.96915 | 0.00137 | 0.00136 | 0.15909 | 0.00637 |
| 0.99255 | 3.09E-04 | 3.16E-04 | 0.04733 | 0.00178 |

**S14 Glutaric Acid Hygroscopicity**

**Figure S14.1**: Hygroscopicity of Glutaric Acid, (Sigma Aldrich, Purity 99 %), at 293.15 K. Open squares, these experiments; solid line, UNIFAC model.

[Figure]

**Table S14.1:** Pure component refractive index ($n_{melt}$) determined using molar refraction where the melt density ($\rho_{melt}$) is determined using a polynomial fit of density to the square root of MFS (MFS$^{1/2}$ =x). Bulk values used are available in Cai et al. (2016). *Upper* and *lower* refer to 95 % confidence limits for fits to experimental data.

| | $n_{melt}$ | $\rho_{melt}/ g.cm^{-3}$ | Polynomial fit ( $\rho_{sol} = a + b_1x + b_2x^2 + b_3x^3$) | | | |
| | | | a | $b_1$ | $b_2$ | $b_3$ |
|---|---|---|---|---|---|---|
| *Best* | 1.4655 | 1.2745 | 997.5 | -1.56 | 238.79 | 39.75 |
| *Upper* | 1.4660 | 1.2760 | 997.5 | 0.401 | 231.59 | 46.55 |
| *Lower* | 1.4649 | 1.2729 | 997.5 | -3.53 | 245.98 | 32.95 |

**Table S14.2:** Tabulated experimental data points shown in **Fig S14.1.**

| $a_w$ | error $a_w$ (+ve) | error $a_w$ (-ve) | MFS | error MFS |
|---|---|---|---|---|
| 0.64988 | 4.93E-04 | 6.03E-04 | 0.80052 | 2.01E-04 |
| 0.66053 | 5.88E-04 | 7.21E-04 | 0.79339 | 2.71E-04 |
| 0.67122 | 0.00348 | 0.00426 | 0.78724 | 9.92E-04 |
| 0.68089 | 0.00458 | 0.00561 | 0.78134 | 0.00256 |
| 0.69112 | 0.00522 | 0.00639 | 0.76761 | 0.00226 |
| 0.70268 | 4.02E-04 | 4.98E-04 | 0.75519 | 1.75E-04 |
| 0.70969 | 0.0019 | 0.00235 | 0.75207 | 8.79E-04 |
| 0.72069 | 0.00357 | 0.00441 | 0.74499 | 0.00276 |
| 0.73156 | 0.0038 | 0.00469 | 0.74131 | 0.003 |
| 0.74152 | 0.00467 | 0.00575 | 0.72636 | 0.00217 |
| 0.75089 | 0.00485 | 0.00598 | 0.71793 | 0.00185 |
| 0.76119 | 0.00482 | 0.00594 | 0.70997 | 0.00323 |
| 0.77157 | 0.00535 | 0.00659 | 0.69846 | 0.00478 |
| 0.7818 | 0.00495 | 0.0061 | 0.68945 | 0.00241 |
| 0.79242 | 0.00502 | 0.00618 | 0.6775 | 0.00399 |
| 0.80236 | 5.99E-03 | 7.39E-03 | 0.6624 | 0.00435 |
| 0.81173 | 5.37E-03 | 6.62E-03 | 0.65327 | 0.00271 |
| 0.82228 | 0.00521 | 0.00642 | 0.64016 | 0.00336 |
| 0.83171 | 0.00538 | 0.00662 | 0.62617 | 0.00336 |
| 0.84134 | 0.00521 | 0.00642 | 0.60886 | 0.0028 |
| 0.852 | 0.00538 | 0.00663 | 0.59616 | 0.00364 |
| 0.86106 | 0.00556 | 0.00684 | 0.58136 | 0.00487 |
| 0.87055 | 0.00521 | 0.00642 | 0.56748 | 0.00361 |
| 0.88031 | 0.00617 | 0.00761 | 0.5545 | 0.00465 |
| 0.89058 | 0.00635 | 0.00785 | 0.5337 | 0.00679 |
| 0.90142 | 0.00628 | 0.00777 | 0.51244 | 0.00601 |
| 0.91173 | 0.00596 | 0.00737 | 0.48949 | 0.00523 |

| | | | | |
|---|---|---|---|---|
| 0.92543 | 1.50E-04 | 2.02E-04 | 0.45608 | 4.73E-04 |
| 0.92905 | 1.56E-04 | 1.58E-04 | 0.44732 | 6.46E-04 |
| 0.94053 | 3.86E-04 | 3.86E-04 | 0.40605 | 0.00233 |
| 0.95111 | 4.52E-04 | 4.52E-04 | 0.35279 | 0.00256 |
| 0.96227 | 3.42E-04 | 3.41E-04 | 0.28818 | 0.00275 |
| 0.97153 | 3.20E-04 | 3.20E-04 | 0.20634 | 0.00243 |
| 0.98066 | 8.31E-04 | 8.09E-04 | 0.12936 | 0.00558 |

**S15 Adipic Acid Hygroscopicity**

**Figure S15.1**: Hygroscopicity of Adipic Acid, (Sigma Aldrich, Purity 99 %), at 293.15 K. Open squares, these experiments; solid line, UNIFAC model.

[Figure]

**Table S15.1:** Pure component refractive index ($n_{melt}$) is determined using molar refraction, assuming ideal mixing for calculation of the melt density ($\rho_{melt}$), from bulk data available in Cai et al. (2016). The variation of density as a function of the root of solute mass fraction ($MFS^{1/2} = x$) is represented by polynomial fit parameters. *Upper* and *lower* refer to 95 % confidence limits for fits to experimental data, (Section 2.2 in manuscript).

| | $n_{melt}$ | $\rho_{melt}/$ g.cm$^{-3}$ | Polynomial fit ( $\rho_{sol} = a + b_1x + b_2x^2 + b_3x^3 + b_4x^4 + b_5x^5 + b_6x^6$) | | | | | | |
|---|---|---|---|---|---|---|---|---|---|
| | | | a | $b_1$ | $b_2$ | $b_3$ | $b_4$ | $b_5$ | $b_6$ |
| *Best* | 1.5052 | 1.2897 | 998.2 | -0.483 | 232.81 | -36.78 | 137.06 | -96.59 | 55.48 |
| *Upper* | 1.5093 | 1.3192 | 998.2 | -0.705 | 253.36 | -53.41 | 183.61 | -139.01 | 77.14 |
| *Lower* | 1.5012 | 1.2614 | 998.2 | -0.323 | 213.1 | -24.73 | 101.55 | -65.53 | 39.14 |

**Table S15.2:** Tabulated experimental data points shown in **Fig S15.1.**

| $a_w$ | error $a_w$ (+ve) | error $a_w$ (-ve) | MFS | error MFS |
|---|---|---|---|---|
| 0.95373 | 3.22E-04 | 6.26E-04 | 0.39071 | 0.00391 |
| 0.95843 | 8.35E-04 | 0.00118 | 0.36348 | 0.00812 |
| 0.9634 | 0.0011 | 0.00133 | 0.3272 | 0.00935 |
| 0.96865 | 0.00107 | 0.00138 | 0.28685 | 0.01043 |
| 0.97365 | 9.42E-04 | 0.00127 | 0.24062 | 0.01007 |
| 0.97863 | 9.10E-04 | 0.00114 | 0.1917 | 0.00876 |
| 0.98405 | 5.88E-04 | 8.82E-04 | 0.13977 | 0.00621 |
| 0.98877 | 3.13E-04 | 4.91E-04 | 0.09086 | 0.0027 |
| 0.99423 | 1.80E-04 | 3.02E-04 | 0.04898 | 0.00153 |
| 0.99692 | 1.66E-04 | 3.62E-04 | 0.02978 | 7.74E-04 |

**S16 Pimelic Acid Hygroscopicity**

**Fig S16.1**: Hygroscopicity of Pimelic Acid, (Sigma Aldrich, Purity 99 %), at 293.15 K. Open squares, these experiments; solid line, UNIFAC model.

[Figure]

**Table S16.1:** Pure component refractive index ($n_{melt}$) is determined using molar refraction, assuming ideal mixing for calculation of the melt density ($\rho_{melt}$), from bulk data available in Cai et al. (2016). The variation of density as a function of the root of solute mass fraction ($MFS^{1/2} = x$) is represented by polynomial fit parameters. *Upper* and *lower* refer to 95 % confidence limits for fits to experimental data, (Section 2.2 in manuscript).

| | $n_{melt}$ | $\rho_{melt}/$ g.cm$^{-3}$ | Polynomial fit ( $\rho_{sol} = a + b_1x + b_2x^2 + b_3x^3 + b_4x^4 + b_5x^5 + b_6x^6$) | | | | | | |
|---|---|---|---|---|---|---|---|---|---|
| | | | a | $b_1$ | $b_2$ | $b_3$ | $b_4$ | $b_5$ | $b_6$ |
| *Best* | 1.4917 | 1.2262 | 998.5 | -0.184 | 188.18 | -14.19 | 67.86 | -37.91 | 23.94 |
| *Upper* | 1.4940 | 1.2435 | 998.5 | -0.246 | 200.41 | -18.89 | 83.16 | -50.18 | 30.74 |
| *Lower* | 1.4894 | 1.2095 | 998.5 | -0.136 | 176.23 | -10.52 | 55.25 | -28.29 | 18.47 |

**Table S16.2:** Tabulated experimental data points shown in **Fig S16.1.**

| $a_w$ | error $a_w$ (+ve) | error $a_w$ (-ve) | MFS | error MFS |
|---|---|---|---|---|
| 0.84466 | 0.00317 | 0.00251 | 0.73296 | 0.0087 |
| 0.87279 | 0.00711 | 0.00413 | 0.70863 | 0.02048 |
| 0.88863 | 0.00919 | 0.00916 | 0.67508 | 0.03342 |
| 0.90517 | 0.00585 | 0.00361 | 0.65697 | 0.02274 |
| 0.92985 | 0.00504 | 0.00334 | 0.57632 | 0.01441 |
| 0.9503 | 0.00434 | 0.00304 | 0.48806 | 0.01436 |
| 0.97207 | 0.0019 | 0.00139 | 0.29782 | 0.01787 |
| 0.99347 | 2.49E-04 | 3.32E-04 | 0.06002 | 0.00268 |

**S17 Methyl Malonic Acid Hygroscopicity**

**Fig S17.1**: Hygroscopicity of methyl malonic acid, (Sigma Aldrich, Purity 99 %), at 293.15 K. Open squares, these experiments; solid line, UNIFAC model.

[Figure]

**Table S17.1:** Pure component refractive index ($n_{melt}$) is determined using molar refraction, assuming ideal mixing for calculation of the melt density ($\rho_{melt}$), from bulk data available in Cai et al. (2016). The variation of density as a function of the root of solute mass fraction ($MFS^{1/2} = x$) is represented by polynomial fit parameters. *Upper* and *lower* refer to 95 % confidence limits for fits to experimental data, (Section 2.2 in manuscript).

| | $n_{melt}$ | $\rho_{melt}/$ g.cm$^{-3}$ | Polynomial fit ( $\rho_{sol} = a + b_1x + b_2x^2 + b_3x^3 + b_4x^4 + b_5x^5 + b_6x^6$) | | | | | | |
|---|---|---|---|---|---|---|---|---|---|
| | | | a | $b_1$ | $b_2$ | $b_3$ | $b_4$ | $b_5$ | $b_6$ |
| *Best* | 1.4817 | 1.3876 | 998.8 | -1.45 | 301.28 | -108.73 | 330.65 | -279.56 | 146.61 |
| *Upper* | 1.4819 | 1.3902 | 998.8 | -1.49 | 303.18 | -111.53 | 337.82 | -286.56 | 149.98 |
| *Lower* | 1.4815 | 1.3851 | 998.8 | -1.42 | 299.45 | -106.09 | 323.86 | -272.94 | 143.43 |

**Table S17.2:** Tabulated experimental data points shown in **Fig S17.1.**

| $a_w$ | error $a_w$ (+ve) | error $a_w$ (-ve) | MFS | error MFS |
|---|---|---|---|---|
| 0.71493 | 0.002 | 0.00248 | 0.62219 | 0.00155 |
| 0.75141 | 0.00657 | 0.00657 | 0.59428 | 0.00609 |
| 0.78527 | 0.0084 | 0.0084 | 0.562 | 0.00836 |
| 0.81777 | 0.004 | 0.00245 | 0.52434 | 0.00364 |
| 0.84355 | 0.00409 | 0.00369 | 0.49609 | 0.00573 |
| 0.875 | 0.00438 | 0.00401 | 0.44143 | 0.00784 |
| 0.90462 | 0.00402 | 0.00333 | 0.3774 | 0.00875 |
| 0.93201 | 0.00335 | 0.00317 | 0.29413 | 0.01184 |
| 0.96865 | 8.29E-04 | 8.90E-04 | 0.16472 | 0.0041 |
| 0.98911 | 4.09E-04 | 4.11E-04 | 0.05691 | 0.00203 |

**S18 Methyl Succinic Acid Hygroscopicity**

**Figure S18.1**: Hygroscopicity of methyl succinic acid, (Sigma Aldrich, Purity 99 %), at 293.15 K. Open squares, these experiments; solid line, UNIFAC model.

[Figure]

**Table S18.1:** Pure component refractive index ($n_{melt}$) is determined using molar refraction, assuming ideal mixing for calculation of the melt density ($\rho_{melt}$), from bulk data available in Cai et al. (2016). The variation of density as a function of the root of solute mass fraction ($MFS^{1/2} = x$) is represented by polynomial fit parameters. *Upper* and *lower* refer to 95 % confidence limits for fits to experimental data, (Section 2.2 in manuscript).

| | $n_{melt}$ | $\rho_{melt}/$ $g.cm^{-3}$ | Polynomial fit ( $\rho_{sol} = a + b_1x + b_2x^2 + b_3x^3 + b_4x^4 + b_5x^5 + b_6x^6$) | | | | | | |
|---|---|---|---|---|---|---|---|---|---|
| | | | a | $b_1$ | $b_2$ | $b_3$ | $b_4$ | $b_5$ | $b_6$ |
| *Best* | 1.4779 | 1.3035 | 998.2 | -0.572 | 242.3 | -43.51 | 156.55 | -114.16 | 64.69 |
| *Upper* | 1.4784 | 1.3090 | 998.2 | -0.614 | 246.13 | -46.62 | 165.26 | -122.12 | 68.76 |
| *Lower* | 1.4774 | 1.2980 | 998.2 | -0.533 | 238.48 | -40.56 | 148.19 | -106.58 | 60.79 |

**Table S18.2:** Tabulated experimental data points shown in **Fig S18.1.**

| $a_w$ | error $a_w$ (+ve) | error $a_w$ (-ve) | MFS | error MFS |
|---|---|---|---|---|
| 0.66772 | 0.00345 | 0.00424 | 0.76125 | 0.00296 |
| 0.71234 | 0.0134 | 0.0134 | 0.72476 | 0.01132 |
| 0.7636 | 0.01237 | 0.01517 | 0.67596 | 0.00785 |
| 0.80135 | 0.00451 | 0.00326 | 0.65118 | 0.00447 |
| 0.83951 | 0.00575 | 0.00629 | 0.59855 | 0.01151 |
| 0.87778 | 0.00688 | 0.00657 | 0.52839 | 0.01469 |
| 0.92249 | 0.00567 | 0.00567 | 0.40343 | 0.01891 |
| 0.96705 | 0.00282 | 0.00249 | 0.19484 | 0.01368 |
| 0.99344 | 3.28E-04 | 3.47E-04 | 0.03075 | 0.00168 |

**S19 Binary Aqueous Diethylmalonic Acid - Hygroscopicity**

**Fig S19.1**: Hygroscopicity of diethylmalonic acid, (Sigma Aldrich, Purity 98 %), at 293.15 K. Open squares, these experiments; solid line, UNIFAC model.

[Figure]

**Table S19.1:** Pure component refractive index ($n_{melt}$) is determined using molar refraction, assuming ideal mixing for calculation of the melt density ($\rho_{melt}$), from bulk data available in Cai et al. (2016). The variation of density as a function of the root of solute mass fraction ($MFS^{1/2} = x$) is represented by polynomial fit parameters. *Upper* and *lower* refer to 95 % confidence limits for fits to experimental data, (Section 2.2 in manuscript).

| | $n_{melt}$ | $\rho_{melt}/$ g.cm$^{-3}$ | Polynomial fit ( $\rho_{sol} = a + b_1x + b_2x^2 + b_3x^3 + b_4x^4 + b_5x^5 + b_6x^6$ ) | | | | | | |
|---|---|---|---|---|---|---|---|---|---|
| | | | a | $b_1$ | $b_2$ | $b_3$ | $b_4$ | $b_5$ | $b_6$ |
| *Best* | 1.4854 | 1.2184 | 998.2 | -0.161 | 182.82 | -12.45 | 61.98 | -33.36 | 21.37 |
| *Upper* | 1.4858 | 1.2219 | 998.2 | -0.172 | 185.32 | -13.25 | 64.69 | -35.44 | 22.55 |
| *Lower* | 1.4850 | 1.2149 | 998.2 | -0.151 | 180.32 | -11.69 | 59.36 | -31.37 | 20.24 |

**Table S19.2:** Tabulated experimental data points shown in **Figure S19.1.**

| $a_w$ | error $a_w$ (+ve) | error $a_w$ (-ve) | MFS | error MFS |
|---|---|---|---|---|
| 0.68895 | 0.00441 | 0.00543 | 0.79565 | 0.00315 |
| 0.70762 | 0.01 | 0.01233 | 0.78448 | 0.00548 |
| 0.7737 | 0.01156 | 0.01425 | 0.7484 | 0.00901 |
| 0.83916 | 0.01287 | 0.00773 | 0.70902 | 0.01617 |
| 0.84654 | 0.00329 | 0.00246 | 0.70885 | 0.00435 |
| 0.86832 | 0.00637 | 0.0062 | 0.68324 | 0.01847 |
| 0.88499 | 0.00646 | 0.00418 | 0.65203 | 0.0186 |
| 0.90928 | 0.00665 | 0.00391 | 0.62123 | 0.00847 |
| 0.93317 | 0.00665 | 0.00374 | 0.56028 | 0.00907 |
| 0.95177 | 0.00586 | 0.00329 | 0.48861 | 0.01646 |
| 0.97321 | 0.00199 | 0.00152 | 0.28968 | 0.01912 |
| 0.99422 | 3.23E-04 | 3.66E-04 | 0.02697 | 0.00157 |

**S20 2,2-Dimethyl Glutaric Acid Hygroscopicity**

**Fig S20.1**: Hygroscopicity of 2,2-dimethyl glutaric acid, (Sigma Aldrich, Purity > 98 %), at 293.15 K. Open squares, these experiments; solid line, UNIFAC model.

[Figure]

**Table S20.1:** Pure component refractive index ($n_{melt}$) is determined using molar refraction, assuming ideal mixing for calculation of the melt density ($\rho_{melt}$), from bulk data available in Cai et al. (2016). The variation of density as a function of the root of solute mass fraction ($MFS^{1/2} =x$) is represented by polynomial fit parameters. *Upper* and *lower* refer to 95 % confidence limits for fits to experimental data, (Section 2.2 in manuscript).

| | $n_{melt}$ | $\rho_{melt}/$ $g.cm^{-3}$ | Polynomial fit ( $\rho_{sol} = a + b_1x + b_2x^2 + b_3x^3 + b_4x^4 + b_5x^5 + b_6x^6$) | | | | | | |
|---|---|---|---|---|---|---|---|---|---|
| | | | a | $b_1$ | $b_2$ | $b_3$ | $b_4$ | $b_5$ | $b_6$ |
| *Best* | 1.4881 | 1.2225 | 998.2 | -0.174 | 185.75 | -13.39 | 65.16 | -35.81 | 22.76 |
| *Upper* | 1.4884 | 1.2248 | 998.2 | -0.181 | 187.39 | -13.93 | 67 | -37.24 | 23.57 |
| *Lower* | 1.4878 | 1.2201 | 998.2 | -0.166 | 184.04 | -12.83 | 63.28 | -34.36 | 21.94 |

**Table S19.2:** Tabulated experimental data points shown in **Fig S20.1**.

| $a_w$ | error $a_w$ (+ve) | error $a_w$ (-ve) | MFS | error MFS |
|---|---|---|---|---|
| 0.66522 | 0.00707 | 0.00713 | 0.87406 | 0.00722 |
| 0.71105 | 0.00493 | 0.00494 | 0.8677 | 0.00654 |
| 0.74996 | 0.00758 | 0.00758 | 0.83334 | 0.01058 |
| 0.79488 | 0.01337 | 0.01338 | 0.80256 | 0.01126 |
| 0.84249 | 0.00573 | 0.00389 | 0.76365 | 0.00522 |
| 0.86987 | 0.00563 | 0.00574 | 0.73768 | 0.00728 |
| 0.91262 | 0.00592 | 0.00605 | 0.65854 | 0.01692 |
| 0.95695 | 0.00508 | 0.00491 | 0.48805 | 0.02723 |
| 0.99362 | 3.59E-04 | 3.74E-04 | 0.05685 | 0.00348 |

**S21 2,2-Dimethyl Succinic Acid Hygroscopicity**

**Fig S21.1**: Hygroscopicity of 2,2-dimethyl succinic acid, (Sigma Aldrich, Purity 99 %), at 293.15 K. Open squares, these experiments; solid line, UNIFAC model.

[Figure]

**Table S21.1:** Pure component refractive index ($n_{melt}$) is determined using molar refraction, assuming ideal mixing for calculation of the melt density ($\rho_{melt}$), from bulk data available in Cai et al. (2016). The variation of density as a function of the root of solute mass fraction ($MFS^{1/2} = x$) is represented by polynomial fit parameters. *Upper* and *lower* refer to 95 % confidence limits for fits to experimental data, (Section 2.2 in manuscript).

| | $n_{melt}$ | $\rho_{melt}/$ g.cm$^{-3}$ | Polynomial fit ( $\rho_{sol} = a + b_1x + b_2x^2 + b_3x^3 + b_4x^4 + b_5x^5 + b_6x^6$) | | | | | | |
|---|---|---|---|---|---|---|---|---|---|
| | | | a | $b_1$ | $b_2$ | $b_3$ | $b_4$ | $b_5$ | $b_6$ |
| *Best* | 1.4889 | 1.2710 | 997.9 | -0.382 | 220.13 | -29.13 | 114.09 | -76.29 | 44.68 |
| *Upper* | 1.4897 | 1.2776 | 997.9 | -0.419 | 224.73 | -31.96 | 122.4 | -83.53 | 48.48 |
| *Lower* | 1.4880 | 1.2644 | 997.9 | -0.347 | 215.51 | -26.48 | 106.23 | -69.5 | 41.09 |

**Table S21.2:** Tabulated experimental data points shown in **Fig S21.1**

| $a_w$ | error $a_w$ (+ve) | error $a_w$ (-ve) | MFS | error MFS |
|---|---|---|---|---|
| 0.6713 | 0.00663 | 0.00663 | 0.78921 | 0.00655 |
| 0.73389 | 0.02256 | 0.02256 | 0.74829 | 0.01579 |
| 0.77076 | 0.00564 | 0.00705 | 0.73908 | 0.00579 |
| 0.84308 | 0.00747 | 0.00776 | 0.67413 | 0.01818 |
| 0.88089 | 0.00536 | 0.00529 | 0.60846 | 0.01212 |
| 0.9367 | 0.00425 | 0.00424 | 0.41751 | 0.01893 |
| 0.99244 | 4.24E-04 | 5.93E-04 | 0.05911 | 0.00313 |

**S22 2-Methyl Glutaric Acid Hygroscopicity**

**Fig S22.1**: Hygroscopicity of 2-methyl glutaric acid, (Sigma Aldrich, Purity 98 %), at 293.15 K. Open squares, these experiments; solid line, UNIFAC model.

[Figure]

**Table S22.1:** Pure component refractive index ($n_{melt}$) is determined using molar refraction, assuming ideal mixing for calculation of the melt density ($\rho_{melt}$), from bulk data available in Cai et al. (2016). The variation of density as a function of the root of solute mass fraction ($MFS^{1/2} = x$) is represented by polynomial fit parameters. *Upper* and *lower* refer to 95 % confidence limits for fits to experimental data, (Section 2.2 in manuscript).

| | $n_{melt}$ | $\rho_{melt}/$ g.cm$^{-3}$ | Polynomial fit ( $\rho_{sol} = a + b_1x + b_2x^2 + b_3x^3 + b_4x^4 + b_5x^5 + b_6x^6$ ) | | | | | | |
|---|---|---|---|---|---|---|---|---|---|
| | | | a | $b_1$ | $b_2$ | $b_3$ | $b_4$ | $b_5$ | $b_6$ |
| *Best* | 1.4866 | 1.2585 | 997.6 | -0.319 | 211.59 | -24.4 | 99.95 | -64.16 | 38.24 |
| *Upper* | 1.4873 | 1.2648 | 997.6 | -0.350 | 216 | -26.78 | 107.1 | -70.26 | 41.49 |
| *Lower* | 1.4858 | 1.2522 | 997.6 | -0.290 | 207.17 | -22.18 | 93.17 | -58.44 | 35.17 |

**Table S22.2:** Tabulated experimental data points shown in **Fig S22.1**

| $a_w$ | error $a_w$ (+ve) | error $a_w$ (-ve) | MFS | error MFS |
|---|---|---|---|---|
| 0.68925 | 0.00271 | 0.00334 | 0.80479 | 0.00208 |
| 0.72204 | 0.01005 | 0.01239 | 0.78383 | 0.00857 |
| 0.76123 | 0.01296 | 0.01422 | 0.75567 | 0.01704 |
| 0.78959 | 0.02339 | 0.02377 | 0.73478 | 0.02713 |
| 0.82836 | 0.01185 | 0.00726 | 0.70077 | 0.02018 |
| 0.84699 | 0.00634 | 0.00601 | 0.68658 | 0.01104 |
| 0.8785 | 0.00611 | 0.00622 | 0.63205 | 0.01527 |
| 0.91076 | 0.00612 | 0.00583 | 0.54437 | 0.02194 |
| 0.94004 | 0.00438 | 0.00438 | 0.4312 | 0.02071 |
| 0.98128 | 4.79E-04 | 0.0012 | 0.14884 | 0.0113 |
| 0.99285 | 2.24E-04 | 2.25E-04 | 0.02928 | 0.00106 |

**S23 3-Methyl Adipic Acid Hygroscopicity**

**Fig S23.1**: Hygroscopicity of 3-methyl adipic acid, (Sigma Aldrich, Purity 99 %), at 293.15 K. Open squares, these experiments; solid line, UNIFAC model.

[Figure]

**Table S23.1:** Pure component refractive index ($n_{melt}$) is determined using molar refraction, assuming ideal mixing for calculation of the melt density ($\rho_{melt}$), from bulk data available in Cai et al. (2016). The variation of density as a function of the root of solute mass fraction ($MFS^{1/2} = x$) is represented by polynomial fit parameters. *Upper* and *lower* refer to 95 % confidence limits for fits to experimental data, (Section 2.2 in manuscript).

| | $n_{melt}$ | $\rho_{melt}/$ g.cm$^{-3}$ | Polynomial fit ( $\rho_{sol} = a + b_1x + b_2x^2 + b_3x^3 + b_4x^4 + b_5x^5 + b_6x^6$) | | | | | | |
|---|---|---|---|---|---|---|---|---|---|
| | | | a | $b_1$ | $b_2$ | $b_3$ | $b_4$ | $b_5$ | $b_6$ |
| *Best* | 1.4865 | 1.2141 | 999.0 | -0.147 | 179.19 | -11.33 | 58.11 | -30.42 | 19.69 |
| *Upper* | 1.4878 | 1.2243 | 999.0 | -0.176 | 186.48 | -13.59 | 65.86 | -36.34 | 23.06 |
| *Lower* | 1.4852 | 1.2041 | 999.0 | -0.121 | 171.99 | -9.4 | 51.21 | -25.33 | 16.75 |

**Table S23.2:** Tabulated experimental data points shown in **Fig S23.1.**

| $a_w$ | error $a_w$ (+ve) | error $a_w$ (-ve) | MFS | error MFS |
|---|---|---|---|---|
| 0.71902 | 0.00897 | 0.00897 | 0.80154 | 0.00624 |
| 0.78015 | 0.03348 | 0.03347 | 0.7615 | 0.02149 |
| 0.82646 | 0.00615 | 0.00574 | 0.73848 | 0.00556 |
| 0.88266 | 0.00886 | 0.00907 | 0.67532 | 0.02097 |
| 0.92986 | 0.00748 | 0.00771 | 0.54185 | 0.02748 |
| 0.993 | 2.61E-04 | 3.72E-04 | 0.06527 | 0.00354 |

**S24 3-Methyl Glutaric Acid Hygroscopicity**

**Fig S24.1**: Hygroscopicity of 3-methyl glutaric acid, (Sigma Aldrich, Purity 99 %), at 293.15 K. Open squares, these experiments; solid line, UNIFAC model.

[Figure]

**Table SI.24.1:** Pure component refractive index ($n_{melt}$) is determined using molar refraction, assuming ideal mixing for calculation of the melt density ($\rho_{melt}$), from bulk data available in Cai et al. (2016). The variation of density as a function of the root of solute mass fraction (MFS$^{1/2}$ =x) is represented by polynomial fit parameters. *Upper* and *lower* refer to 95 % confidence limits for fits to experimental data, (Section 2.2 in manuscript).

| | $n_{melt}$ | $\rho_{melt}/$ g.cm$^{-3}$ | Polynomial fit ( $\rho_{sol} = a + b_1x + b_2x^2 + b_3x^3 + b_4x^4 + b_5x^5 + b_6x^6$) | | | | | | |
| --- | --- | --- | --- | --- | --- | --- | --- | --- | --- |
| | | | a | $b_1$ | $b_2$ | $b_3$ | $b_4$ | $b_5$ | $b_6$ |
| *Best* | 1.4819 | 1.2498 | 997.9 | -0.277 | 205.29 | -21.26 | 90.32 | -56.07 | 33.89 |
| *Upper* | 1.4822 | 1.2531 | 997.9 | -0.292 | 207.6 | -22.37 | 93.74 | -58.92 | 35.43 |
| *Lower* | 1.4816 | 1.2466 | 997.9 | -0.264 | 203.04 | -20.22 | 87.1 | -53.39 | 32.44 |

**Table S24.2:** Tabulated experimental data points shown in **Fig S24.1.**

| $a_w$ | error $a_w$ (+ve) | error $a_w$ (-ve) | MFS | error MFS |
| --- | --- | --- | --- | --- |
| 0.69173 | 0.00299 | 0.00334 | 0.79013 | 0.0038 |
| 0.74649 | 0.00642 | 0.00683 | 0.76025 | 0.00932 |
| 0.81013 | 0.01887 | 0.01884 | 0.70959 | 0.02367 |
| 0.86283 | 0.00343 | 0.00213 | 0.63276 | 0.00618 |
| 0.89131 | 0.00283 | 0.00283 | 0.58884 | 0.00675 |
| 0.95411 | 0.00246 | 0.00245 | 0.3472 | 0.01394 |
| 0.98567 | 6.06E-04 | 6.09E-04 | 0.10123 | 0.00477 |

**S25 3, 3-Dimethyl Glutaric Acid Hygroscopicity**

**Fig S25.1**: Hygroscopicity of 3, 3-dimethyl glutaric acid, (Sigma Aldrich, Purity 98 %), at 293.15 K. Open squares, these experiments; solid line, UNIFAC model.

[Figure]

**Table S25.1:** Pure component refractive index ($n_{melt}$) is determined using molar refraction, assuming ideal mixing for calculation of the melt density ($\rho_{melt}$), from bulk data available in Cai et al. (2016). The variation of density as a function of the root of solute mass fraction ($MFS^{1/2} = x$) is represented by polynomial fit parameters. *Upper* and *lower* refer to 95 % confidence limits for fits to experimental data, (Section 2.2 in manuscript).

| | $n_{melt}$ | $\rho_{melt}/$ g.cm$^{-3}$ | Polynomial fit ( $\rho_{sol} = a + b_1x + b_2x^2 + b_3x^3 + b_4x^4 + b_5x^5 + b_6x^6$) | | | | | | |
| --- | --- | --- | --- | --- | --- | --- | --- | --- | --- |
| | | | a | $b_1$ | $b_2$ | $b_3$ | $b_4$ | $b_5$ | $b_6$ |
| *Best* | 1.4903 | 1.2206 | 998.3 | -0.167 | 184.33 | -12.92 | 63.58 | -34.59 | 22.07 |
| *Upper* | 1.4906 | 1.2231 | 998.3 | -0.175 | 186.11 | -13.5 | 65.55 | -36.11 | 22.93 |
| *Lower* | 1.4900 | 1.2182 | 998.3 | -0.160 | 182.61 | -12.38 | 61.74 | -33.18 | 21.27 |

**Table S25.2:** Tabulated experimental data points shown in **Fig S25.1.**

| $a_w$ | error $a_w$ (+ve) | error $a_w$ (-ve) | MFS | error MFS |
| --- | --- | --- | --- | --- |
| 0.71132 | 0.00345 | 0.00345 | 0.85176 | 0.00384 |
| 0.76078 | 0.006 | 0.00743 | 0.80912 | 0.00721 |
| 0.79151 | 0.01941 | 0.01942 | 0.79788 | 0.01562 |
| 0.83444 | 0.00416 | 0.00451 | 0.75169 | 0.00421 |
| 0.87055 | 0.00543 | 0.00565 | 0.71882 | 0.0105 |
| 0.91582 | 0.00545 | 0.00564 | 0.61641 | 0.02163 |
| 0.96018 | 0.00389 | 0.00389 | 0.39161 | 0.02576 |
| 0.99443 | 2.18E-04 | 2.83E-04 | 0.04485 | 0.00225 |

**S26. PEG3 Hygroscopicity**

**Fig S26.1**: Hygroscopicity of PEG3, at 293.15 K. Open squares, these experiments; solid line, UNIFAC model.

[Figure]

**Table S26.1:** Measured values of pure component melt density ($\rho_{melt}$) and refractive index ($n_{melt}$) (PEG3 is liquid), presented with parameterisation for solution measurements of density where x is the square root of MFS ($MFS^{1/2} = x$). *Upper* and *lower* refer to 95 % confidence limits for fits to experimental data. Upper and lower limit on refractive index and density are determined by the error in the refractometer and by the densitometer respectively.

| | $n_{melt}$ | $\rho_{melt}$/ g.cm$^{-3}$ | Polynomial fit ( $\rho_{sol} = a + b_1x + b_2x^2 + b_3x^3$) | | | |
|---|---|---|---|---|---|---|
| | | | a | $b_1$ | $b_2$ | $b_3$ |
| *Best* | 1.4551 | 1.109 | 999.97 | -75.75 | 431.63 | -246.73 |
| *Upper* | 1.4552 | 1.122 | 999.97 | -0.198 | 268.11 | -144.15 |
| *Lower* | 1.4550 | 1.096 | 999.97 | -151.31 | 595.15 | -349.31 |

**Table S26.2:** Tabulated experimental data points shown in **Fig S26.1.**

| $a_w$ | error $a_w$ (+ve) | error $a_w$ (-ve) | MFS | error MFS |
|---|---|---|---|---|
| 0.524 | 0.0024 | 0.00286 | 0.83232 | 0.00127 |
| 0.61806 | 0.02008 | 0.02389 | 0.77269 | 0.0098 |
| 0.65597 | 0.00198 | 0.00242 | 0.72923 | 0.00152 |
| 0.69291 | 0.00856 | 0.00856 | 0.69867 | 0.0088 |
| 0.75489 | 8.16E-03 | 0.01 | 0.63211 | 7.82E-03 |
| 0.81001 | 0.0263 | 0.0263 | 0.56113 | 0.03211 |
| 0.84347 | 0.00123 | 0.00119 | 0.49753 | 0.00229 |
| 0.89472 | 0.00416 | 0.00414 | 0.39303 | 0.01004 |
| 0.95087 | 3.07E-03 | 3.07E-03 | 0.22603 | 0.01048 |
| 0.97688 | 0.00201 | 0.00112 | 0.12742 | 0.00393 |

**S27. PEG4 Hygroscopicity**

**Fig S27.1**: Hygroscopicity of PEG4, at 293.15 K. Open squares, these CC-EDB experiments; solid line, UNIFAC model; blue line UManSysProp; red line adsorption isotherm model from Dutcher.

[Figure]

**Table S27.1:** Measured values of pure component melt density ($\rho_{melt}$) and refractive index ($n_{melt}$) (PEG4 is liquid), presented with parameterisation for solution measurements of density where x is the square root of MFS ($MFS^{1/2} = x$). *Upper* and *lower* refer to 95 % confidence limits for fits to experimental data. Upper and lower limit on refractive index and density are determined by the error in the refractometer and by the densitometer respectively.

| | $n_{melt}$ | $\rho_{melt}$/ g.cm$^{-3}$ | Polynomial fit ( $\rho_{sol} = a + b_1x + b_2x^2 + b_3x^3$) | | | |
| --- | --- | --- | --- | --- | --- | --- |
| | | | a | $b_1$ | $b_2$ | $b_3$ |
| *Best* | 1.4589 | 1.1271 | 999.97 | -37.39 | 296.85 | -130.68 |
| *Upper* | 1.4590 | 1.13412 | 999.97 | -9.65 | 235.84 | -92.25 |
| *Lower* | 1.4588 | 1.12338 | 999.97 | -65.13 | 357.86 | -169.11 |

**Table S27.2:** Tabulated experimental data points shown in **Fig S27.1.**

| $a_w$ | error $a_w$ (+ve) | error $a_w$ (-ve) | MFS | error MFS |
| --- | --- | --- | --- | --- |
| 0.52052 | 0.00336 | 0.00399 | 0.83006 | 8.065E-4 |
| 0.60966 | 0.00229 | 0.00278 | 0.78149 | 8.220E-4 |
| 0.65636 | 0.00166 | 0.00204 | 0.74058 | 0.00177 |
| 0.69735 | 0.00172 | 0.00212 | 0.71195 | 0.00154 |
| 0.74132 | 0.00556 | 0.00685 | 0.67145 | 0.00929 |
| 0.78212 | 6.975E-4 | 8.803E-4 | 0.63073 | 8.501E-4 |
| 0.81258 | 0.00536 | 0.00535 | 0.58791 | 0.00759 |
| 0.84243 | 0.00132 | 0.00111 | 0.52225 | 0.00213 |
| 0.89453 | 0.00427 | 0.00448 | 0.42827 | 0.01048 |
| 0.93766 | 0.00385 | 0.00376 | 0.31263 | 0.01217 |
| 0.98571 | 0.0013 | 0.00127 | 0.0918 | 0.00662 |
| 0.99969 | 0.00143 | 0.00156 | 0.0475 | 0.00252 |

**S28 Erythritol Hygroscopicity**

**Fig S28.1**: Hygroscopicity of erythritol (Sigma Aldrich 99 %), at 293.15 K. Open squares, these experiments; solid line, UNIFAC model.

[Figure]

**Table S28.1:** Pure component refractive index ($n_{melt}$) is determined using molar refraction, assuming ideal mixing for calculation of the melt density ($\rho_{melt}$), from bulk data available in Cai et al. (2016). The variation of density as a function of the root of solute mass fraction ($MFS^{1/2} = x$) is represented by polynomial fit parameters. *Upper* and *lower* refer to 95 % confidence limits for fits to experimental data, (Section 2.2 in manuscript).

| | $n_{melt}$ | $\rho_{melt}$/ g.cm$^{-3}$ | Polynomial fit ( $\rho_{sol} = a + b_1x + b_2x^2 + b_3x^3$) | | | |
|---|---|---|---|---|---|---|
| | | | a | $b_1$ | $b_2$ | $b_3$ |
| *Best* | 1.5211 | 1.3754 | 998.6 | 58.46 | 37.98 | 278.66 |
| *Upper* | 1.5388 | 1.3813 | 998.6 | 60.21 | 33.79 | 286.94 |
| *Lower* | 1.5204 | 1.3695 | 998.6 | 56.75 | 42.03 | 27.049 |

**Table S28.2:** Tabulated experimental data points shown in **Fig S28.1.**

| $a_w$ | error $a_w$ (+ve) | error $a_w$ (-ve) | MFS | error MFS |
|---|---|---|---|---|
| 0.62602 | 8.77112E-4 | 0.00107 | 0.71188 | 6.08334E-4 |
| 0.66147 | 0.0027 | 0.0033 | 0.66395 | 0.00226 |
| 0.72692 | 0.00104 | 0.00129 | 0.6342 | 6.57702E-4 |
| 0.75582 | 0.00775 | 0.00777 | 0.60723 | 0.00739 |
| 0.78929 | 0.01009 | 0.0101 | 0.57499 | 0.01315 |
| 0.83827 | 7.77253E-4 | 0.001 | 0.50705 | 9.72437E-4 |
| 0.86916 | 7.13427E-4 | 6.96511E-4 | 0.46004 | 0.00138 |
| 0.93195 | 2.64028E-4 | 3.52642E-4 | 0.28987 | 0.00175 |
| 0.95145 | 7.53773E-4 | 7.52526E-4 | 0.20621 | 0.00312 |
| 0.9815 | 5.76107E-4 | 5.56581E-4 | 0.13503 | 0.00279 |

**S29 Sorbitol Hygroscopicity**

**Fig S29.1**: Hygroscopicity of sorbitol (Sigma Aldrich ≥ 98 %), at 293.15 K. Open squares, these experiments; solid line, UNIFAC model. Data taken at RHs lower than indicated by the dashed black line show increased error in hygroscopicity retrieval due to the imposition of a kinetic limitation on water transport.

[Figure]

[Figure]

**Table S29.1:** Pure component refractive index ($n_{melt}$) determined using molar refraction where the melt density ($\rho_{melt}$) is determined using a polynomial fit of density to the square root of MFS ($MFS^{1/2} = x$). Bulk values used are available in Cai et al. (2016). *Upper* and *lower* refer to 95 % confidence limits for fits to experimental data.

| | $n_{melt}$ | $\rho_{melt}$/ g.cm$^{-3}$ | Polynomial fit ( $\rho_{sol} = a + b_1x + b_2x^2 + b_3x^3$) | | | |
| --- | --- | --- | --- | --- | --- | --- |
| | | | a | $b_1$ | $b_2$ | $b_3$ |
| *Best* | 1.5244 | 1.4231 | 997.8 | 8.6 | 286.1 | 130.7 |
| *Upper* | 1.5267 | 1.4333 | 997.8 | 24.74 | 234.56 | 175.54 |
| *Lower* | 1.5220 | 1.4128 | 997.8 | -7.6 | 337.59 | 85.83 |

**Table S29.2:** Tabulated experimental data points shown in **Fig S29.1.**

| $a_w$ | error $a_w$ (+ve) | error $a_w$ (-ve) | MFS | error MFS |
| --- | --- | --- | --- | --- |
| 0.50647 | 0.00432 | 0.00512 | 0.78667 | 0.00341 |
| 0.52291 | 0.0031 | 0.00369 | 0.7771 | 0.00307 |
| 0.54873 | 0.00705 | 0.00838 | 0.74672 | 0.00731 |
| 0.59535 | 0.00322 | 0.00389 | 0.73916 | 0.00193 |
| 0.60809 | 0.0019 | 0.0023 | 0.74976 | 0.00343 |
| 0.63682 | 0.00605 | 0.00728 | 0.70773 | 0.01216 |
| 0.67255 | 0.00497 | 0.00601 | 0.69271 | 0.00773 |
| 0.7035 | 0.00148 | 0.00183 | 0.69648 | 0.00163 |
| 0.73531 | 0.00619 | 0.00619 | 0.66608 | 0.00694 |
| 0.75896 | 0.00493 | 0.00599 | 0.62673 | 0.00941 |
| 0.78492 | 0.00775 | 0.00958 | 0.61901 | 0.00237 |
| 0.83722 | 0.00384 | 0.0025 | 0.56241 | 9.55991E-4 |
| 0.85049 | 9.622E-4 | 8.165E-4 | 0.5556 | 0.00118 |
| 0.88386 | 0.00262 | 0.0027 | 0.51154 | 0.00629 |
| 0.91574 | 0.00253 | 0.00266 | 0.4286 | 0.0076 |
| 0.94681 | 0.00245 | 0.00245 | 0.30429 | 0.01053 |
| 0.97555 | 0.0014 | 0.00139 | 0.14769 | 0.00774 |
| 0.99655 | 0.00112 | 6.78573E-4 | 0.02751 | 0.00293 |

**S30 D-(+)-Trehalose Dihydrate Hygroscopicity**

¶
¶
¶

**Fig S30.1**: Hygroscopicity of D-(+)-trehalose dihydrate (Sigma Aldrich ≥ 99 %), at 293.15 K. Open squares, these experiments; solid line, UNIFAC model. Data taken at RHs lower than indicated by the dashed black line show increased error in hygroscopicity retrieval due to the imposition of a kinetic limitation on water transport.

[Figure]

[Figure]

**Table S30.1:** Pure component refractive index ($n_{melt}$) determined using molar refraction where the melt density ($\rho_{melt}$) is determined using a polynomial fit of density to the square root of MFS ($MFS^{1/2} = x$). Bulk values used are available in Cai et al. (2016). *Upper* and *lower* refer to 95 % confidence limits for fits to experimental data.

| | $n_{melt}$ | $\rho_{melt}$/ g.cm$^{-3}$ | Polynomial fit ( $\rho_{sol} = a + b_1x + b_2x^2 + b_3x^3$) | | | |
| --- | --- | --- | --- | --- | --- | --- |
| | | | a | $b_1$ | $b_2$ | $b_3$ |
| *Best* | 1.5193 | 1.4682 | 997.8 | 8.2 | 284.3 | 177.8 |
| *Upper* | 1.5211 | 1.4734 | 997.8 | 11.6 | 269.79 | 194.19 |
| *Lower* | 1.5175 | 1.4629 | 997.8 | 4.87 | 298.84 | 161.43 |

**Table S30.2:** Tabulated experimental data points shown in **Fig S30.1.**

| $a_w$ | error $a_w$ (+ve) | error $a_w$ (-ve) | MFS | error MFS |
| --- | --- | --- | --- | --- |
| 0.51123 | 0.00397 | 0.0047 | 0.8511 | 0.00561 |
| 0.54636 | 0.01007 | 0.01196 | 0.81364 | 0.01816 |
| 0.5873 | 0.007 | 0.00844 | 0.85121 | 0.00732 |
| 0.60689 | 0.00263 | 0.00319 | 0.84879 | 0.00386 |
| 0.63303 | 0.01031 | 0.01244 | 0.79889 | 0.02031 |
| 0.67154 | 0.00716 | 0.00861 | 0.76858 | 0.00766 |
| 0.70479 | 0.00212 | 0.00262 | 0.80977 | 0.00199 |
| 0.72437 | 0.00577 | 0.00642 | 0.78413 | 0.00669 |
| 0.76384 | 0.01102 | 0.01364 | 0.743 | 0.00611 |
| 0.79679 | 0.00422 | 0.00225 | 0.7399 | 0.01219 |
| 0.81122 | 0.00282 | 0.00195 | 0.73624 | 0.0059 |
| 0.84712 | 0.00837 | 0.00721 | 0.69205 | 0.01427 |
| 0.88007 | 0.00598 | 0.00498 | 0.61945 | 0.01589 |
| 0.9118 | 5.25851E-4 | 5.4066E-4 | 0.58998 | 0.00159 |
| 0.93698 | 0.00204 | 0.00204 | 0.50101 | 0.00792 |
| 0.97142 | 0.00151 | 0.00149 | 0.3233 | 0.01015 |
| 0.99054 | 4.05516E-4 | 4.09208E-4 | 0.15195 | 0.00476 |

**S31. Galactose Hygroscopicity**

**Fig S31.1**: Hygroscopicity of (Sigma Aldrich ≥ 99 %), at 293.15 K. Open squares, these experiments; solid line, UNIFAC model. Data taken at RHs lower than indicated by the dashed black line show increased error in hygroscopicity retrieval due to the imposition of a kinetic limitation on water transport.

[Figure]

[Figure]

**Table SI.31.1:** Pure component refractive index ($n_{melt}$) is determined using molar refraction, assuming ideal mixing for calculation of the melt density ($\rho_{melt}$), from bulk data available in Cai et al. (2016). The variation of density as a function of the root of solute mass fraction ($MFS^{1/2} = x$) is represented by polynomial fit parameters. *Upper* and *lower* refer to 95 % confidence limits for fits to experimental data, (Section 2.2 in manuscript).

| | $n_{melt}$ | $\rho_{melt}$/ g.cm$^{-3}$ | Polynomial fit ( $\rho_{sol} = a + b_1x + b_2x^2 + b_3x^3$) | | | |
| --- | --- | --- | --- | --- | --- | --- |
| | | | a | $b_1$ | $b_2$ | $b_3$ |
| *Best* | 1.5885 | 1.6306 | 997.36 | 403.27 | 83.09 | 150.11 |
| *Upper* | 1.5892 | 1.6351 | 996.67 | 165.3 | -284.07 | 752.22 |
| *Lower* | 1.5878 | 1.6261 | 997.37 | 399.69 | 83.4 | 145.36 |

**Table S31.2:** Tabulated experimental data points shown in **Fig S31.1.**

| $a_w$ | error $a_w$ (+ve) | error $a_w$ (-ve) | MFS | error MFS |
| --- | --- | --- | --- | --- |
| 0.50996 | 0.00287 | 0.0034 | 0.82372 | 0.00382 |
| 0.60189 | 0.00267 | 0.00323 | 0.7993 | 0.00405 |
| 0.63684 | 0.00839 | 0.01012 | 0.76963 | 0.0055 |
| 0.72183 | 0.0016 | 0.00199 | 0.69438 | 0.00194 |
| 0.76282 | 0.00662 | 0.00694 | 0.68348 | 0.01289 |
| 0.80226 | 0.02704 | 0.02704 | 0.6317 | 0.02723 |
| 0.84064 | 0.00138 | 8.91966E-4 | 0.572 | 0.00141 |
| 0.88152 | 0.00559 | 0.00561 | 0.51157 | 0.01025 |
| 0.92485 | 0.00483 | 0.00491 | 0.43437 | 0.01532 |
| 0.96504 | 0.00377 | 0.00374 | 0.29773 | 0.01536 |
| 0.99822 | 0.00115 | 7.88489E-4 | 0.09505 | 0.00656 |

**S32 Xylose Hygroscopicity**

**Fig S32.1**: Hygroscopicity of (Sigma Aldrich ≥ 99 %), at 293.15 K. Open squares, these experiments; solid line, UNIFAC model.

[Figure]

**Table S32.1:** Pure component refractive index ($n_{melt}$) is determined using molar refraction, assuming ideal mixing for calculation of the melt density ($\rho_{melt}$), from bulk data available in Cai et al. (2016). The variation of density as a function of the root of solute mass fraction ($MFS^{1/2} = x$) is represented by polynomial fit parameters. *Upper* and *lower* refer to 95 % confidence limits for fits to experimental data, (Section 2.2 in manuscript).

| | $n_{melt}$ | $\rho_{melt}$/ g.cm$^{-3}$ | Polynomial fit ( $\rho_{sol} = a + b_1x + b_2x^2 + b_3x^3$) | | | |
| --- | --- | --- | --- | --- | --- | --- |
| | | | a | $b_1$ | $b_2$ | $b_3$ |
| *Best* | 1.5615 | 1.5626 | 996.73 | 127.69 | -163.53 | 597.09 |
| *Upper* | 1.5619 | 1.5653 | 996.74 | 126.37 | -159.45 | 591.57 |
| *Lower* | 1.5611 | 1.5598 | 996.72 | 128.97 | -167.5 | 602.42 |

**Table S32.2:** Tabulated experimental data points shown in **Fig S32.1.**

| $a_w$ | error $a_w$ (+ve) | error $a_w$ (-ve) | MFS | error MFS |
| --- | --- | --- | --- | --- |
| 0.97404 | 0.00732 | 0.00429 | 0.1841 | 0.0233 |
| 0.98465 | 0.00361 | 0.00212 | 0.12356 | 0.01215 |
| 0.996 | 0.00127 | 7.43479E-4 | 0.02995 | 0.00361 |
| 1.00081 | 0.00148 | 8.71845E-4 | 0.01372 | 0.0012 |

**S33 2,3-Dimethyl Succinic Acid Hygroscopicity**

**Fig S33.1**: Hygroscopicity of 2,3-dimethyl succinic acid (Sigma Aldrich ≥ 99 %), at 293.15 K. Open squares, these experiments; solid line, UNIFAC model. (Density treatment for 2,2-dimethyl succinic acid used.)

[Figure]

**Table S33.2:** Tabulated experimental data points shown in **Fig S33.1.**

| $a_w$ | error $a_w$ (+ve) | error $a_w$ (-ve) | MFS | error MFS |
|---|---|---|---|---|
| 0.94132 | 5.11673E-4 | 5.12405E-4 | 0.38395 | 0.00207 |
| 0.95214 | 0.00144 | 0.00144 | 0.32979 | 0.00859 |
| 0.96262 | 0.00159 | 0.00159 | 0.26369 | 0.01065 |
| 0.97285 | 0.00138 | 0.00138 | 0.19135 | 0.01011 |
| 0.98303 | 0.001 | 0.001 | 0.11733 | 0.00731 |
| 0.99417 | 2.09751E-4 | 2.24291E-4 | 0.03301 | 0.00121 |
| 0.99844 | 2.59195E-4 | 4.09162E-4 | 0.01724 | 4.61378E-4 |

**S34 Dimethyl Malonic Acid Hygroscopicity**

**Figure S34.1**: Hygroscopicity of (Sigma Aldrich 98 %), at 293.15 K. Open squares, these experiments; solid line, UNIFAC model. (Density treatment for methyl succinic acid used.)

[Figure]

**Table S34.2:** Tabulated experimental data points shown in **Figure S34.1.**

| $a_w$ | error $a_w$ (+ve) | error $a_w$ (-ve) | MFS | error MFS |
|---|---|---|---|---|
| 0.71262 | 0.00362 | 0.00449 | 0.7136 | 0.00301 |
| 0.744 | 0.0141 | 0.0141 | 0.69155 | 0.01343 |
| 0.78481 | 0.01088 | 0.01348 | 0.65412 | 0.00614 |
| 0.81516 | 0.01647 | 0.00985 | 0.62813 | 0.01311 |
| 0.83412 | 0.00246 | 0.00229 | 0.60844 | 0.00357 |
| 0.86818 | 0.00422 | 0.00426 | 0.5554 | 0.00729 |
| 0.90119 | 0.00509 | 0.00506 | 0.48761 | 0.01203 |
| 0.92833 | 0.00366 | 0.00365 | 0.40593 | 0.01475 |
| 0.96965 | 0.00157 | 0.00194 | 0.2089 | 0.01089 |
| 0.9897 | 4.75033E-4 | 4.76981E-4 | 0.05824 | 0.00271 |

**S35 Aspartic Acid Hygroscopicity**

**Fig S35.1**: Hygroscopicity of aspartic acid (Sigma Aldrich ≥ 99 %), at 293.15 K. Open squares, these experiments; solid line, UNIFAC model. (Density treatment for alanine used)

[Figure]

**Table S35.1:** Tabulated experimental data points shown in **Fig S35.1.**

| $a_w$ | error $a_w$ (+ve) | error $a_w$ (-ve) | MFS | error MFS |
|---|---|---|---|---|
| 0.99507 | 0.00448 | 0.00375 | 0.01431 | 7.18E-04 |
| 0.99599 | 0.00202 | 0.0017 | 0.01223 | 6.83E-04 |
| 0.99697 | 0.00141 | 0.00118 | 0.00882 | 5.15E-04 |
| 0.99793 | 0.00111 | 9.28E-04 | 0.00594 | 3.01E-04 |
| 0.99891 | 0.001 | 8.39E-04 | 0.00381 | 1.64E-04 |
| 0.99985 | 9.52E-04 | 7.98E-04 | 0.00266 | 8.72E-05 |

**S36 Asparagine Hygroscopicity**

**Fig S36.1**: Hygroscopicity of asparagine (Sigma Aldrich ≥ 98 %), at 293.15 K. Open squares, these experiments; solid line, UNIFAC model. (Density treatment for alanine used)

[Figure]

**Table S36.1:** Tabulated experimental data points shown in **Figure S36.1.**

| $a_w$ | error $a_w$ (+ve) | error $a_w$ (-ve) | MFS | error MFS |
|---|---|---|---|---|
| 0.53409 | 0.00178 | 0.00213 | 0.77577 | 0.00129 |
| 0.62935 | 0.00189 | 0.0023 | 0.74326 | 0.00101 |
| 0.63444 | 0.00381 | 0.00465 | 0.74081 | 0.00101 |
| 0.71441 | 0.00113 | 0.0014 | 0.68254 | 0.00175 |
| 0.74237 | 0.007 | 0.00854 | 0.67146 | 0.00782 |
| 0.81123 | 8.45796E-4 | 8.49613E-4 | 0.61254 | 0.00185 |
| 0.85278 | 0.00812 | 0.00813 | 0.54286 | 0.03203 |
| 0.9048 | 0.00102 | 9.46055E-4 | 0.46853 | 0.00454 |
| 0.94641 | 0.00108 | 0.0011 | 0.3002 | 0.00693 |
| 0.9951 | 2.80427E-4 | 2.96722E-4 | 0.02083 | 0.00124 |

**S37 Errors in Density and Refractive Index Parametrisations and their Impact on Hygroscopicity**

**Fig S37.1 Parametrisation for (a) density based on ideal mixing and bulk measured values for density up to the solubility limit and (b) refractive index predicted beyond the solubility limit using molar refraction. In both (a) and (b) dashed lines indicate the uncertainty envelope in the parametrisations. All bulk experimental values of aqueous density and refractive index are available in the supplementary information of Cai et al. (2016). In (c) measured equilibrium hygroscopicity curves are presented with upper and lower error envelope arising from the uncertainties in density and refractive index which is too small to be obvious.**

[Figure]

**S38 ΔMFS for Simple Straight Chain Dicarboxylic Acids**

**Fig S38.1 The difference in mass fraction of solute (ΔMFS) between values predicted by UNIFAC and experimental values (a) oxalic acid, (b) malonic acid, (c) succinic acid and (d) glutaric acid.**

[Figure]

**S39 Viscosity, Diffusion Constant and Timescale of Diffusional Mixing**

The kinetic modelling framework used in the analysis of the droplet evaporation events is valid only in the absence of a bulk-kinetic limitation on near surface composition, i.e. the particle must be assumed to be homogeneous in composition. Such a limitation was obvious for hygroscopicity measurements of trehalose, galactose and sorbitol at RH's lower than 80 %. To ensure the measurements are not compromised by bulk diffusion, we consider two important factors.

Firstly, the impact of viscosity on the hygroscopicity retrievals becomes very obvious when we consider the consistency and uncertainty in the raw hygroscopic growth curves determined from different droplets evaporating into differing RHs. Droplets drying into different RHs reach different compositions at different times, and will retain different amounts of water because of different drying rates. This leads to an artificially low MFS at a particular RH which then slowly returns to the equilibrium curve overtime. Thus, an inconsistency is apparent between retrieved hygroscopic growth curves (or MFS vs $a_w$) when drying into different RHs. An example of this is shown in Figure S39.1, where we report unbinned hygroscopicity data for alanine (a non-viscous amino acid) and trehalose (viscous at RHs lower than 80%). It is clear here that the different portions of the hygroscopic curves retrieved from measurements at different RHs are consistent for alanine but not for trehalose. A further easy way to identify this retention of water in a particle that is not fully

equilibrated is simply to measure the much longer time-dependence in size once the initial evaporation of water has stopped. In droplets that have reached a bulk diffusion limitation, the existence of a kinetic limitation is apparent in a steadily decreasing size as water continues to leave over a timescale longer than 10 s.

**Fig S39.1 a) Unbinned hygroscopicity data for the compound alanine.  b) Unbinned hygroscopicity data for the compound trehalose. At 50 % RH trehalose has a viscosity of 3.8 x 10⁵ Pa.s (Song et al. 2016).**

[Figure]

Secondly, we can determine the expected conditions under which we might expect problems to arise in retrieving hygroscopic growth curves from an evaporation measurement. Considering again trehalose at 80 % RH, an aqueous-trehalose droplet has a viscosity of 0.5 Pa.s, increasing to $3.8 \times 10^5$ Pa.s at 50 % RH (Song et al. 2016). Therefore, as the RH of the gas phase for the evaporation measurement is lowered, we can expect the increasing viscosity/decreasing diffusivity to become increasingly important. By contrast, for aqueous-carboxylic acid droplets, the viscosity never gets above 1 Pa s even at the driest RHs considered here (Song et al. 2016).

With these known dependencies of viscosity on water activity, we can estimate the timescale for diffusional mixing within a droplet, assuming that this provides an estimate of the timescale for an evaporating droplet to form a homogeneous mixture. This timescale must be considerably shorter than the evaporation timescale for our hygroscopicity estimations to be valid. First, the Stokes-Einstein equation is used to estimate the diffusion constant of water at varying viscosity (varying RH).

$$D = \frac{k_B T}{6 \pi r_{mol} \eta} \qquad (1.1)$$

$D$ is the diffusion constant, $k_B$ is the Boltzmann constant, $T$ is temperature, $r_{mol}$ is the molecular radius of water (taken as 1.375 Å) and $\eta$ is the viscosity. It should be noted that equation (1.1) is likely to provide a significant underestimate of the diffusion constant due to the failure of the Stokes-Einstein equation. At a viscosity of 100 Pa s, the diffusion constant for water in sucrose is already more than one order of magnitude larger than estimated from the viscosity (Power et al. 2013). However, using diffusion constants estimated from (1.1) will provide an upper limit on the diffusional mixing timescale. The timescale for diffusional mixing, τ, is then estimated using the expression

$$\tau = \frac{a^2}{\pi^2 D} \qquad (1.2)$$

where $a$ is the droplet radius (set as 10 microns in this calculation).

We compare the diffusional mixing timescales for aqueous droplets of trehalose, NaCl, NaNO₃ and glutaric acid in the newly added supplemental Figure S39.2 (and repeated below). Given that we have been able to report accurate hygroscopic growth curves for NaNO₃ down to 50 % RH (see Rovelli et al. 2016 and the

response to referee 2), it is clear that a final viscosity at 50 % of ~ 0.1 Pa.s (Baldelli et al.) is insufficient to impede accurate measurement of the hygroscopicity. Indeed, this suggests that water transport in any aerosol droplet that maintains a viscosity lower than 0.1 Pa.s during drying should remain sufficiently fast to avoid a bulk diffusion limitation, permitting accurate hygrosocpicity measurements. As an example of the diacarboyxlic acids considered in this study, glutaric acid has a considerably lower viscosity at 50 % RH of ~ 0.01 Pa.s (Song et al. 2016), indicative of what we might expect for all such similar systems. By contrast, aqueous-trehalose droplets cross the 0.1 Pa.s viscosity threshold at a water activity of ~0.85 (Song et al. 2016), commensurate with the deviation and increased scatter in the hygroscopicity measurements reported above for this compound.

Based on the two considerations above and to indicate clearly the water activity ranges over which we consider the hygroscopicity measurements to be valid for trehalose (S30), galactose (S31) and sorbitol (S29), we have added a dashed line to indicate where the data appear to become kinetically limited. We have added the following words to the captions of these Figures: "Data taken at RHs lower than indicated by the dashed black line show increased error in hygroscopicity retrieval due to the imposition of a kinetic limitation on water transport."

Again, we must reiterate that the true diffusion constants are generally found to be much larger than values estimated from the Stokes-Einstein equation. A droplet with a viscosity of 0.1 Pa s takes ~0.3 s to mix by diffusion based on our analysis here, but this is an upper limit on the timescale.

**Fig S39.2 a) Viscosity of Trehalose, NaCl, NaNO3 and Glutaric Acid as a function of RH. b) Estimated diffusion constant as a function of RH. c) Timescale for diffusional mixing at the RH shown on x-axis. Dashed green line represents 1 second timescale for diffusional mixing.**

[Figure]

A. Baldelli, R. M. Power, R. E. H. Miles, J. P. Reid and R. Vehring *Effect of crystallization kinetics on the properties of spray dried microparticles,* Aerosol Science and Technology, 2016, 50:7, 693-704, DOI:10.1080/02786826.2016.1177163

R. M. Power, S. H. Simpson, J. P. Reid and A. J. Hudson, *The transition from liquid to solid-like behaviour in ultrahigh viscosity aerosol particles,* Chemical Science, 2013, 4 , 2597, DOI: 10.1039/c3sc50682g

Y. Chul Song, A. E. Haddrell, B. R. Bzdek, J. P. Reid, T. Bannan, D. O. Topping,, C. Percival, and C. Cai *Measurements and Predictions of Binary Component Aerosol Particle Viscosity* J. Phys. Chem. A 2016, 120, 8123−8137, DOI: 10.1021/acs.jpca.6b07835

---

## Referee Report (RR1)

ACPD article:

**Influence of Organic Compound Functionality on Aerosol Hygroscopicity: Dicarboxylic Acids, Alkyl-Substituents, Sugars and Amino Acids**

*Article by Aleksandra Marsh et al.*

Review of the revised manuscript

The authors have revised their manuscript with consideration of the comments by the two referees. Most of my initial comments have been answered and changes implemented accordingly. The manuscript has been improved in clarity and presentation of results. There are only a few minor issues remaining (see below), which I suggest should be addressed in preparation of the final revised manuscript for publication.

**Specific comments**

- P5, Eq. (5): correct the symbol for saturation vapour pressure (currently $\rho°$ *"rho"*) to $p°$. This would be a more typical choice, would be in agreement with the expression given by Rovelli et al (2016) and avoids use of rho which stands for density in Eq. (2). Also, on line 15, correct spelling of "Fuchs-Sutugin" (only one t).

- P5, line 5. (Related to the response to my initial comment P 7, l. 2):
  *When referring to gradient in the text, we are referring to the gradient in water partial pressure and we believe this is correct. We do not refer to a gradient formed from (RH-aw). To be consistent with our previous publications, we have removed the subscript i entirely from the equation but not replaced it with w.*

  The revised sentence reads: "In this equation, the gradient in water partial pressure is the difference between the RH and $a_w$, the instantaneous water activity at the droplet surface."

  This remains a confusing description of what the equation actually states (and a more fitting description is given in Rovelli et al). First, "the difference between the RH and $a_w$" (i.e. RH - $a_w$) is simply not a (mathematical) gradient; rather it is a difference. A gradient is for example a difference per unit distance or its equivalent in partial differential from, but it is not simply a difference as implied in the statement. Second, "the difference between the RH and $a_w$" is not the gradient in water (vapour) partial pressure and does not directly represent it, even though there exist similar mass flux expressions with differences in partial pressures or differences in vapour densities as part of the formula. Both gradients as well as differences in water vapour partial pressures carry units different from RH - $a_w$. Third, from the given statement it is unclear to what "the instantaneous water activity at the droplet surface" refers to: should it refer to RH or to $a_w$? This needs to be clarified in the text as well as pointing out that RH in this equation refers to $RH_\infty$, the RH in the surrounding gas phase sufficiently far away from the droplet surface ($S_\infty$ in Rovelli et al.).

- P10, line 29: I suggest to modify the new sentence to read: "Hence thermodynamic model predictions for amino acids were generated using E-AIM, Model III (Clegg et al., 1998), using the standard UNIFAC model including certain modified main group interaction parameters introduced by Peng et al. (2001)." This modification is more clear in that it does not imply that Peng et al. parameterized the whole UNIFAC model (they only modified a small subset of main group interaction parameters).

- P12, line 15 *(related to Referee Comment: P12, l. 27: "Molecular structures presented in Fig. 10 are the open chain form, which must be used during modelling using UNIFAC."; Why "must"? AIOMFAC also allows you to use the cyclic structure of sugars in aqueous solution, e.g. glucopyranose instead of glucose, if desired.*
  *Response: Cyclic sugar structures do not appear to be available on AIOMFAC-web. Amended P11 L27 to read 'Molecular structures presented in Fig. 10 are the open chain form, which must be used during modelling using AIOMFAC-web.'*

  I do not understand how the authors come to that "which must be used" conclusion about the availability of cyclic sugar structures in AIOMFAC-web. AIOMFAC-web allows you to select from a wide range of organic subgroups and there is no problem in choosing those subgroups referring to cyclic sugar structures to define an organic compound (using the optin "Define Subgroups" on the input form for organic compounds). There are even examples given in the "Predefined List" input option, e.g. for D-mannopyranose, $(CH_2^{[OH]})(CH^{[OH]})_4(CHO[ether])(OH)_5$, the cyclic structure equivalent of the open-chain form of mannose. The manuscript text should be corrected and the authors may want to check whether replacing the open chain forms in Fig. 10 by equivalent cyclic sugar structures would lead to significantly different model curves.

---

## Author Response (AR2)

**Author Response to Referee #1 (Dr. Andreas Zuend) of "Influence of Organic Compound Functionality on Aerosol Hygroscopicity: Dicarboxylic Acids, Alkyl-Substituents, Sugars and Amino Acids"**

Aleksandra Marsh[1], Rachael E. H. Miles[1], Grazia Rovelli[1], Alexander G. Cowling[1], Lucy Nandy[2], Cari S. Dutcher[2] and Jonathan. P Reid[1]

[1] School of Chemistry, University of Bristol, Bristol, BS8 1TS, UK
[2] Department of Mechanical Engineering, University of Minnesota, 111 Church Street SE, Minneapolis, MN 55455, USA

*Correspondence to:* Jonathan. P. Reid j.p.reid@bristol.ac.uk

The authors would like to thank the referee for their supportive comments and the additional suggestions they have made for minor revision. We respond to these comments below.

*Referee Comment: P5, Eq. (5): correct the symbol for saturation vapour pressure (currently $\rho°$ "rho") to $p°$. This would be a more typical choice, would be in agreement with the expression given by Rovelli et al (2016) and avoids use of rho which stands for density in Eq. (2). Also, on line 15, correct spelling of "Fuchs-Sutugin" (only one t).*

Response: Both of these changes have been made on page 5.

*Referee Comment: P5, line 5. (Related to the response to my initial comment P 7, l. 2): "When referring to gradient in the text, we are referring to the gradient in water partial pressure and we believe this is correct. We do not refer to a gradient formed from (RH-aw). To be consistent with our previous publications, we have removed the subscript i entirely from the equation but not replaced it with w."*

*The revised sentence reads: "In this equation, the gradient in water partial pressure is the difference between the RH and aw, the instantaneous water activity at the droplet surface."*

*This remains a confusing description of what the equation actually states (and a more fitting description is given in Rovelli et al). First, "the difference between the RH and aw" (i.e. RH - aw) is simply not a (mathematical) gradient; rather it is a difference. A gradient is for example a difference per unit distance or its equivalent in partial differential from, but it is not simply a difference as implied in the statement. Second, "the difference between the RH and aw" is not the gradient in water (vapour) partial pressure and does not directly represent it, even though there exist similar mass flux expressions with differences in partial pressures or differences in vapour densities as part of the formula. Both gradients as well as differences in water vapour partial pressures carry units different from RH - aw. Third, from the given statement it is unclear to what "the instantaneous water activity at the droplet surface" refers to: should it refer to RH or to aw? This needs to be clarified in the text as well as pointing out that RH in this equation refers to RH∞, the RH in the surrounding gas phase sufficiently far away from the droplet surface (S∞ in Rovelli et al.).*

Response: We apologise to the lack of clarity remaining in this sentence and have now reworded it to read: "In this equation, the difference in water partial pressure between infinite distance and the droplet surface, which drives diffusional mass transport in the gas phase, is quantified by the difference between the RH and the instantaneous water activity at the droplet surface, $a_w$, respectively. This difference, a fraction of 1, should be considered in combination with the saturation vapour pressure $p^0$ which appears in the denominator of the first bracketed term in the equation, giving the true difference in vapour pressure between infinite distance and the droplet surface."

*Referee Comment:* P10, line 29: I suggest to modify the new sentence to read: "Hence thermodynamic model predictions for amino acids were generated using E-AIM, Model III (Clegg et al., 1998), using the standard UNIFAC model including certain modified main group interaction parameters introduced by Peng etal. (2001)." This modification is more clear in that it does not imply that Peng et al. parameterized the whole UNIFAC model (they only modified a small subset of main group interaction parameters).

Response: This has been added as suggested.

*Referee Comment:* P12, line 15 (related to Referee Comment: P12, l. 27: "Molecular structures presented in Fig. 10 are the open chain form, which must be used during modelling using UNIFAC."; Why "must"? AIOMFAC also allows you to use the cyclic structure of sugars in aqueous solution, e.g.glucopyranose instead of glucose, if desired.
"Cyclic sugar structures do not appear to be available on AIOMFAC-web. Amended P11 L27 to read 'Molecular structures presented in Fig. 10 are the open chain form, which must be used during modelling using AIOMFAC-web.'"
I do not understand how the authors come to that "which must be used" conclusion about the availability of cyclic sugar structures in AIOMFAC-web. AIOMFAC-web allows you to select from a wide range of organic subgroups and there is no problem in choosing those subgroups referring to cyclic sugar structures to define an organic compound (using the option "Define Subgroups" on the input form for organic compounds). There are even examples given in the "Predefined List" input option, e.g. for D-mannopyranose, $(CH_2^{[OH]})(CH^{[OH]})_4(CHO^{[ether]})(OH)_5$, the cyclic structure equivalent of the open-chain form of mannose. The manuscript text should be corrected and the authors may want to check whether replacing the open chain forms in Fig. 10 by equivalent cyclic sugar structures would lead to significantly different model curves.*

Response: We are very sorry for this misunderstanding, we were not initially aware of how to represent these sugars in their cyclic form in AIOMFAC-web but the referee's comment has been helpful at correcting this. Predictions for galactose and xylose have now been generated in their cyclic forms and the results are summarised in the Table below, also included in the Supplementary Information. An additional Figure (S40.0) has been provided in the supplementary information to clarify the difference between the open chain and cyclic predictions for these two compounds. However, we have not added these predictions to Figure 10 - we feel that the addition of further curves would make the figure too cluttered. UNIFAC predictions for trehalose are in its cyclic form.

**Table S40.0: Table of UNIFAC groups for cyclic and open chain galactose and xylose.**

| Compound | Open Chain (In Manuscript) | Cyclic |
|---|---|---|
| Galactose | $CHO\ (CH_1^{(OH)})_4\ CH_2^{(alc)}\ (OH)_5$ | $(CH^{[alc]})_4(CH_2^{[OH]})(CHO^{[ether]})(OH)_4$ |
| Xylose | $(CH_2(OH))_3\ CH_2^{(alc)}\ CHO\ (OH)_4$ | $(CH^{[OH]})_4(CHO^{[ether]})(OH)_4$ |

Further, we have deleted P12 L15-16: 'Which must be used during modelling with AIOMFAC-web.'

We have added P12 L15-16: 'Molecular structures presented in Fig. 10 are the open chain form for galactose and xylose and trehalose is represented using its cyclic form. Comparison of predictions for the open chain and cyclic structural forms for xylose and galactose are shown in Figure S40.0'

**Figure S40.0 Galactose and Xylose CK-EDB data as a function of MFS and water activity compared with predictions for both cyclic and open chain UNIFAC group thermodynamic predictions.**

[revised manuscript text omitted]

**Table S0.1** Fitted parameters for upper and lower MFS vs water activity of compounds in each class, amino and organic acids, sugars and alcohols, as shown in Figure 11b) in the manuscript. The power law coefficient $P$ is used to calculate energy parameter $C$ for the first to $(n-1)$th layers, hence $C_i = (i/n)^P$, where $i$ is the layer number and $n$ is the total number of hydration layers, here $n = 8$ for all compounds except glycine ($n = 3$) and 2,2-dimethyl glutaric acid ($n = 16$). MSE is a normalized mean-square error, equal to $\left(\frac{1}{n_p}\right)\Sigma_{i=1}^{n_p}((m_{model,i} - m_{data,i})/(m_{model,i}))^2$, where $n_p$ is the number of data points.

| Solute | $P$ | MSE |
|---|---|---|
| Amino acid Upper (Glycine) | -1.934 | 0.00321 |
| Amino acid Lower (Asparagine) | -0.171 | 0.04151 |
| Organic acid Upper (Malonic acid) | -0.212 | 0.00819 |
| Organic acid Lower (2,2 dimethyl glutaric acid) | 0.206 | 0.08315 |
| Sugar Upper (Sorbitol) | -0.522 | 0.01025 |
| Sugar Lower (Trehalose) | -0.870 | 0.01687 |
| Alcohol Upper (Erythritol) | -0.238 | 0.01311 |
| Alcohol Lower (PEG4) | -1.180 | 0.16205 |

**Table S0.2** Fitted parameters for nine amino acids. The power law coefficient $P$ is used to calculate energy parameter $C$ for the first to $(n-1)$th layers, hence $C_i = (i/n)^P$, where $i$ is the layer number and $n$ is the total number of hydration layers, here $n = 8$ for all compounds except glycine ($n = 3$) and threonine ($n = 5$). MSE is a normalized mean-square error, equal to $\left(\frac{1}{n_p}\right)\Sigma_{i=1}^{n_p}((m_{model,i} - m_{data,i})/(m_{model,i}))^2$, where $n_p$ is the number of data points. (Parameter for L-aspartic acid could not be determined due to data range available.)

| Solute | $P$ | MSE |
|---|---|---|
| Alanine | -0.356 | 0.00051 |
| Asparagine | -0.171 | 0.04151 |
| Arginine | -0.993 | 0.04039 |
| Glycine | -1.934 | 0.00321 |
| Histidine | -0.502 | 0.02211 |
| Lysine | -1.225 | 0.00667 |
| Proline | -0.619 | 0.03764 |
| Threonine | -0.960 | 0.20107 |
| Valine | -0.892 | 0.00397 |

**S1 DL-Alanine Hygroscopicity**

**Fig. S1.1**: Hygroscopicity of DL-Alanine (Sigma Aldrich, Purity 99 %) at 293.15 K.

[Figure]

**Table S1.1:** Pure component refractive index ($n_{melt}$) is determined using molar refraction, assuming ideal mixing for calculation of the melt density ($\rho_{melt}$), from bulk data available in Cai et al. (2016). The variation of density as a function of the root of solute mass fraction ($MFS^{1/2} = x$) is represented by polynomial fit parameters. *Upper* and *lower* refer to 95 % confidence limits for fits to experimental data, (Section 2.2 in manuscript).

| | $n_{melt}$ | $\rho_{melt}$/ g.cm$^{-3}$ | Polynomial fit ( $\rho_{sol} = a + b_1x + b_2x^2 + b_3x^3$) | | | |
|---|---|---|---|---|---|---|
| | | | a | $b_1$ | $b_2$ | $b_3$ |
| *Best* | 1.6205 | 1.4961 | 999 | 94.14 | -66.93 | 466.48 |
| *Upper* | 1.6222 | 1.5042 | 999 | 97.38 | -76.61 | 480.88 |
| *Lower* | 1.6188 | 1.4881 | 999 | 90.98 | -57.61 | 452.44 |

**Table S1.2:** Tabulated experimental data points shown in **Fig S1.1.**

| $a_w$ | error $a_w$ (+ve) | error $a_w$ (-ve) | MFS | error MFS |
|---|---|---|---|---|
| 0.75966 | 0.00182 | 0.00228 | 0.48336 | 8.66E-04 |
| 0.76866 | 1.02E-03 | 0.00128 | 0.47642 | 3.48E-04 |
| 0.77876 | 0.00413 | 0.00519 | 0.46428 | 0.00314 |
| 0.78887 | 0.00613 | 0.00771 | 0.45228 | 0.00412 |
| 0.80001 | 0.00674 | 0.00847 | 0.43959 | 0.00395 |
| 0.81774 | 0.00674 | 8.47E-03 | 0.41748 | 0.00512 |
| 0.83836 | 6.17E-03 | 7.75E-03 | 0.38848 | 6.39E-03 |
| 0.85334 | 0.00473 | 3.00E-03 | 0.37246 | 0.00116 |
| 0.86108 | 0.00116 | 7.54E-04 | 0.3655 | 8.92E-04 |
| 0.87144 | 6.02E-04 | 4.18E-04 | 0.34973 | 5.50E-04 |
| 0.87774 | 0.00139 | 9.81E-04 | 0.3386 | 0.00165 |
| 0.88866 | 0.00217 | 0.00153 | 0.31805 | 0.00274 |
| 0.89931 | 2.66E-03 | 1.92E-03 | 0.2981 | 0.00333 |
| 0.9087 | 0.00257 | 0.00183 | 0.27841 | 0.00381 |
| 0.91923 | 0.00256 | 0.00191 | 0.25483 | 0.00426 |
| 0.92957 | 0.00248 | 0.00181 | 0.23142 | 0.00416 |
| 0.93936 | 0.00243 | 0.00179 | 0.20672 | 0.00392 |
| 0.94936 | 0.00206 | 1.54E-03 | 0.18027 | 0.00361 |
| 0.9595 | 1.95E-03 | 1.43E-03 | 0.15211 | 0.00347 |
| 0.96935 | 1.49E-03 | 1.11E-03 | 0.12252 | 0.003 |
| 0.97954 | 0.00127 | 9.28E-04 | 0.08929 | 0.00254 |
| 0.99143 | 6.34E-04 | 7.22E-04 | 0.04259 | 0.00238 |

**S2 L-Arginine Hygroscopicity**

**Fig S2.1**: Hygroscopicity of L-Arginine, (Acros Organics, Purity > 98 %), at 293.15 K. Open squares, these experiments.

[Figure]

**Table S2.1:** Pure component refractive index ($n_{melt}$) is determined using molar refraction, assuming ideal mixing for calculation of the melt density ($\rho_{melt}$), from bulk data available in Cai et al. (2016). The variation of density as a function of the root of solute mass fraction (MFS $^{1/2}$ =x) is represented by polynomial fit parameters. *Upper* and *lower* refer to 95 % confidence limits for fits to experimental data, (Section 2.2 in manuscript).

|  | $n_{melt}$ | $\rho_{melt}$/ g.cm$^{-3}$ | Polynomial fit ( $\rho_{sol} = a + b_1x + b_2x^2 + b_3x^3$) | | | |
|---|---|---|---|---|---|---|
|  |  |  | a | $b_1$ | $b_2$ | $b_3$ |
| *Best* | 1.637 | 1.3995 | 998.6 | 59.85 | 28.54 | 310.48 |
| *Upper* | 1.6382 | 1.4045 | 998.6 | 61.44 | 24.47 | 317.9 |
| *Lower* | 1.6358 | 1.3945 | 998.6 | 58.28 | 32.51 | 303.13 |

**Table S2.2:** Tabulated experimental data points shown in **Fig S2.1.**

| $a_w$ | error $a_w$ (+ve) | error $a_w$ (-ve) | MFS | error MFS |
|---|---|---|---|---|
| 0.50205 | 0.00177 | 0.0021 | 0.69041 | 0.00113 |
| 0.59399 | 0.00171 | 0.00206 | 0.65822 | 0.00115 |
| 0.69132 | 0.00127 | 0.00157 | 0.62391 | 1.74E-04 |
| 0.71788 | 0.01297 | 0.01296 | 0.61768 | 0.00607 |
| 0.74796 | 0.0139 | 0.01716 | 0.6026 | 0.00755 |
| 0.80315 | 7.87E-04 | 0.001 | 0.55889 | 4.50E-04 |
| 0.84439 | 0.00739 | 0.00741 | 0.52351 | 0.014 |
| 0.89694 | 0.00128 | 0.00112 | 0.47038 | 0.00138 |
| 0.91074 | 0.00174 | 0.00175 | 0.44361 | 0.00473 |
| 0.96538 | 0.00317 | 0.00317 | 0.24814 | 0.01569 |
| 0.99761 | 5.53E-04 | 5.28E-04 | 0.03416 | 0.00266 |

**S3 Glycine Hygroscopicity**

**Fig S3.1**: Hygroscopicity of Glycine, (Santa Cruz Biotech LTD), at 293.15 K. Solid line standard UNIFAC prediction.

[Figure]

**Table S3.1:** Pure component refractive index ($n_{melt}$) is determined using molar refraction, assuming ideal mixing for calculation of the melt density ($\rho_{melt}$), from bulk data available in Cai et al. (2016). The variation of density as a function of the root of solute mass fraction (MFS$^{1/2}$ =x) is represented by polynomial fit parameters. *Upper* and *lower* refer to 95 % confidence limits for fits to experimental data, (Section 2.2 in manuscript).

| | $n_{melt}$ | $\rho_{melt}$/ $g.cm^{-3}$ | Polynomial fit ( $\rho_{sol} = a + b_1x + b_2x^2 + b_3x^3$) | | | |
|---|---|---|---|---|---|---|
| | | | a | $b_1$ | $b_2$ | $b_3$ |
| *Best* | 1.6634 | 1.6905 | 999.47 | 186.75 | -363.66 | 860.4 |
| *Upper* | 1.6654 | 1.7006 | 999.47 | 192.41 | -382.69 | 883.61 |
| *Lower* | 1.6613 | 1.6805 | 999.47 | 181.22 | -345.14 | 837.67 |

**Table S3.2:** Tabulated experimental data points shown in **Fig S3.1.**

| $a_w$ | error $a_w$ (+ve) | error $a_w$ (-ve) | MFS | error MFS |
|---|---|---|---|---|
| 0.51061 | 0.00328 | 0.00389 | 0.63189 | 0.00159 |
| 0.52315 | 0.0204 | 0.02421 | 0.62993 | 0.00129 |
| 0.57855 | 0.01673 | 0.01985 | 0.61512 | 0.01113 |
| 0.60598 | 0.00228 | 0.00276 | 0.59862 | 6.15E-04 |
| 0.65105 | 0.00995 | 0.01205 | 0.57691 | 0.00441 |
| 0.70256 | 0.00157 | 0.00195 | 0.53551 | 0.00146 |
| 0.73844 | 0.0068 | 0.00678 | 0.51233 | 0.00686 |
| 0.77309 | 0.01453 | 0.01453 | 0.48382 | 0.01515 |
| 0.83663 | 0.0021 | 0.00115 | 0.4015 | 0.00347 |
| 0.84998 | 0.00206 | 0.00204 | 0.38496 | 0.00336 |
| 0.90029 | 0.00391 | 0.00391 | 0.29984 | 0.00906 |
| 0.94266 | 0.00341 | 0.00339 | 0.19519 | 0.00966 |
| 0.98152 | 0.00147 | 0.00147 | 0.07624 | 0.00455 |

**S4 Histidine Hygroscopicity**

**Fig S4.1**: Hygroscopicity of L-Histidine, (VWR Chemicals), open symbols, these CC-EDB experiments.

[Figure]

**Table S4.1:** Pure component refractive index ($n_{melt}$) is determined using molar refraction, assuming ideal mixing for calculation of the melt density ($\rho_{melt}$), from bulk data available in Cai et al. (2016). The variation of density as a function of the root of solute mass fraction ($MFS^{1/2} = x$) is represented by polynomial fit parameters. *Upper* and *lower* refer to 95 % confidence limits for fits to experimental data, (Section 2.2 in manuscript).

| | $n_{melt}$ | $\rho_{melt}$/ $g.cm^{-3}$ | \multicolumn{4}{c}{Polynomial fit ( $\rho_{sol} = a + b_1x + b_2x^2 + b_3x^3$)} |
|---|---|---|---|---|---|---|
| | | | a | $b_1$ | $b_2$ | $b_3$ |
| *Best* | 1.6892 | 1.5378 | 998.9 | 111.5 | -119.61 | 542.86 |
| *Upper* | 1.6914 | 1.5462 | 998.9 | 115.17 | -130.97 | 558.8 |
| *Lower* | 1.6871 | 1.5296 | 998.9 | 107.98 | -108.77 | 527.49 |

**Table S4.2:** Tabulated experimental data points shown in **Fig S4.1.**

| $a_w$ | error $a_w$ (+ve) | error $a_w$ (-ve) | MFS | error MFS |
|---|---|---|---|---|
| \multicolumn{5}{c}{293.15 K} | | | | |
| 0.66801 | 0.00175 | 0.00214 | 0.64888 | 5.85836E-4 |
| 0.75174 | 0.00105 | 0.00131 | 0.61182 | 3.82887E-4 |
| 0.77265 | 0.00527 | 0.00661 | 0.59614 | 0.00177 |
| 0.83375 | 0.0064 | 0.00643 | 0.51198 | 0.01825 |
| 0.87281 | 0.00111 | 0.00101 | 0.48826 | 0.0027 |
| 0.9239 | 8.9548E-4 | 9.46372E-4 | 0.38721 | 0.00439 |
| 0.99296 | 6.37951E-4 | 6.3374E-4 | 0.03829 | 0.00295 |

**S5 L-Lysine Hygroscopicity**

**Fig S5.1**: Hygroscopicity of L-Lysine, (Sigma Aldrich, Purity $\geq$ 98 %), at 293.15 K. Open squares, these experiments; solid line, UNIFAC model.

[Figure]

**Table S5.1:** Pure component refractive index ($n_{melt}$) determined using molar refraction where the melt density ($\rho_{melt}$), is determined using a polynomial fit of density to the square root of MFS ($MFS^{1/2}$ =x). Bulk values used are available in Cai et al. (2016). *Upper* and *lower* refer to 95 % confidence limits for fits to experimental data.

|  | $n_{melt}$ | $\rho_{melt}$/ $g.cm^{-3}$ | Polynomial fit ( $\rho_{sol} = a + b_1x + b_2x^2 + b_3x^3$) | | | |
|---|---|---|---|---|---|---|
|  | | | a | $b_1$ | $b_2$ | $b_3$ |
| *Best* | 1.5586 | 1.2362 | 998.2 | -15.22 | 309.14 | -56.02 |
| *Upper* | 1.5614 | 1.2418 | 998.2 | -4.29 | 271.92 | -23.99 |
| *Lower* | 1.5558 | 1.2306 | 998.2 | -25.93 | 346.35 | -88.05 |

**Table S5.2:** Tabulated experimental data points shown in **Fig S5.1**.

| $a_w$ | error $a_w$ (+ve) | error $a_w$ (-ve) | MFS | error MFS |
|---|---|---|---|---|
| 0.50605 | 0.00267 | 0.00316 | 0.65479 | 0.00157 |
| 0.52404 | 0.01187 | 0.01406 | 0.63621 | 0.00337 |
| 0.57666 | 0.02059 | 0.02439 | 0.60815 | 0.00948 |
| 0.58372 | 0.02793 | 0.03308 | 0.60867 | 0.01931 |
| 0.64049 | 0.00405 | 0.00494 | 0.58275 | 4.04084E-4 |
| 0.64559 | 0.00205 | 0.00251 | 0.57997 | 4.926E-4 |
| 0.67292 | 0.00999 | 0.0122 | 0.56365 | 0.00675 |
| 0.68839 | 0.0225 | 0.02735 | 0.54807 | 0.02742 |
| 0.71755 | 0.00885 | 0.01092 | 0.53328 | 0.00647 |
| 0.72732 | 0.00179 | 0.00223 | 0.53359 | 0.00179 |
| 0.75098 | 0.0056 | 0.00696 | 0.51408 | 0.00626 |
| 0.77291 | 0.00939 | 0.01164 | 0.49095 | 0.01194 |
| 0.79224 | 0.00736 | 0.00909 | 0.46681 | 0.01099 |
| 0.80926 | 0.01092 | 0.01352 | 0.45505 | 0.01535 |
| 0.82751 | 0.01292 | 0.01604 | 0.43489 | 0.02026 |
| 0.85916 | 0.00152 | 0.00197 | 0.41093 | 0.00309 |
| 0.87288 | 0.00143 | 0.00143 | 0.39407 | 0.00288 |
| 0.88688 | 0.00151 | 0.00151 | 0.3739 | 0.00294 |
| 0.90999 | 0.00294 | 0.00337 | 0.33998 | 0.00983 |
| 0.93683 | 3.20551E-4 | 3.24824E-4 | 0.27222 | 0.00154 |
| 0.94931 | 0.00162 | 0.00162 | 0.23179 | 0.00544 |
| 0.97255 | 0.00147 | 0.00147 | 0.14683 | 0.00623 |
| 0.99465 | 4.49456E-4 | 4.50168E-4 | 0.03174 | 0.0021 |
| 1.00277 | 0.00113 | 0.00149 | 0.02039 | 0.00198 |

**S6 L-Proline Hygroscopicity**

**Figure S6.1**: Hygroscopicity of L-Proline, (Acros Organics, Purity + 99 %), at 293.15 K. Open squares, these experiments.

[Figure]

**Table S6.1:** Pure component refractive index ($n_{melt}$) is determined using molar refraction, assuming ideal mixing for calculation of the melt density ($\rho_{melt}$), from bulk data available in Cai et al. (2016). The variation of density as a function of the root of solute mass fraction (MFS$^{1/2}$ =x) is represented by polynomial fit parameters. *Upper* and *lower* refer to 95 % confidence limits for fits to experimental data, (Section 2.2 in manuscript).

| | $n_{melt}$ | $\rho_{melt}$/ $g.cm^{-3}$ | Polynomial fit ( $\rho_{sol} = a + b_1x + b_2x^2 + b_3x^3$) | | | |
| --- | --- | --- | --- | --- | --- | --- |
| | | | a | $b_1$ | $b_2$ | $b_3$ |
| *Best* | 1.5948 | 1.3866 | 999 | 55.7 | 39.01 | 291 |
| *Upper* | 1.5964 | 1.3945 | 999 | 58.13 | 32.96 | 302.44 |
| *Lower* | 1.5932 | 1.3788 | 999 | 53.36 | 44.73 | 279.93 |

**Table S6.2:** Tabulated experimental data points shown in **Fig S6.1.**

| $a_w$ | error $a_w$ (+ve) | error $a_w$ (-ve) | MFS | error MFS |
| --- | --- | --- | --- | --- |
| 0.52739 | 0.00183 | 0.00218 | 0.647 | 1.53E-04 |
| 0.59111 | 0.01061 | 0.01264 | 0.62414 | 0.00218 |
| 0.61213 | 0.00154 | 0.00186 | 0.6013 | 1.44E-04 |
| 0.64995 | 0.00905 | 0.01098 | 0.58549 | 0.00122 |
| 0.70619 | 9.52E-04 | 0.00118 | 0.54716 | 3.08E-04 |
| 0.73823 | 0.0057 | 0.00705 | 0.52349 | 0.00436 |
| 0.79982 | 7.99E-04 | 0.00101 | 0.46617 | 7.45E-04 |
| 0.80883 | 0.00112 | 0.00112 | 0.45742 | 1.45E-03 |
| 0.87515 | 1.30E-03 | 0.00103 | 0.36217 | 2.56E-03 |
| 0.91551 | 0.00145 | 0.00184 | 0.30701 | 0.00352 |
| 0.93455 | 9.07E-04 | 9.29E-04 | 0.24258 | 0.00279 |
| 0.99172 | 5.01E-04 | 5.60E-04 | 0.01734 | 0.0011 |

**S7 L-Threonine Hygroscopicity**

**Fig S7.1**: Hygroscopicity of L-Threonine, (Acros Organics, Purity 98 %), at 293.15 K. Open squares, these experiments; solid line, UNIFAC model.

[Figure]

**Table S7.1:** Pure component refractive index ($n_{melt}$) is determined using molar refraction, assuming ideal mixing for calculation of the melt density ($\rho_{melt}$), from bulk data available in Cai et al. (2016). The variation of density as a function of the root of solute mass fraction ($MFS^{1/2} = x$) is represented by polynomial fit parameters. *Upper* and *lower* refer to 95 % confidence limits for fits to experimental data, (Section 2.2 in manuscript).

|  | $n_{melt}$ | $\rho_{melt}/ g.cm^{-3}$ | Polynomial fit ( $\rho_{sol} = a + b_1 x + b_2 x^2 + b_3 x^3$) | | | |
|---|---|---|---|---|---|---|
|  |  |  | a | $b_1$ | $b_2$ | $b_3$ |
| *Best* | 1.6185 | 1.4977 | 999.4 | 94.57 | -68.14 | 468.44 |
| *Upper* | 1.6274 | 1.5403 | 999.4 | 112.31 | -121.99 | 546.4 |
| *Lower* | 1.6102 | 1.4575 | 999.4 | 79.24 | -23.63 | 399.69 |

**Table S7.2:** Tabulated experimental data points shown in **Fig S7.1.**

| $a_w$ | error $a_w$ (+ve) | error $a_w$ (-ve) | MFS | error MFS |
|---|---|---|---|---|
| 0.47807 | 0.00212 | 0.0025 | 0.70511 | 0.00173 |
| 0.53888 | 0.0052 | 0.0062 | 0.69319 | 0.00723 |
| 0.58237 | 0.01442 | 0.01711 | 0.67043 | 0.00714 |
| 0.60978 | 0.00127 | 0.00154 | 0.65941 | 3.88E-04 |
| 0.63867 | 0.01393 | 0.01689 | 0.65875 | 0.00707 |
| 0.68779 | 0.00158 | 0.00195 | 0.61529 | 0.00161 |
| 0.73352 | 0.0081 | 0.00812 | 0.59255 | 0.0041 |
| 0.75945 | 0.02291 | 0.02781 | 0.58815 | 0.04754 |
| 0.80118 | 0.00157 | 7.31E-04 | 0.51778 | 0.00135 |
| 0.86674 | 7.26E-04 | 4.84E-04 | 0.44784 | 3.66E-04 |
| 0.89045 | 0.00426 | 0.00419 | 0.39429 | 0.0104 |
| 0.93289 | 0.00438 | 0.00418 | 0.29104 | 0.01212 |
| 0.98064 | 0.00213 | 0.00214 | 0.10966 | 0.00862 |
| 0.99865 | 7.16E-04 | 4.45E-04 | 0.02317 | 0.00125 |

**S8 L-Valine Hygroscopicity**

**Figure S8.1**: Hygroscopicity of L-Valine, (Sigma Aldrich, Purity ≥ 98 %), at 293.15 K (blue Open symbols, these CK-EDB experiments; black filled circles, literature data (Kuramochi *et al.*); solid black line, UNIFAC model (293.15 K).

[Figure]

**Table S8.1:** Pure component refractive index ($n_{melt}$) is determined using molar refraction, assuming ideal mixing for calculation of the melt density ($\rho_{melt}$), from bulk data available in Cai et al. (2016). The variation of density as a function of the root of solute mass fraction ($MFS^{1/2} =x$) is represented by polynomial fit parameters. *Upper* and *lower* refer to 95 % confidence limits for fits to experimental data, (Section 2.2 in manuscript).

| | $n_{melt}$ | $\rho_{melt}/\ g.cm^{-3}$ | Polynomial fit ( $\rho_{sol} = a + b_1x + b_2x^2 + b_3x^3$) | | | |
|---|---|---|---|---|---|---|
| | | | a | $b_1$ | $b_2$ | $b_3$ |
| *Best* | 1.5791 | 1.2824 | 998.77 | 28.73 | 94.37 | 159.64 |
| *Upper* | 1.58 | 1.2872 | 998.77 | 29.8 | 92.82 | 164.91 |
| *Lower* | 1.5781 | 1.2776 | 998.77 | 27.71 | 95.81 | 154.45 |

**Table S8.2:** Tabulated experimental data points shown in **Fig S8.1.**

| $a_w$ | error $a_w$ (+ve) | error $a_w$ (-ve) | MFS | error MFS |
|---|---|---|---|---|
| | | 293.15 K | | |
| 0.92062 | 0.0027 | 0.00232 | 0.29295 | 0.00499 |
| 0.93004 | 2.26E-03 | 0.00195 | 0.26962 | 4.18E-03 |
| 0.93941 | 0.00173 | 0.00148 | 0.24396 | 0.00404 |
| 0.94943 | 0.00169 | 0.00145 | 0.21478 | 0.00403 |
| 0.9599 | 0.00138 | 0.00118 | 0.18125 | 0.00388 |
| 0.97008 | 0.00118 | 0.00101 | 0.14482 | 0.00345 |
| 0.98014 | 8.66E-04 | 7.41E-04 | 0.10314 | 2.94E-03 |
| 0.99117 | 5.34E-04 | 5.69E-04 | 0.04451 | 0.00201 |
| 0.99669 | 4.13E-04 | 4.57E-04 | 0.02205 | 8.65E-04 |

**S9 Citric Acid Hygroscopicity**

**Figure S9.1**: Hygroscopicity of Citric Acid, (Sigma Aldrich, Purity 99 %), at 293.15 K. Open squares, these EDB experiments; solid line, UNIFAC model.

[Figure]

**Table S9.1:** Pure component refractive index ($n_{melt}$) determined using molar refraction where the melt density ($\rho_{melt}$) is determined using a polynomial fit of density to the square root of MFS (MFS$^{\frac{1}{2}}$ = x). Bulk values used are available in Cai et al. (2016). *Upper* and *lower* refer to 95 % confidence limits for fits to experimental data.

|  | $n_{melt}$ | $\rho_{melt}$/ $g.cm^{-3}$ | Polynomial fit ( $\rho_{sol} = a + b_1x + b_2x^2 + b_3x^3$) | | | |
|---|---|---|---|---|---|---|
|  |  |  | a | $b_1$ | $b_2$ | $b_3$ |
| *Best* | 1.5054 | 1.550 | 998.0 | 25.0 | 253.84 | 273.2 |
| *Upper* | 1.5071 | 1.5565 | 998.0 | 37.88 | 211.13 | 309.49 |
| *Lower* | 1.5037 | 1.5436 | 998.0 | 12.11 | 296.56 | 236.92 |

**Table S9.2:** Tabulated experimental data points shown in **Fig S9.1.**

| $a_w$ | error $a_w$ (+ve) | error $a_w$ (-ve) | MFS | error MFS |
|---|---|---|---|---|
| 0.53894 | 0.0024 | 0.00286 | 0.76226 | 0.00233 |
| 0.59688 | 0.01043 | 0.01244 | 0.73375 | 0.00875 |
| 0.62961 | 0.00223 | 0.00271 | 0.70793 | 7.62E-04 |
| 0.6837 | 0.00876 | 0.01065 | 0.67914 | 0.00409 |
| 0.72762 | 0.00123 | 0.00153 | 0.64368 | 0.00135 |
| 0.74592 | 0.00403 | 0.00404 | 0.63069 | 0.00342 |
| 0.80229 | 0.00504 | 0.0029 | 0.58246 | 0.00401 |
| 0.82734 | 0.00237 | 0.00196 | 0.56406 | 0.00314 |
| 0.88104 | 0.00107 | 0.0012 | 0.4682 | 0.00149 |
| 0.90968 | 0.00331 | 0.00327 | 0.41761 | 0.00738 |
| 0.95487 | 0.0028 | 0.00279 | 0.26562 | 0.01165 |
| 0.99255 | 6.21E-04 | 6.70E-04 | 0.03973 | 0.00355 |

**S10 L-Tartaric Acid Hygroscopicity**

**Figure S10.1**: Hygroscopicity of Tartaric Acid, (Sigma Aldrich, Purity $\geq$ 99.5 %), at 293.15 K. Open squares, these EDB experiments; solid line, UNIFAC model.

[Figure]

**Table S10.1:** Pure component refractive index ($n_{melt}$) determined using molar refraction where the melt density ($\rho_{melt}$) is determined using a polynomial fit of density to the square root of MFS (MFS$^{\frac{1}{2}}$ = x). Bulk values used are available in Cai et al. (2016). *Upper* and *lower* refer to 95 % confidence limits for fits to experimental data.

| | $n_{melt}$ | $\rho_{melt}$/ g.cm$^{-3}$ | Polynomial fit ( $\rho_{sol} = a + b_1x + b_2x^2 + b_3x^3$) | | | |
| --- | --- | --- | --- | --- | --- | --- |
| | | | a | $b_1$ | $b_2$ | $b_3$ |
| *Best* | 1.4992 | 1.6007 | 999 | 15.08 | 325.84 | 260.78 |
| *Upper* | 1.4996 | 1.6128 | 999 | 29.23 | 273.11 | 311.49 |
| *Lower* | 1.4936 | 1.5886 | 999 | 93.2 | 378.58 | 210.06 |

**Table S10.2:** Tabulated experimental data points shown in **Fig S10.1.**

| $a_w$ | error $a_w$ (+ve) | error $a_w$ (-ve) | MFS | error MFS |
| --- | --- | --- | --- | --- |
| 0.49764 | 0.00285 | 0.00337 | 0.74075 | 0.00145 |
| 0.59229 | 0.00275 | 0.00332 | 0.70005 | 0.00184 |
| 0.62457 | 0.00897 | 0.01082 | 0.68273 | 0.00361 |
| 0.67107 | 0.00165 | 0.00203 | 0.64893 | 7.95E-04 |
| 0.70799 | 0.00711 | 0.00826 | 0.6255 | 0.0076 |
| 0.75229 | 0.00853 | 0.01048 | 0.59046 | 0.00337 |
| 0.79666 | 0.00778 | 0.00946 | 0.56049 | 0.00992 |
| 0.84739 | 9.37E-04 | 5.51E-04 | 0.45906 | 0.00122 |
| 0.86463 | 0.00206 | 0.00206 | 0.44068 | 0.00269 |
| 0.91248 | 0.00302 | 0.00302 | 0.34362 | 0.00774 |
| 0.95599 | 0.00217 | 0.00216 | 0.20415 | 0.00789 |
| 0.98847 | 0.00104 | 0.00105 | 0.03363 | 0.00337 |

**S11 Oxalic Acid Hygroscopicity**

**Fig S11.1**: Hygroscopicity of Oxalic Acid, (Sigma Aldrich, Purity 98 %), at 293.15 K. Open squares, these experiments; solid line, UNIFAC model.

[Figure]

**Table S11.1:** Pure component refractive index ($n_{melt}$) is determined using molar refraction, assuming ideal mixing for calculation of the melt density ($\rho_{melt}$), from bulk data available in Cai et al. (2016). The variation of density as a function of the root of solute mass fraction ($MFS^{1/2} = x$) is represented by polynomial fit parameters. *Upper* and *lower* refer to 95 % confidence limits for fits to experimental data, (Section 2.2 in manuscript).

| | $n_{melt}$ | $\rho_{melt}/$ g.cm$^{-3}$ | Polynomial fit ( $\rho_{sol} = a + b_1x + b_2x^2 + b_3x^3 + b_4x^4 + b_5x^5 + b_6x^6$) | | | | | | |
| --- | --- | --- | --- | --- | --- | --- | --- | --- | --- |
| | | | a | $b_1$ | $b_2$ | $b_3$ | $b_4$ | $b_5$ | $b_6$ |
| *Best* | 1.5167 | 1.7237 | 998.4 | -14.98 | 636.47 | -1074.2 | 2603.92 | -2596.5 | 1170.54 |
| *Upper* | 1.5185 | 1.7403 | 998.4 | -16.27 | 660.48 | -1165.1 | 2811.06 | -2809 | 1260.66 |
| *Lower* | 1.5149 | 1.7073 | 998.4 | -13.78 | 613.65 | -989.39 | 2409.96 | -2397.5 | 1085.84 |

**Table S11.2:** Tabulated experimental data points shown in **Fig S11.1.**

| $a_w$ | error $a_w$ (+ve) | error $a_w$ (-ve) | MFS | error MFS |
| --- | --- | --- | --- | --- |
| 0.75352 | 0.00146 | 0.00183 | 0.49497 | 0.00148 |
| 0.77652 | 0.00502 | 0.00629 | 0.47731 | 0.0077 |
| 0.79664 | 0.00652 | 0.00817 | 0.46116 | 0.00912 |
| 0.81614 | 0.00716 | 0.00896 | 0.44009 | 0.00613 |
| 0.83841 | 0.00803 | 0.01005 | 0.41068 | 0.00808 |
| 0.85938 | 0.0012 | 7.60E-04 | 0.36829 | 0.00113 |
| 0.87602 | 0.00199 | 0.00188 | 0.34275 | 0.00355 |
| 0.89702 | 0.00235 | 0.00215 | 0.30804 | 0.00596 |
| 0.92388 | 8.50E-04 | 0.00115 | 0.25275 | 0.00331 |
| 0.93784 | 0.00106 | 0.00106 | 0.21515 | 0.00299 |
| 0.9589 | 8.73E-04 | 8.75E-04 | 0.15314 | 0.00313 |
| 0.98012 | 5.72E-04 | 5.68E-04 | 0.07645 | 0.00227 |
| 0.99129 | 2.93E-04 | 3.54E-04 | 0.03567 | 8.98E-04 |

**S12 Malonic Acid Hygroscopicity**

**Figure S12.1**: Hygroscopicity of Malonic Acid, (Sigma Aldrich, Purity 98 %), at 293.15 K. Open squares, these experiments; solid line, UNIFAC model.

[Figure]

**Table S12.1:** Pure component refractive index determined using molar refraction where the melt density is determined using a polynomial fit of density to the square root of MFS ($MFS^{1/2} = x$). Bulk values used are available in Cai et al. (2016). *Upper* and *lower* refer to 95 % confidence limits for fits to experimental data.

| | $n_{melt}$ | $\rho_{melt}/\ g.cm^{-3}$ | Polynomial fit ( $\rho_{sol} = a + b_1x + b_2x^2 + b_3x^3$ ) | | | |
|---|---|---|---|---|---|---|
| | | | a | $b_1$ | $b_2$ | $b_3$ |
| *Best* | 1.4611 | 1.4558 | 997.2 | 13.47 | 262.36 | 182.76 |
| *Upper* | 1.4627 | 1.4612 | 997.2 | 20.7 | 235.91 | 207.37 |
| *Lower* | 1.4594 | 1.4504 | 997.2 | 6.24 | 288.82 | 158.15 |

**Table S12.2:** Tabulated experimental data points shown in **Fig S12.1**.

| $a_w$ | error $a_w$ (+ve) | error $a_w$ (-ve) | MFS | error MFS |
|---|---|---|---|---|
| 0.63613 | 4.14E-04 | 5.04E-04 | 0.6718 | 1.62E-04 |
| 0.64803 | 0.00253 | 0.00308 | 0.66275 | 0.00161 |
| 0.65776 | 0.00301 | 0.00367 | 0.65273 | 0.00197 |
| 0.66822 | 0.00457 | 0.00558 | 0.64421 | 0.00162 |
| 0.67795 | 0.00624 | 0.00761 | 0.64043 | 0.00324 |
| 0.68747 | 0.00679 | 0.00828 | 0.62992 | 0.00337 |
| 0.6994 | 0.0051 | 0.00625 | 0.63085 | 0.00413 |
| 0.7117 | 4.61E-04 | 5.72E-04 | 0.6176 | 2.85E-04 |
| 0.71572 | 0.00132 | 0.00164 | 0.61275 | 0.00135 |
| 0.72728 | 0.00371 | 0.00458 | 0.59849 | 0.00119 |
| 0.73786 | 0.00398 | 0.0049 | 0.59485 | 0.00362 |
| 0.74792 | 0.00441 | 0.00544 | 0.58075 | 0.00309 |
| 0.75777 | 0.00438 | 0.00539 | 0.57442 | 0.00518 |
| 0.76852 | 0.00437 | 0.00539 | 0.56217 | 0.00227 |
| 0.77901 | 0.00483 | 0.00595 | 0.55399 | 0.00568 |
| 0.78948 | 0.00564 | 0.00694 | 0.53743 | 0.0039 |
| 0.79831 | 0.00665 | 0.00816 | 0.53501 | 0.00406 |
| 0.80703 | 0.00469 | 0.00576 | 0.52388 | 0.00468 |
| 0.81779 | 0.00517 | 0.00637 | 0.50809 | 0.00307 |
| 0.82931 | 0.00495 | 0.0061 | 0.49855 | 0.00449 |
| 0.83997 | 0.00501 | 0.00616 | 0.48341 | 0.0058 |
| 0.85259 | 0.0013 | 8.20E-04 | 0.46721 | 9.32E-04 |
| 0.85596 | 0.00112 | 6.98E-04 | 0.46233 | 0.00114 |
| 0.86726 | 0.00223 | 0.00146 | 0.44257 | 0.00286 |
| 0.87898 | 0.00278 | 0.00183 | 0.4203 | 0.0033 |
| 0.8885 | 0.00291 | 0.00204 | 0.4002 | 0.0044 |

| | | | | |
|---|---|---|---|---|
| 0.89906 | 0.00294 | 0.00205 | 0.37643 | 0.00371 |
| 0.90919 | 0.00337 | 0.00233 | 0.35563 | 0.00559 |
| 0.9213 | 1.65E-04 | 2.03E-04 | 0.32987 | 3.33E-04 |
| 0.92743 | 2.61E-04 | 2.85E-04 | 0.31545 | 0.00122 |
| 0.93737 | 3.90E-04 | 4.01E-04 | 0.27835 | 0.00185 |
| 0.94793 | 4.69E-04 | 4.71E-04 | 0.24233 | 0.00205 |
| 0.95802 | 4.67E-04 | 4.69E-04 | 0.20227 | 0.00223 |
| 0.96932 | 0.0018 | 0.0011 | 0.14857 | 0.00427 |
| 0.97897 | 0.00144 | 8.79E-04 | 0.10171 | 0.00358 |
| 0.98599 | 0.00158 | 9.62E-04 | 0.07431 | 0.00205 |

**S13 Succinic Acid Hygroscopicity**

**Fig S13.1**: Hygroscopicity of Succinic Acid, (Sigma Aldrich, Purity 99 %), at 293.15 K. Open squares, these experiments; solid line, UNIFAC model.

[Figure]

**Table S13.1:** Pure component refractive index ($n_{melt}$) is determined using molar refraction, assuming ideal mixing for calculation of the melt density ($\rho_{melt}$), from bulk data available in Cai et al. (2016). The variation of density as a function of the root of solute mass fraction ($MFS^{1/2} = x$) is represented by polynomial fit parameters. *Upper* and *lower* refer to 95 % confidence limits for fits to experimental data, (Section 2.2 in manuscript).

| | $n_{melt}$ | $\rho_{melt}/$ g.cm$^{-3}$ | Polynomial fit ( $\rho_{sol} = a + b_1x + b_2x^2 + b_3x^3 + b_4x^4 + b_5x^5 + b_6x^6$) | | | | | | |
|---|---|---|---|---|---|---|---|---|---|
| | | | a | $b_1$ | $b_2$ | $b_3$ | $b_4$ | $b_5$ | $b_6$ |
| *Best* | 1.4928 | 1.4185 | 998.2 | -1.96 | 324.69 | -146.48 | 426.3 | -373.62 | 191.37 |
| *Upper* | 1.4935 | 1.4249 | 998.2 | -2.08 | 329.57 | -155.12 | 447.91 | -395.04 | 201.45 |
| *Lower* | 1.4920 | 1.4122 | 998.2 | -1.85 | 319.93 | -138.32 | 405.79 | -353.36 | 181.79 |

**Table S13.2:** Tabulated experimental data points shown in **Fig S13.1.**

| $a_w$ | error $a_w$ (+ve) | error $a_w$ (-ve) | MFS | error MFS |
|---|---|---|---|---|
| 0.69803 | 0.00919 | 0.00918 | 0.70299 | 0.00502 |
| 0.76653 | 0.0191 | 0.02355 | 0.64896 | 0.00963 |
| 0.83176 | 0.0191 | 0.02355 | 0.56672 | 0.01018 |
| 0.86728 | 0.00142 | 0.00142 | 0.47926 | 0.0025 |
| 0.91247 | 0.00444 | 0.0044 | 0.37868 | 0.01328 |
| 0.96915 | 0.00137 | 0.00136 | 0.15909 | 0.00637 |
| 0.99255 | 3.09E-04 | 3.16E-04 | 0.04733 | 0.00178 |

**S14 Glutaric Acid Hygroscopicity**

**Figure S14.1**: Hygroscopicity of Glutaric Acid, (Sigma Aldrich, Purity 99 %), at 293.15 K. Open squares, these experiments; solid line, UNIFAC model.

[Figure]

**Table S14.1:** Pure component refractive index ($n_{melt}$) determined using molar refraction where the melt density ($\rho_{melt}$) is determined using a polynomial fit of density to the square root of MFS (MFS$^{1/2}$ =x). Bulk values used are available in Cai et al. (2016). *Upper* and *lower* refer to 95 % confidence limits for fits to experimental data.

| | $n_{melt}$ | $\rho_{melt}$/ $g.cm^{-3}$ | \multicolumn{4}{c}{Polynomial fit ( $\rho_{sol} = a + b_1x + b_2x^2 + b_3x^3$) } |
| --- | --- | --- | --- | --- | --- | --- |
| | | | a | $b_1$ | $b_2$ | $b_3$ |
| *Best* | 1.4655 | 1.2745 | 997.5 | -1.56 | 238.79 | 39.75 |
| *Upper* | 1.4660 | 1.2760 | 997.5 | 0.401 | 231.59 | 46.55 |
| *Lower* | 1.4649 | 1.2729 | 997.5 | -3.53 | 245.98 | 32.95 |

**Table S14.2:** Tabulated experimental data points shown in **Fig S14.1.**

| $a_w$ | error $a_w$ (+ve) | error $a_w$ (-ve) | MFS | error MFS |
| --- | --- | --- | --- | --- |
| 0.64988 | 4.93E-04 | 6.03E-04 | 0.80052 | 2.01E-04 |
| 0.66053 | 5.88E-04 | 7.21E-04 | 0.79339 | 2.71E-04 |
| 0.67122 | 0.00348 | 0.00426 | 0.78724 | 9.92E-04 |
| 0.68089 | 0.00458 | 0.00561 | 0.78134 | 0.00256 |
| 0.69112 | 0.00522 | 0.00639 | 0.76761 | 0.00226 |
| 0.70268 | 4.02E-04 | 4.98E-04 | 0.75519 | 1.75E-04 |
| 0.70969 | 0.0019 | 0.00235 | 0.75207 | 8.79E-04 |
| 0.72069 | 0.00357 | 0.00441 | 0.74499 | 0.00276 |
| 0.73156 | 0.0038 | 0.00469 | 0.74131 | 0.003 |
| 0.74152 | 0.00467 | 0.00575 | 0.72636 | 0.00217 |
| 0.75089 | 0.00485 | 0.00598 | 0.71793 | 0.00185 |
| 0.76119 | 0.00482 | 0.00594 | 0.70997 | 0.00323 |
| 0.77157 | 0.00535 | 0.00659 | 0.69846 | 0.00478 |
| 0.7818 | 0.00495 | 0.0061 | 0.68945 | 0.00241 |
| 0.79242 | 0.00502 | 0.00618 | 0.6775 | 0.00399 |
| 0.80236 | 5.99E-03 | 7.39E-03 | 0.6624 | 0.00435 |
| 0.81173 | 5.37E-03 | 6.62E-03 | 0.65327 | 0.00271 |
| 0.82228 | 0.00521 | 0.00642 | 0.64016 | 0.00336 |
| 0.83171 | 0.00538 | 0.00662 | 0.62617 | 0.00336 |
| 0.84134 | 0.00521 | 0.00642 | 0.60886 | 0.0028 |
| 0.852 | 0.00538 | 0.00663 | 0.59616 | 0.00364 |
| 0.86106 | 0.00556 | 0.00684 | 0.58136 | 0.00487 |
| 0.87055 | 0.00521 | 0.00642 | 0.56748 | 0.00361 |
| 0.88031 | 0.00617 | 0.00761 | 0.5545 | 0.00465 |
| 0.89058 | 0.00635 | 0.00785 | 0.5337 | 0.00679 |
| 0.90142 | 0.00628 | 0.00777 | 0.51244 | 0.00601 |

| aw | error aw (+ve) | error aw (-ve) | MFS | error MFS |
|---|---|---|---|---|
| 0.91173 | 0.00596 | 0.00737 | 0.48949 | 0.00523 |
| 0.92543 | 1.50E-04 | 2.02E-04 | 0.45608 | 4.73E-04 |
| 0.92905 | 1.56E-04 | 1.58E-04 | 0.44732 | 6.46E-04 |
| 0.94053 | 3.86E-04 | 3.86E-04 | 0.40605 | 0.00233 |
| 0.95111 | 4.52E-04 | 4.52E-04 | 0.35279 | 0.00256 |
| 0.96227 | 3.42E-04 | 3.41E-04 | 0.28818 | 0.00275 |
| 0.97153 | 3.20E-04 | 3.20E-04 | 0.20634 | 0.00243 |
| 0.98066 | 8.31E-04 | 8.09E-04 | 0.12936 | 0.00558 |

**S15 Adipic Acid Hygroscopicity**

**Figure S15.1**: Hygroscopicity of Adipic Acid, (Sigma Aldrich, Purity 99 %), at 293.15 K. Open squares, these experiments; solid line, UNIFAC model.

[Figure]

**Table S15.1:** Pure component refractive index ($n_{melt}$) is determined using molar refraction, assuming ideal mixing for calculation of the melt density ($\rho_{melt}$), from bulk data available in Cai et al. (2016). The variation of density as a function of the root of solute mass fraction ($MFS^{1/2} = x$) is represented by polynomial fit parameters. *Upper* and *lower* refer to 95 % confidence limits for fits to experimental data, (Section 2.2 in manuscript).

| | $n_{melt}$ | $\rho_{melt}/$ g.cm$^{-3}$ | Polynomial fit ( $\rho_{sol} = a + b_1x + b_2x^2 + b_3x^3 + b_4x^4 + b_5x^5 + b_6x^6$) | | | | | | |
|---|---|---|---|---|---|---|---|---|---|
| | | | a | $b_1$ | $b_2$ | $b_3$ | $b_4$ | $b_5$ | $b_6$ |
| *Best* | 1.5052 | 1.2897 | 998.2 | -0.483 | 232.81 | -36.78 | 137.06 | -96.59 | 55.48 |
| *Upper* | 1.5093 | 1.3192 | 998.2 | -0.705 | 253.36 | -53.41 | 183.61 | -139.01 | 77.14 |
| *Lower* | 1.5012 | 1.2614 | 998.2 | -0.323 | 213.1 | -24.73 | 101.55 | -65.53 | 39.14 |

**Table S15.2:** Tabulated experimental data points shown in **Fig S15.1.**

| aw | error aw (+ve) | error aw (-ve) | MFS | error MFS |
|---|---|---|---|---|
| 0.95373 | 3.22E-04 | 6.26E-04 | 0.39071 | 0.00391 |
| 0.95843 | 8.35E-04 | 0.00118 | 0.36348 | 0.00812 |
| 0.9634 | 0.0011 | 0.00133 | 0.3272 | 0.00935 |
| 0.96865 | 0.00107 | 0.00138 | 0.28685 | 0.01043 |
| 0.97365 | 9.42E-04 | 0.00127 | 0.24062 | 0.01007 |
| 0.97863 | 9.10E-04 | 0.00114 | 0.1917 | 0.00876 |
| 0.98405 | 5.88E-04 | 8.82E-04 | 0.13977 | 0.00621 |
| 0.98877 | 3.13E-04 | 4.91E-04 | 0.09086 | 0.0027 |
| 0.99423 | 1.80E-04 | 3.02E-04 | 0.04898 | 0.00153 |
| 0.99692 | 1.66E-04 | 3.62E-04 | 0.02978 | 7.74E-04 |

**S16 Pimelic Acid Hygroscopicity**

**Fig S16.1**: Hygroscopicity of Pimelic Acid, (Sigma Aldrich, Purity 99 %), at 293.15 K. Open squares, these experiments; solid line, UNIFAC model.

[Figure]

**Table S16.1:** Pure component refractive index ($n_{melt}$) is determined using molar refraction, assuming ideal mixing for calculation of the melt density ($\rho_{melt}$), from bulk data available in Cai et al. (2016). The variation of density as a function of the root of solute mass fraction ($MFS^{1/2} = x$) is represented by polynomial fit parameters. *Upper* and *lower* refer to 95 % confidence limits for fits to experimental data, (Section 2.2 in manuscript).

| | $n_{melt}$ | $\rho_{melt}/$ g.cm$^{-3}$ | Polynomial fit ( $\rho_{sol} = a + b_1x + b_2x^2 + b_3x^3 + b_4x^4 + b_5x^5 + b_6x^6$) | | | | | | |
| --- | --- | --- | --- | --- | --- | --- | --- | --- | --- |
| | | | a | $b_1$ | $b_2$ | $b_3$ | $b_4$ | $b_5$ | $b_6$ |
| *Best* | 1.4917 | 1.2262 | 998.5 | -0.184 | 188.18 | -14.19 | 67.86 | -37.91 | 23.94 |
| *Upper* | 1.4940 | 1.2435 | 998.5 | -0.246 | 200.41 | -18.89 | 83.16 | -50.18 | 30.74 |
| *Lower* | 1.4894 | 1.2095 | 998.5 | -0.136 | 176.23 | -10.52 | 55.25 | -28.29 | 18.47 |

**Table S16.2:** Tabulated experimental data points shown in **Fig S16.1.**

| $a_w$ | error $a_w$ (+ve) | error $a_w$ (-ve) | MFS | error MFS |
| --- | --- | --- | --- | --- |
| 0.84466 | 0.00317 | 0.00251 | 0.73296 | 0.0087 |
| 0.87279 | 0.00711 | 0.00413 | 0.70863 | 0.02048 |
| 0.88863 | 0.00919 | 0.00916 | 0.67508 | 0.03342 |
| 0.90517 | 0.00585 | 0.00361 | 0.65697 | 0.02274 |
| 0.92985 | 0.00504 | 0.00334 | 0.57632 | 0.01441 |
| 0.9503 | 0.00434 | 0.00304 | 0.48806 | 0.01436 |
| 0.97207 | 0.0019 | 0.00139 | 0.29782 | 0.01787 |
| 0.99347 | 2.49E-04 | 3.32E-04 | 0.06002 | 0.00268 |

**S17 Methyl Malonic Acid Hygroscopicity**

**Fig S17.1**: Hygroscopicity of methyl malonic acid, (Sigma Aldrich, Purity 99 %), at 293.15 K. Open squares, these experiments; solid line, UNIFAC model.

[Figure]

**Table S17.1:** Pure component refractive index ($n_{melt}$) is determined using molar refraction, assuming ideal mixing for calculation of the melt density ($\rho_{melt}$), from bulk data available in Cai et al. (2016). The variation of density as a function of the root of solute mass fraction ($MFS^{1/2} = x$) is represented by polynomial fit parameters. *Upper* and *lower* refer to 95 % confidence limits for fits to experimental data, (Section 2.2 in manuscript).

| | $n_{melt}$ | $\rho_{melt}/$ $g.cm^{-3}$ | \multicolumn{7}{c}{Polynomial fit ( $\rho_{sol} = a + b_1x + b_2x^2 + b_3x^3 + b_4x^4 + b_5x^5 + b_6x^6$)} |
| | | | a | $b_1$ | $b_2$ | $b_3$ | $b_4$ | $b_5$ | $b_6$ |
|---|---|---|---|---|---|---|---|---|---|
| *Best* | 1.4817 | 1.3876 | 998.8 | -1.45 | 301.28 | -108.73 | 330.65 | -279.56 | 146.61 |
| *Upper* | 1.4819 | 1.3902 | 998.8 | -1.49 | 303.18 | -111.53 | 337.82 | -286.56 | 149.98 |
| *Lower* | 1.4815 | 1.3851 | 998.8 | -1.42 | 299.45 | -106.09 | 323.86 | -272.94 | 143.43 |

**Table S17.2:** Tabulated experimental data points shown in **Fig S17.1.**

| $a_w$ | error $a_w$ (+ve) | error $a_w$ (-ve) | MFS | error MFS |
|---|---|---|---|---|
| 0.71493 | 0.002 | 0.00248 | 0.62219 | 0.00155 |
| 0.75141 | 0.00657 | 0.00657 | 0.59428 | 0.00609 |
| 0.78527 | 0.0084 | 0.0084 | 0.562 | 0.00836 |
| 0.81777 | 0.004 | 0.00245 | 0.52434 | 0.00364 |
| 0.84355 | 0.00409 | 0.00369 | 0.49609 | 0.00573 |
| 0.875 | 0.00438 | 0.00401 | 0.44143 | 0.00784 |
| 0.90462 | 0.00402 | 0.00333 | 0.3774 | 0.00875 |
| 0.93201 | 0.00335 | 0.00317 | 0.29413 | 0.01184 |
| 0.96865 | 8.29E-04 | 8.90E-04 | 0.16472 | 0.0041 |
| 0.98911 | 4.09E-04 | 4.11E-04 | 0.05691 | 0.00203 |

**S18 Methyl Succinic Acid Hygroscopicity**

**Figure S18.1**: Hygroscopicity of methyl succinic acid, (Sigma Aldrich, Purity 99 %), at 293.15 K. Open squares, these experiments; solid line, UNIFAC model.

[Figure]

**Table S18.1:** Pure component refractive index ($n_{melt}$) is determined using molar refraction, assuming ideal mixing for calculation of the melt density ($\rho_{melt}$), from bulk data available in Cai et al. (2016). The variation of density as a function of the root of solute mass fraction ($MFS^{1/2} = x$) is represented by polynomial fit parameters. *Upper* and *lower* refer to 95 % confidence limits for fits to experimental data, (Section 2.2 in manuscript).

| | $n_{melt}$ | $\rho_{melt}/$ $g.cm^{-3}$ | Polynomial fit ( $\rho_{sol} = a + b_1x + b_2x^2 + b_3x^3 + b_4x^4 + b_5x^5 + b_6x^6$) | | | | | | |
|---|---|---|---|---|---|---|---|---|---|
| | | | a | $b_1$ | $b_2$ | $b_3$ | $b_4$ | $b_5$ | $b_6$ |
| *Best* | 1.4779 | 1.3035 | 998.2 | -0.572 | 242.3 | -43.51 | 156.55 | -114.16 | 64.69 |
| *Upper* | 1.4784 | 1.3090 | 998.2 | -0.614 | 246.13 | -46.62 | 165.26 | -122.12 | 68.76 |
| *Lower* | 1.4774 | 1.2980 | 998.2 | -0.533 | 238.48 | -40.56 | 148.19 | -106.58 | 60.79 |

**Table S18.2:** Tabulated experimental data points shown in **Fig S18.1.**

| $a_w$ | error $a_w$ (+ve) | error $a_w$ (-ve) | MFS | error MFS |
|---|---|---|---|---|
| 0.66772 | 0.00345 | 0.00424 | 0.76125 | 0.00296 |
| 0.71234 | 0.0134 | 0.0134 | 0.72476 | 0.01132 |
| 0.7636 | 0.01237 | 0.01517 | 0.67596 | 0.00785 |
| 0.80135 | 0.00451 | 0.00326 | 0.65118 | 0.00447 |
| 0.83951 | 0.00575 | 0.00629 | 0.59855 | 0.01151 |
| 0.87778 | 0.00688 | 0.00657 | 0.52839 | 0.01469 |
| 0.92249 | 0.00567 | 0.00567 | 0.40343 | 0.01891 |
| 0.96705 | 0.00282 | 0.00249 | 0.19484 | 0.01368 |
| 0.99344 | 3.28E-04 | 3.47E-04 | 0.03075 | 0.00168 |

**S19 Binary Aqueous Diethylmalonic Acid - Hygroscopicity**

**Fig S19.1**: Hygroscopicity of diethylmalonic acid, (Sigma Aldrich, Purity 98 %), at 293.15 K. Open squares, these experiments; solid line, UNIFAC model.

[Figure]

**Table S19.1:** Pure component refractive index ($n_{melt}$) is determined using molar refraction, assuming ideal mixing for calculation of the melt density ($\rho_{melt}$), from bulk data available in Cai et al. (2016). The variation of density as a function of the root of solute mass fraction (MFS$^{1/2}$ =x) is represented by polynomial fit parameters. *Upper* and *lower* refer to 95 % confidence limits for fits to experimental data, (Section 2.2 in manuscript).

| | $n_{melt}$ | $\rho_{melt}/$ g.cm$^{-3}$ | Polynomial fit ( $\rho_{sol} = a + b_1x + b_2x^2 + b_3x^3 + b_4x^4 + b_5x^5 + b_6x^6$) | | | | | | |
|---|---|---|---|---|---|---|---|---|---|
| | | | a | $b_1$ | $b_2$ | $b_3$ | $b_4$ | $b_5$ | $b_6$ |
| *Best* | 1.4854 | 1.2184 | 998.2 | -0.161 | 182.82 | -12.45 | 61.98 | -33.36 | 21.37 |
| *Upper* | 1.4858 | 1.2219 | 998.2 | -0.172 | 185.32 | -13.25 | 64.69 | -35.44 | 22.55 |
| *Lower* | 1.4850 | 1.2149 | 998.2 | -0.151 | 180.32 | -11.69 | 59.36 | -31.37 | 20.24 |

**Table S19.2:** Tabulated experimental data points shown in **Figure S19.1.**

| $a_w$ | error $a_w$ (+ve) | error $a_w$ (-ve) | MFS | error MFS |
|---|---|---|---|---|
| 0.68895 | 0.00441 | 0.00543 | 0.79565 | 0.00315 |
| 0.70762 | 0.01 | 0.01233 | 0.78448 | 0.00548 |
| 0.7737 | 0.01156 | 0.01425 | 0.7484 | 0.00901 |
| 0.83916 | 0.01287 | 0.00773 | 0.70902 | 0.01617 |
| 0.84654 | 0.00329 | 0.00246 | 0.70885 | 0.00435 |
| 0.86832 | 0.00637 | 0.0062 | 0.68324 | 0.01847 |
| 0.88499 | 0.00646 | 0.00418 | 0.65203 | 0.0186 |
| 0.90928 | 0.00665 | 0.00391 | 0.62123 | 0.00847 |
| 0.93317 | 0.00665 | 0.00374 | 0.56028 | 0.00907 |
| 0.95177 | 0.00586 | 0.00329 | 0.48861 | 0.01646 |
| 0.97321 | 0.00199 | 0.00152 | 0.28968 | 0.01912 |
| 0.99422 | 3.23E-04 | 3.66E-04 | 0.02697 | 0.00157 |

**S20 2,2-Dimethyl Glutaric Acid Hygroscopicity**

**Fig S20.1**: Hygroscopicity of 2,2-dimethyl glutaric acid, (Sigma Aldrich, Purity > 98 %), at 293.15 K. Open squares, these experiments; solid line, UNIFAC model.

[Figure]

**Table S20.1:** Pure component refractive index ($n_{melt}$) is determined using molar refraction, assuming ideal mixing for calculation of the melt density ($\rho_{melt}$), from bulk data available in Cai et al. (2016). The variation of density as a function of the root of solute mass fraction ($MFS^{1/2} = x$) is represented by polynomial fit parameters. *Upper* and *lower* refer to 95 % confidence limits for fits to experimental data, (Section 2.2 in manuscript).

| | $n_{melt}$ | $\rho_{melt}/$ $g.cm^{-3}$ | Polynomial fit ( $\rho_{sol} = a + b_1x + b_2x^2 + b_3x^3 + b_4x^4 + b_5x^5 + b_6x^6$) | | | | | | |
| --- | --- | --- | --- | --- | --- | --- | --- | --- | --- |
| | | | a | $b_1$ | $b_2$ | $b_3$ | $b_4$ | $b_5$ | $b_6$ |
| *Best* | 1.4881 | 1.2225 | 998.2 | -0.174 | 185.75 | -13.39 | 65.16 | -35.81 | 22.76 |
| *Upper* | 1.4884 | 1.2248 | 998.2 | -0.181 | 187.39 | -13.93 | 67 | -37.24 | 23.57 |
| *Lower* | 1.4878 | 1.2201 | 998.2 | -0.166 | 184.04 | -12.83 | 63.28 | -34.36 | 21.94 |

**Table S19.2:** Tabulated experimental data points shown in **Fig S20.1.**

| $a_w$ | error $a_w$ (+ve) | error $a_w$ (-ve) | MFS | error MFS |
| --- | --- | --- | --- | --- |
| 0.66522 | 0.00707 | 0.00713 | 0.87406 | 0.00722 |
| 0.71105 | 0.00493 | 0.00494 | 0.8677 | 0.00654 |
| 0.74996 | 0.00758 | 0.00758 | 0.83334 | 0.01058 |
| 0.79488 | 0.01337 | 0.01338 | 0.80256 | 0.01126 |
| 0.84249 | 0.00573 | 0.00389 | 0.76365 | 0.00522 |
| 0.86987 | 0.00563 | 0.00574 | 0.73768 | 0.00728 |
| 0.91262 | 0.00592 | 0.00605 | 0.65854 | 0.01692 |
| 0.95695 | 0.00508 | 0.00491 | 0.48805 | 0.02723 |
| 0.99362 | 3.59E-04 | 3.74E-04 | 0.05685 | 0.00348 |

**S21 2,2-Dimethyl Succinic Acid Hygroscopicity**

**Fig S21.1**: Hygroscopicity of 2,2-dimethyl succinic acid, (Sigma Aldrich, Purity 99 %), at 293.15 K. Open squares, these experiments; solid line, UNIFAC model.

[Figure]

**Table S21.1:** Pure component refractive index ($n_{melt}$) is determined using molar refraction, assuming ideal mixing for calculation of the melt density ($\rho_{melt}$), from bulk data available in Cai et al. (2016). The variation of density as a function of the root of solute mass fraction (MFS$^{1/2}$ =x) is represented by polynomial fit parameters. *Upper* and *lower* refer to 95 % confidence limits for fits to experimental data, (Section 2.2 in manuscript).

|  | $n_{melt}$ | $\rho_{melt}/$ $g.cm^{-3}$ | Polynomial fit ( $\rho_{sol} = a + b_1x + b_2x^2 + b_3x^3 + b_4x^4 + b_5x^5 + b_6x^6$) | | | | | | |
|---|---|---|---|---|---|---|---|---|---|
|  |  |  | a | $b_1$ | $b_2$ | $b_3$ | $b_4$ | $b_5$ | $b_6$ |
| *Best* | 1.4889 | 1.2710 | 997.9 | -0.382 | 220.13 | -29.13 | 114.09 | -76.29 | 44.68 |
| *Upper* | 1.4897 | 1.2776 | 997.9 | -0.419 | 224.73 | -31.96 | 122.4 | -83.53 | 48.48 |
| *Lower* | 1.4880 | 1.2644 | 997.9 | -0.347 | 215.51 | -26.48 | 106.23 | -69.5 | 41.09 |

**Table S21.2:** Tabulated experimental data points shown in **Fig S21.1**

| $a_w$ | error $a_w$ (+ve) | error $a_w$ (-ve) | MFS | error MFS |
|---|---|---|---|---|
| 0.6713 | 0.00663 | 0.00663 | 0.78921 | 0.00655 |
| 0.73389 | 0.02256 | 0.02256 | 0.74829 | 0.01579 |
| 0.77076 | 0.00564 | 0.00705 | 0.73908 | 0.00579 |
| 0.84308 | 0.00747 | 0.00776 | 0.67413 | 0.01818 |
| 0.88089 | 0.00536 | 0.00529 | 0.60846 | 0.01212 |
| 0.9367 | 0.00425 | 0.00424 | 0.41751 | 0.01893 |
| 0.99244 | 4.24E-04 | 5.93E-04 | 0.05911 | 0.00313 |

**S22 2-Methyl Glutaric Acid Hygroscopicity**

**Fig S22.1**: Hygroscopicity of 2-methyl glutaric acid, (Sigma Aldrich, Purity 98 %), at 293.15 K. Open squares, these experiments; solid line, UNIFAC model.

[Figure]

**Table S22.1:** Pure component refractive index ($n_{melt}$) is determined using molar refraction, assuming ideal mixing for calculation of the melt density ($\rho_{melt}$), from bulk data available in Cai et al. (2016). The variation of density as a function of the root of solute mass fraction ($MFS^{1/2} = x$) is represented by polynomial fit parameters. *Upper* and *lower* refer to 95 % confidence limits for fits to experimental data, (Section 2.2 in manuscript).

| | $n_{melt}$ | $\rho_{melt}/$ $g.cm^{-3}$ | Polynomial fit ( $\rho_{sol} = a + b_1x + b_2x^2 + b_3x^3 + b_4x^4 + b_5x^5 + b_6x^6$ ) | | | | | | |
|---|---|---|---|---|---|---|---|---|---|
| | | | a | $b_1$ | $b_2$ | $b_3$ | $b_4$ | $b_5$ | $b_6$ |
| *Best* | 1.4866 | 1.2585 | 997.6 | -0.319 | 211.59 | -24.4 | 99.95 | -64.16 | 38.24 |
| *Upper* | 1.4873 | 1.2648 | 997.6 | -0.350 | 216 | -26.78 | 107.1 | -70.26 | 41.49 |
| *Lower* | 1.4858 | 1.2522 | 997.6 | -0.290 | 207.17 | -22.18 | 93.17 | -58.44 | 35.17 |

**Table S22.2:** Tabulated experimental data points shown in **Fig S22.1**

| $a_w$ | error $a_w$ (+ve) | error $a_w$ (-ve) | MFS | error MFS |
|---|---|---|---|---|
| 0.68925 | 0.00271 | 0.00334 | 0.80479 | 0.00208 |
| 0.72204 | 0.01005 | 0.01239 | 0.78383 | 0.00857 |
| 0.76123 | 0.01296 | 0.01422 | 0.75567 | 0.01704 |
| 0.78959 | 0.02339 | 0.02377 | 0.73478 | 0.02713 |
| 0.82836 | 0.01185 | 0.00726 | 0.70077 | 0.02018 |
| 0.84699 | 0.00634 | 0.00601 | 0.68658 | 0.01104 |
| 0.8785 | 0.00611 | 0.00622 | 0.63205 | 0.01527 |
| 0.91076 | 0.00612 | 0.00583 | 0.54437 | 0.02194 |
| 0.94004 | 0.00438 | 0.00438 | 0.4312 | 0.02071 |
| 0.98128 | 4.79E-04 | 0.0012 | 0.14884 | 0.0113 |
| 0.99285 | 2.24E-04 | 2.25E-04 | 0.02928 | 0.00106 |

**S23 3-Methyl Adipic Acid Hygroscopicity**

**Fig S23.1**: Hygroscopicity of 3-methyl adipic acid, (Sigma Aldrich, Purity 99 %), at 293.15 K. Open squares, these experiments; solid line, UNIFAC model.

[Figure]

**Table S23.1:** Pure component refractive index ($n_{melt}$) is determined using molar refraction, assuming ideal mixing for calculation of the melt density ($\rho_{melt}$), from bulk data available in Cai et al. (2016). The variation of density as a function of the root of solute mass fraction ($MFS^{1/2} = x$) is represented by polynomial fit parameters. *Upper* and *lower* refer to 95 % confidence limits for fits to experimental data, (Section 2.2 in manuscript).

| | $n_{melt}$ | $\rho_{melt}/$ $g.cm^{-3}$ | Polynomial fit ( $\rho_{sol} = a + b_1x + b_2x^2 + b_3x^3 + b_4x^4 + b_5x^5 + b_6x^6$) | | | | | | |
|---|---|---|---|---|---|---|---|---|---|
| | | | a | $b_1$ | $b_2$ | $b_3$ | $b_4$ | $b_5$ | $b_6$ |
| *Best* | 1.4865 | 1.2141 | 999.0 | -0.147 | 179.19 | -11.33 | 58.11 | -30.42 | 19.69 |
| *Upper* | 1.4878 | 1.2243 | 999.0 | -0.176 | 186.48 | -13.59 | 65.86 | -36.34 | 23.06 |
| *Lower* | 1.4852 | 1.2041 | 999.0 | -0.121 | 171.99 | -9.4 | 51.21 | -25.33 | 16.75 |

**Table S23.2:** Tabulated experimental data points shown in **Fig S23.1.**

| $a_w$ | error $a_w$ (+ve) | error $a_w$ (-ve) | MFS | error MFS |
|---|---|---|---|---|
| 0.71902 | 0.00897 | 0.00897 | 0.80154 | 0.00624 |
| 0.78015 | 0.03348 | 0.03347 | 0.7615 | 0.02149 |
| 0.82646 | 0.00615 | 0.00574 | 0.73848 | 0.00556 |
| 0.88266 | 0.00886 | 0.00907 | 0.67532 | 0.02097 |
| 0.92986 | 0.00748 | 0.00771 | 0.54185 | 0.02748 |
| 0.993 | 2.61E-04 | 3.72E-04 | 0.06527 | 0.00354 |

**S24 3-Methyl Glutaric Acid Hygroscopicity**

**Fig S24.1**: Hygroscopicity of 3-methyl glutaric acid, (Sigma Aldrich, Purity 99 %), at 293.15 K. Open squares, these experiments; solid line, UNIFAC model.

[Figure]

**Table SI.24.1:** Pure component refractive index ($n_{melt}$) is determined using molar refraction, assuming ideal mixing for calculation of the melt density ($\rho_{melt}$), from bulk data available in Cai et al. (2016). The variation of density as a function of the root of solute mass fraction (MFS$^{1/2}$ =x) is represented by polynomial fit parameters. *Upper* and *lower* refer to 95 % confidence limits for fits to experimental data, (Section 2.2 in manuscript).

| | $n_{melt}$ | $\rho_{melt}/$ g.cm$^{-3}$ | Polynomial fit ( $\rho_{sol} = a + b_1x + b_2x^2 + b_3x^3 + b_4x^4 + b_5x^5 + b_6x^6$) | | | | | | |
|---|---|---|---|---|---|---|---|---|---|
| | | | a | $b_1$ | $b_2$ | $b_3$ | $b_4$ | $b_5$ | $b_6$ |
| *Best* | 1.4819 | 1.2498 | 997.9 | -0.277 | 205.29 | -21.26 | 90.32 | -56.07 | 33.89 |
| *Upper* | 1.4822 | 1.2531 | 997.9 | -0.292 | 207.6 | -22.37 | 93.74 | -58.92 | 35.43 |
| *Lower* | 1.4816 | 1.2466 | 997.9 | -0.264 | 203.04 | -20.22 | 87.1 | -53.39 | 32.44 |

**Table S24.2:** Tabulated experimental data points shown in **Fig S24.1.**

| $a_w$ | error $a_w$ (+ve) | error $a_w$ (-ve) | MFS | error MFS |
|---|---|---|---|---|
| 0.69173 | 0.00299 | 0.00334 | 0.79013 | 0.0038 |
| 0.74649 | 0.00642 | 0.00683 | 0.76025 | 0.00932 |
| 0.81013 | 0.01887 | 0.01884 | 0.70959 | 0.02367 |
| 0.86283 | 0.00343 | 0.00213 | 0.63276 | 0.00618 |
| 0.89131 | 0.00283 | 0.00283 | 0.58884 | 0.00675 |
| 0.95411 | 0.00246 | 0.00245 | 0.3472 | 0.01394 |
| 0.98567 | 6.06E-04 | 6.09E-04 | 0.10123 | 0.00477 |

**S25 3, 3-Dimethyl Glutaric Acid Hygroscopicity**

**Fig S25.1**: Hygroscopicity of 3, 3-dimethyl glutaric acid, (Sigma Aldrich, Purity 98 %), at 293.15 K. Open squares, these experiments; solid line, UNIFAC model.

[Figure]

**Table S25.1:** Pure component refractive index ($n_{melt}$) is determined using molar refraction, assuming ideal mixing for calculation of the melt density ($\rho_{melt}$), from bulk data available in Cai et al. (2016). The variation of density as a function of the root of solute mass fraction ($MFS^{1/2} = x$) is represented by polynomial fit parameters. *Upper* and *lower* refer to 95 % confidence limits for fits to experimental data, (Section 2.2 in manuscript).

| | $n_{melt}$ | $\rho_{melt}/$ $g.cm^{-3}$ | Polynomial fit ( $\rho_{sol} = a + b_1x + b_2x^2 + b_3x^3 + b_4x^4 + b_5x^5 + b_6x^6$) | | | | | | |
| | | | a | $b_1$ | $b_2$ | $b_3$ | $b_4$ | $b_5$ | $b_6$ |
|---|---|---|---|---|---|---|---|---|---|
| *Best* | 1.4903 | 1.2206 | 998.3 | -0.167 | 184.33 | -12.92 | 63.58 | -34.59 | 22.07 |
| *Upper* | 1.4906 | 1.2231 | 998.3 | -0.175 | 186.11 | -13.5 | 65.55 | -36.11 | 22.93 |
| *Lower* | 1.4900 | 1.2182 | 998.3 | -0.160 | 182.61 | -12.38 | 61.74 | -33.18 | 21.27 |

**Table S25.2:** Tabulated experimental data points shown in **Fig S25.1.**

| $a_w$ | error $a_w$ (+ve) | error $a_w$ (-ve) | MFS | error MFS |
|---|---|---|---|---|
| 0.71132 | 0.00345 | 0.00345 | 0.85176 | 0.00384 |
| 0.76078 | 0.006 | 0.00743 | 0.80912 | 0.00721 |
| 0.79151 | 0.01941 | 0.01942 | 0.79788 | 0.01562 |
| 0.83444 | 0.00416 | 0.00451 | 0.75169 | 0.00421 |
| 0.87055 | 0.00543 | 0.00565 | 0.71882 | 0.0105 |
| 0.91582 | 0.00545 | 0.00564 | 0.61641 | 0.02163 |
| 0.96018 | 0.00389 | 0.00389 | 0.39161 | 0.02576 |
| 0.99443 | 2.18E-04 | 2.83E-04 | 0.04485 | 0.00225 |

**S26. PEG3 Hygroscopicity**

**Fig S26.1**: Hygroscopicity of PEG3, at 293.15 K. Open squares, these experiments; solid line, UNIFAC model.

[Figure]

**Table S26.1:** Measured values of pure component melt density ($\rho_{melt}$) and refractive index ($n_{melt}$) (PEG3 is liquid), presented with parameterisation for solution measurements of density where x is the square root of MFS ($MFS^{1/2} = x$). *Upper* and *lower* refer to 95 % confidence limits for fits to experimental data. Upper and lower limit on refractive index and density are determined by the error in the refractometer and by the densitometer respectively.

|  | $n_{melt}$ | $\rho_{melt}$/ g.cm$^{-3}$ | Polynomial fit ( $\rho_{sol} = a + b_1x + b_2x^2 + b_3x^3$) | | | |
|---|---|---|---|---|---|---|
|  | | | a | $b_1$ | $b_2$ | $b_3$ |
| *Best* | 1.4551 | 1.109 | 999.97 | -75.75 | 431.63 | -246.73 |
| *Upper* | 1.4552 | 1.122 | 999.97 | -0.198 | 268.11 | -144.15 |
| *Lower* | 1.4550 | 1.096 | 999.97 | -151.31 | 595.15 | -349.31 |

**Table S26.2:** Tabulated experimental data points shown in **Fig S26.1.**

| $a_w$ | error $a_w$ (+ve) | error $a_w$ (-ve) | MFS | error MFS |
|---|---|---|---|---|
| 0.524 | 0.0024 | 0.00286 | 0.83232 | 0.00127 |
| 0.61806 | 0.02008 | 0.02389 | 0.77269 | 0.0098 |
| 0.65597 | 0.00198 | 0.00242 | 0.72923 | 0.00152 |
| 0.69291 | 0.00856 | 0.00856 | 0.69867 | 0.0088 |
| 0.75489 | 8.16E-03 | 0.01 | 0.63211 | 7.82E-03 |
| 0.81001 | 0.0263 | 0.0263 | 0.56113 | 0.03211 |
| 0.84347 | 0.00123 | 0.00119 | 0.49753 | 0.00229 |
| 0.89472 | 0.00416 | 0.00414 | 0.39303 | 0.01004 |
| 0.95087 | 3.07E-03 | 3.07E-03 | 0.22603 | 0.01048 |
| 0.97688 | 0.00201 | 0.00112 | 0.12742 | 0.00393 |

**S27. PEG4 Hygroscopicity**

**Fig S27.1**: Hygroscopicity of PEG4, at 293.15 K. Open squares, these CC-EDB experiments; solid line, UNIFAC model; blue line UManSysProp; red line adsorption isotherm model from Dutcher.

[Figure]

**Table S27.1:** Measured values of pure component melt density ($\rho_{melt}$) and refractive index ($n_{melt}$) (PEG4 is liquid), presented with parameterisation for solution measurements of density where x is the square root of MFS ($MFS^{1/2} =x$). *Upper* and *lower* refer to 95 % confidence limits for fits to experimental data. Upper and lower limit on refractive index and density are determined by the error in the refractometer and by the densitometer respectively.

| | $n_{melt}$ | $\rho_{melt}$/ g.cm$^{-3}$ | Polynomial fit ( $\rho_{sol} = a + b_1x + b_2x^2 + b_3x^3$) | | | |
|---|---|---|---|---|---|---|
| | | | a | $b_1$ | $b_2$ | $b_3$ |
| *Best* | 1.4589 | 1.1271 | 999.97 | -37.39 | 296.85 | -130.68 |
| *Upper* | 1.4590 | 1.13412 | 999.97 | -9.65 | 235.84 | -92.25 |
| *Lower* | 1.4588 | 1.12338 | 999.97 | -65.13 | 357.86 | -169.11 |

**Table S27.2:** Tabulated experimental data points shown in **Fig S27.1.**

| $a_w$ | error $a_w$ (+ve) | error $a_w$ (-ve) | MFS | error MFS |
|---|---|---|---|---|
| 0.52052 | 0.00336 | 0.00399 | 0.83006 | 8.065E-4 |
| 0.60966 | 0.00229 | 0.00278 | 0.78149 | 8.220E-4 |
| 0.65636 | 0.00166 | 0.00204 | 0.74058 | 0.00177 |
| 0.69735 | 0.00172 | 0.00212 | 0.71195 | 0.00154 |
| 0.74132 | 0.00556 | 0.00685 | 0.67145 | 0.00929 |
| 0.78212 | 6.975E-4 | 8.803E-4 | 0.63073 | 8.501E-4 |
| 0.81258 | 0.00536 | 0.00535 | 0.58791 | 0.00759 |
| 0.84243 | 0.00132 | 0.00111 | 0.52225 | 0.00213 |
| 0.89453 | 0.00427 | 0.00448 | 0.42827 | 0.01048 |
| 0.93766 | 0.00385 | 0.00376 | 0.31263 | 0.01217 |
| 0.98571 | 0.0013 | 0.00127 | 0.0918 | 0.00662 |
| 0.99969 | 0.00143 | 0.00156 | 0.0475 | 0.00252 |

**S28 Erythritol Hygroscopicity**

**Fig S28.1**: Hygroscopicity of erythritol (Sigma Aldrich 99 %), at 293.15 K. Open squares, these experiments; solid line, UNIFAC model.

[Figure]

**Table S28.1:** Pure component refractive index ($n_{melt}$) is determined using molar refraction, assuming ideal mixing for calculation of the melt density ($\rho_{melt}$), from bulk data available in Cai et al. (2016). The variation of density as a function of the root of solute mass fraction (MFS$^{1/2}$ =x) is represented by polynomial fit parameters. *Upper* and *lower* refer to 95 % confidence limits for fits to experimental data, (Section 2.2 in manuscript).

|  | $n_{melt}$ | $\rho_{melt}$/ g.cm$^{-3}$ | Polynomial fit ( $\rho_{sol} = a + b_1x + b_2x^2 + b_3x^3$) | | | |
|---|---|---|---|---|---|---|
|  |  |  | a | $b_1$ | $b_2$ | $b_3$ |
| *Best* | 1.5211 | 1.3754 | 998.6 | 58.46 | 37.98 | 278.66 |
| *Upper* | 1.5388 | 1.3813 | 998.6 | 60.21 | 33.79 | 286.94 |
| *Lower* | 1.5204 | 1.3695 | 998.6 | 56.75 | 42.03 | 27.049 |

**Table S28.2:** Tabulated experimental data points shown in **Fig S28.1.**

| $a_w$ | error $a_w$ (+ve) | error $a_w$ (-ve) | MFS | error MFS |
|---|---|---|---|---|
| 0.62602 | 8.77112E-4 | 0.00107 | 0.71188 | 6.08334E-4 |
| 0.66147 | 0.0027 | 0.0033 | 0.66395 | 0.00226 |
| 0.72692 | 0.00104 | 0.00129 | 0.6342 | 6.57702E-4 |
| 0.75582 | 0.00775 | 0.00777 | 0.60723 | 0.00739 |
| 0.78929 | 0.01009 | 0.0101 | 0.57499 | 0.01315 |
| 0.83827 | 7.77253E-4 | 0.001 | 0.50705 | 9.72437E-4 |
| 0.86916 | 7.13427E-4 | 6.96511E-4 | 0.46004 | 0.00138 |
| 0.93195 | 2.64028E-4 | 3.52642E-4 | 0.28987 | 0.00175 |
| 0.95145 | 7.53773E-4 | 7.52526E-4 | 0.20621 | 0.00312 |
| 0.9815 | 5.76107E-4 | 5.56581E-4 | 0.13503 | 0.00279 |

**S29 Sorbitol Hygroscopicity**

**Fig S29.1**: Hygroscopicity of sorbitol (Sigma Aldrich $\geq$ 98 %), at 293.15 K. Open squares, these experiments; solid line, UNIFAC model. Data taken at RHs lower than indicated by the dashed black line show increased error in hygroscopicity retrieval due to the imposition of a kinetic limitation on water transport.

[Figure]

**Table S29.1:** Pure component refractive index ($n_{melt}$) determined using molar refraction where the melt density ($\rho_{melt}$) is determined using a polynomial fit of density to the square root of MFS (MFS$^{1/2}$ =x). Bulk values used are available in Cai et al. (2016). *Upper* and *lower* refer to 95 % confidence limits for fits to experimental data.

|  | $n_{melt}$ | $\rho_{melt}$/ g.cm$^{-3}$ | Polynomial fit ( $\rho_{sol} = a + b_1x + b_2x^2 + b_3x^3$) | | | |
|---|---|---|---|---|---|---|
|  |  |  | a | $b_1$ | $b_2$ | $b_3$ |
| *Best* | 1.5244 | 1.4231 | 997.8 | 8.6 | 286.1 | 130.7 |
| *Upper* | 1.5267 | 1.4333 | 997.8 | 24.74 | 234.56 | 175.54 |
| *Lower* | 1.5220 | 1.4128 | 997.8 | -7.6 | 337.59 | 85.83 |

**Table S29.2:** Tabulated experimental data points shown in **Fig S29.1.**

| $a_w$ | error $a_w$ (+ve) | error $a_w$ (-ve) | MFS | error MFS |
|---|---|---|---|---|
| 0.50647 | 0.00432 | 0.00512 | 0.78667 | 0.00341 |
| 0.52291 | 0.0031 | 0.00369 | 0.7771 | 0.00307 |
| 0.54873 | 0.00705 | 0.00838 | 0.74672 | 0.00731 |
| 0.59535 | 0.00322 | 0.00389 | 0.73916 | 0.00193 |
| 0.60809 | 0.0019 | 0.0023 | 0.74976 | 0.00343 |
| 0.63682 | 0.00605 | 0.00728 | 0.70773 | 0.01216 |
| 0.67255 | 0.00497 | 0.00601 | 0.69271 | 0.00773 |
| 0.7035 | 0.00148 | 0.00183 | 0.69648 | 0.00163 |
| 0.73531 | 0.00619 | 0.00619 | 0.66608 | 0.00694 |
| 0.75896 | 0.00493 | 0.00599 | 0.62673 | 0.00941 |
| 0.78492 | 0.00775 | 0.00958 | 0.61901 | 0.00237 |
| 0.83722 | 0.00384 | 0.0025 | 0.56241 | 9.55991E-4 |
| 0.85049 | 9.622E-4 | 8.165E-4 | 0.5556 | 0.00118 |
| 0.88386 | 0.00262 | 0.0027 | 0.51154 | 0.00629 |
| 0.91574 | 0.00253 | 0.00266 | 0.4286 | 0.0076 |
| 0.94681 | 0.00245 | 0.00245 | 0.30429 | 0.01053 |
| 0.97555 | 0.0014 | 0.00139 | 0.14769 | 0.00774 |
| 0.99655 | 0.00112 | 6.78573E-4 | 0.02751 | 0.00293 |

**S30 D-(+)-Trehalose Dihydrate Hygroscopicity**

**Fig S30.1**: Hygroscopicity of D-(+)-trehalose dihydrate (Sigma Aldrich ≥ 99 %), at 293.15 K. Open squares, these experiments; solid line, UNIFAC model. Data taken at RHs lower than indicated by the dashed black line show increased error in hygroscopicity retrieval due to the imposition of a kinetic limitation on water transport.

[Figure]

**Table S30.1:** Pure component refractive index ($n_{melt}$) determined using molar refraction where the melt density ($\rho_{melt}$) is determined using a polynomial fit of density to the square root of MFS ($MFS^{1/2}$ =x). Bulk values used are available in Cai et al. (2016). *Upper* and *lower* refer to 95 % confidence limits for fits to experimental data.

|  | $n_{melt}$ | $\rho_{melt}$/ g.cm$^{-3}$ | Polynomial fit ( $\rho_{sol} = a + b_1x + b_2x^2 + b_3x^3$) | | | |
|---|---|---|---|---|---|---|
|  |  |  | a | $b_1$ | $b_2$ | $b_3$ |
| *Best* | 1.5193 | 1.4682 | 997.8 | 8.2 | 284.3 | 177.8 |
| *Upper* | 1.5211 | 1.4734 | 997.8 | 11.6 | 269.79 | 194.19 |
| *Lower* | 1.5175 | 1.4629 | 997.8 | 4.87 | 298.84 | 161.43 |

**Table S30.2:** Tabulated experimental data points shown in **Fig S30.1.**

| $a_w$ | error $a_w$ (+ve) | error $a_w$ (-ve) | MFS | error MFS |
|---|---|---|---|---|
| 0.51123 | 0.00397 | 0.0047 | 0.8511 | 0.00561 |
| 0.54636 | 0.01007 | 0.01196 | 0.81364 | 0.01816 |
| 0.5873 | 0.007 | 0.00844 | 0.85121 | 0.00732 |
| 0.60689 | 0.00263 | 0.00319 | 0.84879 | 0.00386 |
| 0.63303 | 0.01031 | 0.01244 | 0.79889 | 0.02031 |
| 0.67154 | 0.00716 | 0.00861 | 0.76858 | 0.00766 |
| 0.70479 | 0.00212 | 0.00262 | 0.80977 | 0.00199 |
| 0.72437 | 0.00577 | 0.00642 | 0.78413 | 0.00669 |
| 0.76384 | 0.01102 | 0.01364 | 0.743 | 0.00611 |
| 0.79679 | 0.00422 | 0.00225 | 0.7399 | 0.01219 |
| 0.81122 | 0.00282 | 0.00195 | 0.73624 | 0.0059 |
| 0.84712 | 0.00837 | 0.00721 | 0.69205 | 0.01427 |
| 0.88007 | 0.00598 | 0.00498 | 0.61945 | 0.01589 |
| 0.9118 | 5.25851E-4 | 5.4066E-4 | 0.58998 | 0.00159 |
| 0.93698 | 0.00204 | 0.00204 | 0.50101 | 0.00792 |
| 0.97142 | 0.00151 | 0.00149 | 0.3233 | 0.01015 |
| 0.99054 | 4.05516E-4 | 4.09208E-4 | 0.15195 | 0.00476 |

**S31. Galactose Hygroscopicity**

**Fig S31.1**: Hygroscopicity of (Sigma Aldrich ≥ 99 %), at 293.15 K. Open squares, these experiments; solid line, UNIFAC model. Data taken at RHs lower than indicated by the dashed black line show increased error in hygroscopicity retrieval due to the imposition of a kinetic limitation on water transport.

[Figure]

**Table SI.31.1:** Pure component refractive index ($n_{melt}$) is determined using molar refraction, assuming ideal mixing for calculation of the melt density ($\rho_{melt}$), from bulk data available in Cai et al. (2016). The variation of density as a function of the root of solute mass fraction ($MFS^{1/2}$ =x) is represented by polynomial fit parameters. *Upper* and *lower* refer to 95 % confidence limits for fits to experimental data, (Section 2.2 in manuscript).

| | $n_{melt}$ | $\rho_{melt}$/ g.cm$^{-3}$ | Polynomial fit ( $\rho_{sol} = a + b_1x + b_2x^2 + b_3x^3$) | | | |
|---|---|---|---|---|---|---|
| | | | a | $b_1$ | $b_2$ | $b_3$ |
| *Best* | 1.5885 | 1.6306 | 997.36 | 403.27 | 83.09 | 150.11 |
| *Upper* | 1.5892 | 1.6351 | 996.67 | 165.3 | -284.07 | 752.22 |
| *Lower* | 1.5878 | 1.6261 | 997.37 | 399.69 | 83.4 | 145.36 |

**Table S31.2:** Tabulated experimental data points shown in **Fig S31.1.**

| $a_w$ | error $a_w$ (+ve) | error $a_w$ (-ve) | MFS | error MFS |
|---|---|---|---|---|
| 0.50996 | 0.00287 | 0.0034 | 0.82372 | 0.00382 |
| 0.60189 | 0.00267 | 0.00323 | 0.7993 | 0.00405 |
| 0.63684 | 0.00839 | 0.01012 | 0.76963 | 0.0055 |
| 0.72183 | 0.0016 | 0.00199 | 0.69438 | 0.00194 |
| 0.76282 | 0.00662 | 0.00694 | 0.68348 | 0.01289 |
| 0.80226 | 0.02704 | 0.02704 | 0.6317 | 0.02723 |
| 0.84064 | 0.00138 | 8.91966E-4 | 0.572 | 0.00141 |
| 0.88152 | 0.00559 | 0.00561 | 0.51157 | 0.01025 |
| 0.92485 | 0.00483 | 0.00491 | 0.43437 | 0.01532 |
| 0.96504 | 0.00377 | 0.00374 | 0.29773 | 0.01536 |
| 0.99822 | 0.00115 | 7.88489E-4 | 0.09505 | 0.00656 |

**S32 Xylose Hygroscopicity**

**Fig S32.1**: Hygroscopicity of (Sigma Aldrich ≥ 99 %), at 293.15 K. Open squares, these experiments; solid line, UNIFAC model.

[Figure]

**Table S32.1:** Pure component refractive index ($n_{melt}$) is determined using molar refraction, assuming ideal mixing for calculation of the melt density ($\rho_{melt}$), from bulk data available in Cai et al. (2016). The variation of density as a function of the root of solute mass fraction (MFS$^{1/2}$ =x) is represented by polynomial fit parameters. *Upper* and *lower* refer to 95 % confidence limits for fits to experimental data, (Section 2.2 in manuscript).

|  | $n_{melt}$ | $\rho_{melt}$/ g.cm$^{-3}$ | Polynomial fit ( $\rho_{sol} = a + b_1x + b_2x^2 + b_3x^3$) | | | |
|---|---|---|---|---|---|---|
|  |  |  | a | $b_1$ | $b_2$ | $b_3$ |
| *Best* | 1.5615 | 1.5626 | 996.73 | 127.69 | -163.53 | 597.09 |
| *Upper* | 1.5619 | 1.5653 | 996.74 | 126.37 | -159.45 | 591.57 |
| *Lower* | 1.5611 | 1.5598 | 996.72 | 128.97 | -167.5 | 602.42 |

**Table S32.2:** Tabulated experimental data points shown in **Fig S32.1.**

| $a_w$ | error $a_w$ (+ve) | error $a_w$ (-ve) | MFS | error MFS |
|---|---|---|---|---|
| 0.97404 | 0.00732 | 0.00429 | 0.1841 | 0.0233 |
| 0.98465 | 0.00361 | 0.00212 | 0.12356 | 0.01215 |
| 0.996 | 0.00127 | 7.43479E-4 | 0.02995 | 0.00361 |
| 1.00081 | 0.00148 | 8.71845E-4 | 0.01372 | 0.0012 |

**S33 2,3-Dimethyl Succinic Acid Hygroscopicity**

**Fig S33.1**: Hygroscopicity of 2,3-dimethyl succinic acid (Sigma Aldrich ≥ 99 %), at 293.15 K. Open squares, these experiments; solid line, UNIFAC model. (Density treatment for 2,2-dimethyl succinic acid used.)

[Figure]

**Table S33.2:** Tabulated experimental data points shown in **Fig S33.1.**

| $a_w$ | error $a_w$ (+ve) | error $a_w$ (-ve) | MFS | error MFS |
|---|---|---|---|---|
| 0.94132 | 5.11673E-4 | 5.12405E-4 | 0.38395 | 0.00207 |
| 0.95214 | 0.00144 | 0.00144 | 0.32979 | 0.00859 |
| 0.96262 | 0.00159 | 0.00159 | 0.26369 | 0.01065 |
| 0.97285 | 0.00138 | 0.00138 | 0.19135 | 0.01011 |
| 0.98303 | 0.001 | 0.001 | 0.11733 | 0.00731 |
| 0.99417 | 2.09751E-4 | 2.24291E-4 | 0.03301 | 0.00121 |
| 0.99844 | 2.59195E-4 | 4.09162E-4 | 0.01724 | 4.61378E-4 |

**S34 Dimethyl Malonic Acid Hygroscopicity**

**Figure S34.1**: Hygroscopicity of (Sigma Aldrich 98 %), at 293.15 K. Open squares, these experiments; solid line, UNIFAC model. (Density treatment for methyl succinic acid used.)

[Figure]

**Table S34.2:** Tabulated experimental data points shown in **Figure S34.1.**

| $a_w$ | error $a_w$ (+ve) | error $a_w$ (-ve) | MFS | error MFS |
|---|---|---|---|---|
| 0.71262 | 0.00362 | 0.00449 | 0.7136 | 0.00301 |
| 0.744 | 0.0141 | 0.0141 | 0.69155 | 0.01343 |
| 0.78481 | 0.01088 | 0.01348 | 0.65412 | 0.00614 |
| 0.81516 | 0.01647 | 0.00985 | 0.62813 | 0.01311 |
| 0.83412 | 0.00246 | 0.00229 | 0.60844 | 0.00357 |
| 0.86818 | 0.00422 | 0.00426 | 0.5554 | 0.00729 |
| 0.90119 | 0.00509 | 0.00506 | 0.48761 | 0.01203 |
| 0.92833 | 0.00366 | 0.00365 | 0.40593 | 0.01475 |
| 0.96965 | 0.00157 | 0.00194 | 0.2089 | 0.01089 |
| 0.9897 | 4.75033E-4 | 4.76981E-4 | 0.05824 | 0.00271 |

**S35 Aspartic Acid Hygroscopicity**

**Fig S35.1**: Hygroscopicity of aspartic acid (Sigma Aldrich ≥ 99 %), at 293.15 K. Open squares, these experiments; solid line, UNIFAC model. (Density treatment for alanine used)

[Figure]

**Table S35.1:** Tabulated experimental data points shown in **Fig S35.1.**

| $a_w$ | error $a_w$ (+ve) | error $a_w$ (-ve) | MFS | error MFS |
|---|---|---|---|---|
| 0.99507 | 0.00448 | 0.00375 | 0.01431 | 7.18E-04 |
| 0.99599 | 0.00202 | 0.0017 | 0.01223 | 6.83E-04 |
| 0.99697 | 0.00141 | 0.00118 | 0.00882 | 5.15E-04 |
| 0.99793 | 0.00111 | 9.28E-04 | 0.00594 | 3.01E-04 |
| 0.99891 | 0.001 | 8.39E-04 | 0.00381 | 1.64E-04 |
| 0.99985 | 9.52E-04 | 7.98E-04 | 0.00266 | 8.72E-05 |

**S36 Asparagine Hygroscopicity**

**Fig S36.1**: Hygroscopicity of asparagine (Sigma Aldrich ≥ 98 %), at 293.15 K. Open squares, these experiments; solid line, UNIFAC model. (Density treatment for alanine used)

[Figure]

**Table S36.1:** Tabulated experimental data points shown in **Figure S36.1.**

| $a_w$ | error $a_w$ (+ve) | error $a_w$ (-ve) | MFS | error MFS |
|---|---|---|---|---|
| 0.53409 | 0.00178 | 0.00213 | 0.77577 | 0.00129 |
| 0.62935 | 0.00189 | 0.0023 | 0.74326 | 0.00101 |
| 0.63444 | 0.00381 | 0.00465 | 0.74081 | 0.00101 |
| 0.71441 | 0.00113 | 0.0014 | 0.68254 | 0.00175 |
| 0.74237 | 0.007 | 0.00854 | 0.67146 | 0.00782 |
| 0.81123 | 8.45796E-4 | 8.49613E-4 | 0.61254 | 0.00185 |
| 0.85278 | 0.00812 | 0.00813 | 0.54286 | 0.03203 |
| 0.9048 | 0.00102 | 9.46055E-4 | 0.46853 | 0.00454 |
| 0.94641 | 0.00108 | 0.0011 | 0.3002 | 0.00693 |
| 0.9951 | 2.80427E-4 | 2.96722E-4 | 0.02083 | 0.00124 |

**S37 Errors in Density and Refractive Index Parametrisations and their Impact on Hygroscopicity**

**Fig S37.1 Parametrisation for (a) density based on ideal mixing and bulk measured values for density up to the solubility limit and (b) refractive index predicted beyond the solubility limit using molar refraction. In both (a) and (b) dashed lines indicate the uncertainty envelope in the parametrisations. All bulk experimental values of aqueous density and refractive index are available in the supplementary information of Cai et al. (2016). In (c) measured equilibrium hygroscopicity curves are presented with upper and lower error envelope arising from the uncertainties in density and refractive index which is too small to be obvious.**

[Figure]

**S38 ΔMFS for Simple Straight Chain Dicarboxylic Acids**

**Fig S38.1 The difference in mass fraction of solute (ΔMFS) between values predicted by UNIFAC and experimental values (a) oxalic acid, (b) malonic acid, (c) succinic acid and (d) glutaric acid.**

[Figure]

**S39 Viscosity, Diffusion Constant and Timescale of Diffusional Mixing**

The kinetic modelling framework used in the analysis of the droplet evaporation events is valid only in the absence of a bulk-kinetic limitation on near surface composition, i.e. the particle must be assumed to be homogeneous in composition. Such a limitation was obvious for hygroscopicity measurements of trehalose, galactose and sorbitol at RH's lower than 80 %. To ensure the measurements are not compromised by bulk diffusion, we consider two important factors.

Firstly, the impact of viscosity on the hygroscopicity retrievals becomes very obvious when we consider the consistency and uncertainty in the raw hygroscopic growth curves determined from different droplets evaporating into differing RHs. Droplets drying into different RHs reach different compositions at different times, and will retain different amounts of water because of different drying rates. This leads to an artificially low MFS at a particular RH which then slowly returns to the equilibrium curve overtime. Thus, an inconsistency is apparent between retrieved hygroscopic growth curves (or MFS vs $a_w$) when drying into different RHs. An example of this is shown in Figure S39.1, where we report unbinned hygroscopicity data for alanine (a non-viscous amino acid) and trehalose (viscous at RHs lower than 80%). It is clear here that the different portions of the hygroscopic curves retrieved from measurements at different RHs are consistent for alanine but not for trehalose. A further easy way to identify this retention of water in a particle that is not fully

equilibrated is simply to measure the much longer time-dependence in size once the initial evaporation of water has stopped. In droplets that have reached a bulk diffusion limitation, the existence of a kinetic limitation is apparent in a steadily decreasing size as water continues to leave over a timescale longer than 10 s.

**Fig S39.1 a) Unbinned hygroscopicity data for the compound alanine.  b) Unbinned hygroscopicity data for the compound trehalose. At 50 % RH trehalose has a viscosity of 3.8 x $10^5$ Pa.s (Song et al. 2016).**

[Figure]

Secondly, we can determine the expected conditions under which we might expect problems to arise in retrieving hygroscopic growth curves from an evaporation measurement. Considering again trehalose at 80 % RH, an aqueous-trehalose droplet has a viscosity of 0.5 Pa.s, increasing to $3.8 \times 10^5$ Pa.s at 50 % RH (Song et al. 2016). Therefore, as the RH of the gas phase for the evaporation measurement is lowered, we can expect the increasing viscosity/decreasing diffusivity to become increasingly important. By contrast, for aqueous-carboxylic acid droplets, the viscosity never gets above 1 Pa s even at the driest RHs considered here (Song et al. 2016).

With these known dependencies of viscosity on water activity, we can estimate the timescale for diffusional mixing within a droplet, assuming that this provides an estimate of the timescale for an evaporating droplet to form a homogeneous mixture. This timescale must be considerably shorter than the evaporation timescale for our hygroscopicity estimations to be valid. First, the Stokes-Einstein equation is used to estimate the diffusion constant of water at varying viscosity (varying RH).

$$D = \frac{k_B T}{6\pi r_{mol}\eta} \qquad (1.1)$$

$D$ is the diffusion constant, $k_B$ is the Boltzmann constant, $T$ is temperature, $r_{mol}$ is the molecular radius of water (taken as 1.375 Å) and $\eta$ is the viscosity. It should be noted that equation (1.1) is likely to provide a significant underestimate of the diffusion constant due to the failure of the Stokes-Einstein equation. At a viscosity of 100 Pa s, the diffusion constant for water in sucrose is already more than one order of magnitude larger than estimated from the viscosity (Power et al. 2013). However, using diffusion constants estimated from (1.1) will provide an upper limit on the diffusional mixing timescale. The timescale for diffusional mixing, τ, is then estimated using the expression

$$\tau = \frac{a^2}{\pi^2 D} \qquad (1.2)$$

where $a$ is the droplet radius (set as 10 microns in this calculation).

We compare the diffusional mixing timescales for aqueous droplets of trehalose, NaCl, NaNO₃ and glutaric acid in the newly added supplemental Figure S39.2 (and repeated below). Given that we have been able to report accurate hygroscopic growth curves for NaNO₃ down to 50 % RH (see Rovelli et al. 2016 and the

response to referee 2), it is clear that a final viscosity at 50 % of ~ 0.1 Pa.s (Baldelli et al.) is insufficient to impede accurate measurement of the hygroscopicity. Indeed, this suggests that water transport in any aerosol droplet that maintains a viscosity lower than 0.1 Pa.s during drying should remain sufficiently fast to avoid a bulk diffusion limitation, permitting accurate hygrosocpicity measurements. As an example of the diacarboyxlic acids considered in this study, glutaric acid has a considerably lower viscosity at 50 % RH of ~ 0.01 Pa.s (Song et al. 2016), indicative of what we might expect for all such similar systems. By contrast, aqueous-trehalose droplets cross the 0.1 Pa.s viscosity threshold at a water activity of ~0.85 (Song et al. 2016), commensurate with the deviation and increased scatter in the hygroscopicity measurements reported above for this compound.

Based on the two considerations above and to indicate clearly the water activity ranges over which we consider the hygroscopicity measurements to be valid for trehalose (S30), galactose (S31) and sorbitol (S29), we have added a dashed line to indicate where the data appear to become kinetically limited. We have added the following words to the captions of these Figures: "Data taken at RHs lower than indicated by the dashed black line show increased error in hygroscopicity retrieval due to the imposition of a kinetic limitation on water transport."

**Fig S39.2 a) Viscosity of Trehalose, NaCl, NaNO3 and Glutaric Acid as a function of RH. b) Estimated diffusion constant as a function of RH. c) Timescale for diffusional mixing at the RH shown on x-axis. Dashed green line represents 1 second timescale for diffusional mixing.**

[Figure]

A. Baldelli, R. M. Power, R. E. H. Miles, J. P. Reid and R. Vehring *Effect of crystallization kinetics on the properties of spray dried microparticles,* Aerosol Science and Technology, 2016, 50:7, 693-704, DOI:10.1080/02786826.2016.1177163

R. M. Power, S. H. Simpson, J. P. Reid and A. J. Hudson, *The transition from liquid to solid-like behaviour in ultrahigh viscosity aerosol particles,* Chemical Science, 2013, 4 , 2597, DOI: 10.1039/c3sc50682g

Y. Chul Song, A. E. Haddrell, B. R. Bzdek, J. P. Reid, T. Bannan, D. O. Topping,, C. Percival, and C. Cai *Measurements and Predictions of Binary Component Aerosol Particle Viscosity* J. Phys. Chem. A 2016, 120, 8123−8137, DOI: 10.1021/acs.jpca.6b07835

**Table S40.0: Table of UNIFAC groups for cyclic and open chain galactose and xylose.**

| Compound | Open Chain (In Manuscript) | Cyclic |
|---|---|---|
| Galactose | $CHO\ (CH_1^{(OH)})_4\ CH_2^{(alc)}\ (OH)_5$ | $(CH^{[alc]})_4(CH_2^{[OH]})(CHO^{[ether]})(OH)_4$ |
| Xylose | $(CH_2(OH))_3\ CH_2^{(alc)}\ CHO\ (OH)_4$ | $(CH^{[OH]})_4(CHO^{[ether]})(OH)_4$ |

**Figure S40.0 Galactose and Xylose CK-EDB data as a function of MFS and water activity compared with predictions for both cyclic and open chain UNIFAC group thermodynamic predictions**

[Figure]